# The maximum-average subtensor problem: equilibrium and out-of-equilibrium properties

Vittorio Erba[1], Nathan Malo Kupferschmid[1], Rodrigo Pérez Ortiz[2], Lenka Zdeborová[1]

[1] Statistical Physics of Computation Laboratory,
École polytechnique fédérale de Lausanne (EPFL) CH-1015 Lausanne

[2] Department of Mathematics,
Alma Mater Studiorum, Università di Bologna (Unibo), IT-40126 Bologna

**Abstract**

In this paper we introduce and study the Maximum-Average Subtensor ($p$-MAS) problem, in which one wants to find a subtensor of size $k$ of a given random tensor of size $N$, both of order $p$, with maximum sum of entries. We are motivated by recent work on the matrix case of the problem [1, 2, 3] in which several equilibrium and non-equilibrium properties have been characterized analytically in the asymptotic regime $1 \ll k \ll N$, and a puzzling phenomenon was observed involving the coexistence of a clustered equilibrium phase and an efficient algorithm which produces submatrices in this phase. Here we extend previous results on equilibrium and algorithmic properties for the matrix case to the tensor case. We show that the tensor case has a similar equilibrium phase diagram as the matrix case, and an overall similar phenomenology for the considered algorithms. Additionally, we consider out-of-equilibrium landscape properties using Overlap Gap Properties and Franz-Parisi analysis, and discuss the implications or lack-thereof for average-case algorithmic hardness.

In recent years, our understanding of typical-case algorithmic hardness, i.e. the failure of efficient algorithms in solving problems in typical (non-worst-case) instances, has grown considerably. Based on the nature of the problem, i.e. optimization of random landscapes, inference of hidden signals, detection of hidden signals, etc..., we have several theoretical frameworks that can be deployed to argue about hardness: Approximate Message Passing, Overlap Gap Property, Sum of Squares, Statistical Query, and Low Degree polynomials; see for e.g. [4, 5, 6, 7, 8]. Each of these frameworks has a different "intuitive" reason for why hardness arises. For example, the Low Degree framework argues about hardness by showing that algorithms that only perform polynomial operations on the problem's input (a particular class of efficient algorithms) fail, while the Overlap Gap framework studies the topology of the solution set of the problem and links hardness to extreme forms of shattering of such set. Understanding the scopes and limitations of these frameworks, their mutual relationships, and uncovering a unified trigger for typical-case algorithmic hardness is a major open problem in the field.

One way to rephrase the problem of algorithmic hardness stems from statistical mechanics. Indeed, most computational problems can be seen as statistical mechanics systems, whose specific Gibbs-Boltzmann distribution is related to the nature of the problem. Then, algorithms can be seen as out-of-equilibrium dynamical processes, and the question is when such dynamical process can or cannot detect/access given portions of the configuration space of the system in sub-exponential time (when started from uninformed initial conditions).

The connection between the study of algorithmic hardness and out-of-equilibrium statistical mechanics has deep roots. For example, modern mathematical hardness proofs through Overlap Gap Properties (see [7] for a review) have been inspired by early physics works on energy landscape properties [9], and powerful classes of efficient algorithms (i.e. Approximate Message Passing algorithms, see [4] for a review) are, in their simpler forms, iterative schemes for the celebrated Thouless-Anderson-Palmer (TAP) equations [10]. More recently, hardness under the Low Degree framework was linked to a Franz-Parisi-style analysis [11], and the role of run-time of physics-based algorithms was discussed in [12]. Yet, the full extent to which statistical mechanics can shed light on algorithmic properties is not clear.

A prime example of this is given by the Symmetric Binary Perceptron (SBP) problem [13]. This is a constraint satisfaction problem, where one is given $n$ random points $\xi^\mu$, $\mu = 1, \ldots, n$ on the $d$-dimensional hypercube, and the objective is to find a vector $w$ also on the hypercube such that $|\xi^\mu \cdot w| \geq \kappa\sqrt{d}$ for a given parameter $\kappa > 0$. It is known rigorously [14] that the typical solutions to this problem are isolated, i.e. each pair of them is at Hamming distance $\mathcal{O}(d)$. This clustering of the equilibrium solution space can be rationalized within statistical mechanics as a frozen one-step replica-symmetry-broken phase (frozen 1-RSB), long conjectured to imply algorithmic hardness [15, 16] (and surely implying the failure of equilibrium sampling algorithms). Yet, it is known that efficient algorithm can find solutions to this problem [17]. This can be understood in terms of out-of-equilibrium properties. In a series of works [18, 19, 20] and the pedagogical review [21], dealing also with a non symmetric version of the problem, it was shown that exponentially rare connected sets of solutions exists in the background of the typical isolated solutions, and it was suggested that these connected sets are the key feature allowing algorithms to work (also in line with insights from the Overlap Gap Property framework, applied to this problem in [22]). This phenomenology is not unique of the SBP, but has been observed also in constraint satisfaction problems over sparse random graphs [23].

It is still unclear whether rare solutions sets can be systematically detected, which of these rare sets and which of their properties favor the functioning of efficient algorithms, and whether this can be rationalized directly within replica formalism (see [24, 25] for a recent attempt). The "local entropy" framework [18] has been proposed to systematically detect rare algorithmically-accessible regions in the SBP and in the more classical non-symmetric version of the problem, but to our knowledge the framework has not been explored thoroughly in other problems, and a lack of analytic algorithmic thresholds in the binary perceptron problem makes it hard to confirm or rule out the effectiveness of this method to detect hardness. Parallely, a powerful version of the OGP framework based on the so called "branching OGP" has been successfully used to argue for algorithmic hardness in, for e.g., the optimization of mean-field spin glass Hamiltonians [26], in random constraint satisfaction problems [27] and in the random graph alignment problem [28]. Branching OGP is based on showing that certain ultra-metric trees of configurations with specified overlap profiles can be constructed by efficient algorithms, but are not present in the hard phase. Notice the natural affinity between this idea and equilibrium full-RSB [29].

Recently, a similar phenomenology (frozen 1-RSB phase which efficient algorithms provably enter) has been observed in another problem, the well-studied Maximum-Average Submatrix (MAS) problem [30, 1, 2, 3, 31, 32]. In this problem, one considers a $N \times N$ random matrix $J$, and wants to find the $k \times k$ submatrix of $J$ with maximum average of entries. In the thermodynamic limit of $k, N \to \infty$ with $k \ll N$, it has been argued that the model is in a frozen 1-RSB phase at equilibrium, yet it has been proven that efficient algorithms can find solutions within this strongly clustered phase. Remarkably, both the equilibrium [3] and algorithmic thresholds [1, 2] are available analytically, which is not at all the case for the other examples mentioned above, where algorithmic thresholds are available only empirically (and thus they are affected by finite-size scaling), and statistical mechanics properties are accessible only through the numerical solution of non-linear systems of equations (called replica state equations). This makes the MAS problem a special testbed to explore analytically the relationship between equilibrium and non-equilibrium properties, and algorithmic hardness.

Motivated by this discussion, in this paper we study equilibrium and non-equilibrium properties of a generalized version of the MAS problem, i.e. the Maximum-Average Subtensor ($p$-MAS) problem. In this problem, one considers an order $p$ random tensor $J \in (\mathbb{R}^N)^{\otimes p}$, and wants to find the subtensor of $J$ with size $k \times \cdots \times k$ with maximum average of entries. This problem interpolates between the matrix case at $p = 2$, and a Random Energy Model for $p \to +\infty$, allowing to tune the degree of correlations present in the system. Our main results are the following.

- **Equilibrium phase diagram.** We characterize the equilibrium phase diagram of the $p$-MAS problem in the limit $1 \ll k \ll N$ for all integer values of the tensor rank $p \geq 2$. We extend the analysis of the matrix case $p = 2$ reported in [3] to higher-order tensors, and find that for all tensor orders $p$ the phase diagram is qualitatively the same, featuring an replica symmetric (RS) phase at small values of the subtensor average, and a frozen one-step replica-symmetry-broken (frozen 1-RSB) phase at larger values of the subtensor average. In this phase, subtensor average level sets are dominated by exponentially many isolated configurations. We also derive analytically the value of the subtensor average at which the phase transition happens, as well as the maximum value of the subtensor average achievable.

- **Algorithmic thresholds.** We derive analytically the asymptotic value of the subtensor average achieved by two greedy algorithms already considered in the literature for the matrix case $p = 2$, the Largest Average Submatrix $\mathcal{LAS}$ [1, 2] and the Incremental Greedy Procedure $\mathcal{IGP}$ [2]. For the $\mathcal{LAS}$ algorithm, we argue that the proof of [1] extends naturally to the tensor case, while for the $\mathcal{IGP}$ algorithm we adapt a heuristic argument given in [2]. We find that the $\mathcal{IGP}$ algorithm reaches subtensor averages well inside the equilibrium frozen 1-RSB phase.

- **Overlap gap property.** We compute an algorithmic hardness upper bound base on the Overlap Gap Property (OGP). This extends a similar analysis in the matrix case $p = 2$ of [2] to the tensor case, and to multi-replica overlap gap analysis. We find that also in the tensor case there exists a gap between the OGP upper bound and the performance of the $\mathcal{IGP}$ algorithm, meaning that more refined landscape arguments are required to understand whether the $\mathcal{IGP}$ algorithm is optimal or not. We also briefly discuss the technical difficulties linked to computing quenched OGP entropies (linked to the local entropy analysis) within the replica formalism for the $p$-MAS problem.

- **Franz-Parisi analysis.** Inspired by analogous work [33, 34, 35] in the context of the binary perceptron problems, we finally perform a Franz-Parisi [36] analysis of the matrix $p = 2$-MAS problem. The Franz-Parisi analysis allows to probe some exponentially rare configurations in submatrix average level sets. We find a rich phenomenology of clusters of rare configurations in the frozen 1-RSB phase, but cannot identify any specific property that seems to appear/break at the submatrix average reached by the $\mathcal{IGP}$ algorithm. We conclude that the $\mathcal{IGP}$ algorithm is exploring a different set of non-equilibrium configuration than those accessible with the Franz-Parisi analysis.

Table 1 subs up the main thresholds we derive in the paper. The Franz-Parisi analysis is more complex, and cannot be reduced to thresholds: see Section 5 for our results.

| Model | Freezing $= \mathcal{LAS}$ | $\mathcal{IGP}$ | $p$-body OGP | Max avg |
|---|---|---|---|---|
| Symm $p$-MAS | $2/\sqrt{p}$ | $4p/[(p+1)\sqrt{p}]$ | $a_{\mathrm{OGP}}^{\mathrm{ann}}(p)$, Eq. (44) | 2 |
| Non-symm $p$-MAS | $2/\sqrt{p!}$ | $4p/[(p+1)\sqrt{p!}]$ | $a_{\mathrm{OGP}}^{\mathrm{ann}}(p)/\sqrt{(p-1)!}$, Eq. (44) | $2/\sqrt{(p-1)!}$ |

Table 1: Thresholds derived in the paper for the $p$-MAS problem, in a symmetric and non-symmetric random setting (see Section 1). Freezing denotes the 1-RSB dynamic transition to a clustered phase, $\mathcal{LAS}$ and $\mathcal{IGP}$ are the two greedy algorithms considered in the literature (Section 3), "$p$-body OGP" denotes the upper bound we obtain on algorithmic harness through annealed multi-body OGP (Section 4), and "Max avg" denotes the maximum subtensor average achievable. All thresholds are sorted in increasing order from left to right.

**Perspectives.** Natural directions to extend our work involve exploring analytically quenched OGPs and their relationship with local entropy, and considering more complicated OGPs such as the branching OGP. Further, one wonders if other hardness frameworks, such as Low Degree Polynomials and Sum-Of-Squares could be meaningfully deployed on the $p$-MAS problem, again with the underlying idea of using the $p$-MAS problem as an analytically solvable testbed to compare different techniques. Finally, understanding which subclass of efficient algorithms do not work in equilibrium frozen 1-RSB phases (besides those that respect in some sense the detailed balance) would be interesting.

**Concurrent work.** While we were finalizing this manuscript, we became aware of independent work studying algorithmic hardness in the $p$-MAS problem [37]. In that work, the authors prove an upper-bound on the performance of stable and online algorithms (including $\mathcal{LAS}$ and $\mathcal{IGP}$) through a branching OGP computation. They also provide a matching lower-bound by analyzing the performance of the $\mathcal{IGP}$ algorithm and finding results compatible with ours (see Section 3). Thus, the question of whether the $\mathcal{IGP}$ algorithm is optimal is settled (it is), and the branching OGP proves to be once more the correct landscape tool to probe for hardness. We still find surprising that such a simple greedy algorithm saturates the algorithmic upper-bound. We stress that our results on the equilibrium phase diagram, on the $\mathcal{LAS}$ performance coinciding with the freezing threshold, and the non-equilibrium Franz-Parisi analysis are still completely novel,

and shed light on the cost landscape in the algorithmically feasible phase. On the other hand, our results on the annealed OGP bounds (Section 4) become more technical curiosities in the light of the more powerful branching OGP bound.

**Organization of the manuscript.**  The manuscript is organized as follows. In Section 1 we define the $p$-MAS problem and highlights the mapping onto a Boolean $p$-spin model. In Section 2 we present the replica equilibrium analysis for the $p$-MAS problem, and discuss the phase diagram for $k \ll N$. In Section 3 we considers previously studied greedy algorithms for the matrix case, and analyze their performance in the tensor problem. In Section 4 we discuss overlap gap properties, and derive the annealed multi-body OGP thresholds for the $p$-MAS problem. In Section 5 we present the Franz-Parisi analysis of the matrix $p = 2$-MAS problem, and discuss the apparent lack of algorithmic insights given by this framework.

# 1 The $p$-MAS problem and its mapping onto a spin model

## 1.1 Preliminary definitions

Consider a rank-$p$ tensor $J \in \mathbb{R}^{N_1 \times \cdots \times N_p}$ with $p \geq 2$. A sub-tensor $\sigma$ of $J$ is obtained by deleting any number of rows of $J$ along each of its $p$ dimensions. A compact way to represent sub-tensors is to use $p$ Boolean vectors $\sigma^{(g)} \in \{0,1\}^{N_g}$ with $g = 1, \ldots, p$. Each boolean vector identifies, along dimension $g$, which rows $i$ have been deleted ($\sigma_i^{(g)} = 0$) and which rows have not ($\sigma_i^{(g)} = 1$). Then, the tensor

$$(\sigma^{(1)} \otimes \cdots \otimes \sigma^{(p)})_{i_1, \ldots, i_p} = \sigma_{i_1}^{(1)} \ldots \sigma_{i_p}^{(p)} \tag{1}$$

acts as a masking tensor identifying the subtensor $\sigma$, with all components equal to zero apart from those included in the subtensor, which equal one. The average of the entries of the subtensor $\sigma$ can then be written compactly as

$$\mathrm{avg}(\sigma) = \frac{1}{N_1 \ldots N_p} \sum_{i_1=1}^{N_1} \cdots \sum_{i_p=1}^{N_p} \sigma_{i_1}^{(1)} \ldots \sigma_{i_p}^{(p)} J_{i_1, \ldots, i_p} . \tag{2}$$

In the following we will be interested in square tensors, where $N_g = N$ for all $g = 1, \ldots, p$, and square sub-tensors, i.e. with $N - k$ deleted rows along each of the $p$ dimensions. The general "rectangular" case comes with several additional hyper-parameters to keep under control, and we believe that its phenomenology will not be qualitatively different from the square case (see [3, SM Sec. VII] for a discussion of the $p = 2$ rectangular case). For these reasons, we leave this case for future work.

We will consider two random versions of the square problem.

- *Sub-tensors of a random tensor (non-symmetric case):* $J$ is a random tensor with independent standard Gaussian entries (mean zero and variance one). Any square sub-tensor is allowed.

- *Principal sub-tensors of a random symmetric tensor (symmetric case):* $J$ is a symmetric random tensor with independent standard Gaussian entries (mean zero and variance one) for all the components $J_{i_1, \ldots, i_p}$ for $1 \leq i_1 < \cdots < i_p \leq N$, and all other components obtained by symmetry. The diagonals (all entries in which at least two indices are equal) will not matter in the limit $N \to \infty$, and they can be put to zero. Only principal sub-tensors are allowed, i.e. sub-tensors that have the same deleted rows along each of the $p$ dimensions. Notice that principal sub-tensors can be represented by a single Boolean vector $\sigma \in \{0,1\}^N$, as all of the $p$ dimensions share the same deleted rows.

The distinction between the non-symmetric and the symmetric cases was already highlighted in [3], where we showed that the symmetric case maps naturally onto a spin model with reciprocal interactions, while the non-symmetric maps onto a multi-species spin model. We also showed that the phase diagram of the non-symmetric case reduces, under some specific assumption, to the one of the symmetric case. We will define the $p$-MAS problem and its mapping onto a spin model for both cases, and then focus our attention to the symmetric case only commenting about the non-symmetric one when deemed useful.

We are interested in characterizing the behavior of the sub-tensor average of sub-tensors $\sigma$ of size $k$. The sub-tensor average is defined as

$$\text{avg}(\sigma) = \frac{1}{k^p} \sum_{i_1,\ldots,i_p=1}^{N} \sigma_{i_1}^{(1)} \ldots \sigma_{i_p}^{(p)} J_{i_1 \ldots i_p} \,, \tag{3}$$

i.e. as the average of the entries of the sub-tensors $\sigma$.

## 1.2 Statistical mechanics framework

The $p$-MAS problem is defined, in the symmetric and non-symmetric cases, as the problem of finding the subtensor of largest average of its entries. To study its properties, it is useful to compute the number of sub-tensors of size $k$ with a given sub-tensor average (and in particular the maximum sub-tensor average achievable) and to understand the geometric structure of the sub-tensor average level sets (in order to study non-equilibrium algorithmic properties). Thus, we introduce an associated Gibbs measure

$$\text{Prob}(\sigma) = \exp\left(\beta E(\sigma) + N\beta h\, m(\sigma)\right)/Z \tag{4}$$

where $Z$ is the partition function / normalization factor, and for the symmetric problem $\sigma \in \{0,1\}^N$,

$$E_{\text{symm}}(\sigma) = \sqrt{\frac{p!}{N^{p-1}}} \sum_{i_1 < \cdots < i_p} \sigma_{i_1} \ldots \sigma_{i_p} J_{i_1 \ldots i_p} \,,$$
$$m_{\text{symm}}(\sigma) = \frac{1}{N} \sum_{i=1}^{N} \sigma_i \,, \tag{5}$$

while for the non-symmetric problem $\sigma \in \left(\{0,1\}^N\right)^{\otimes p}$,

$$E_{\text{non-symm}}(\sigma) = \sqrt{\frac{1}{N^{p-1}}} \sum_{i_1,\ldots,i_p} \sigma_{i_1}^{(1)} \ldots \sigma_{i_p}^{(p)} J_{i_1 \ldots i_p} \,,$$
$$m_{\text{non-symm}}(\sigma) = \frac{1}{pN} \sum_{g=1}^{p} \sum_{i=1}^{N} \sigma_i^{(g)} \,. \tag{6}$$

Notice that in both cases the energy $E(\sigma)$ is a multiple of the sub-tensor average so that by tuning $\beta$ we can scan across the level sets of the sub-tensor average. In particular we have

$$\text{avg}(\sigma) = N^{-(p+1)/2} m^{-p} E_{\text{non-symm}}(\sigma) = \sqrt{p!}\, N^{-(p+1)/2} m^{-p} E_{\text{symm}}(\sigma) \,. \tag{7}$$

Moreover, the energy is normalized such that in both cases it has extensive covariance

$$\mathbb{E}_J E(\sigma) E(\tau) = N \prod_{g=1}^{p} \frac{1}{N} \sum_{i=1}^{N} \sigma_i^{(g)} \tau_i^{(g)} = N \prod_{g=1}^{k} q\left(\sigma^{(g)}, \tau^{(g)}\right) \,, \tag{8}$$

where we introduced the overlap $q(\sigma, \tau)$ between two $N$-dimensional Boolean vectors as

$$q(\sigma, \tau) = \frac{1}{N} \sum_{i=1}^{N} \sigma_i \tau_i \,, \tag{9}$$

and we remark that $0 \leq q(\sigma, \tau) \leq k/N$ if $\sigma, \tau$ have at most $k$ non-zero entries. Notice also that we defined the Gibbs measure with a $+\beta$ inverse temperature. This is due to the maximization nature of the problem, but we notice that due to the distributional symmetry $J \to -J$ the sign of the inverse temperature will not have any effect on the properties of the problem.

The external field $h$ can instead be used to tune the sub-tensor size $k$, as the corresponding magnetization satisfies $m(\sigma) = k/N$ if sub-tensor $\sigma$ has size $k$ in both the symmetric and non-symmetric case.

## 1.3 Observables

As usual in mean-field disordered systems [29], we expect that the free-entropy $\Phi(J) = N^{-1} \log Z$ is asymptotically self-averaging, i.e. its variance w.r.t. the disorder $J$ goes to zero as $N \to \infty$. Self-averaging allows us to study the properties of typical realizations of the disorder $J$ by computing the averaged free entropy $\Phi = \mathbb{E}_J N^{-1} \log Z$. Once $\Phi$ is computed, we can then fix $\beta$ by requiring that the average energy density equals $e$ and fix $h$ by requiring that the average magnetization equals $m$, i.e.

$$\partial_\beta \Phi \overset{!}{=} e, \qquad \partial_h \Phi \overset{!}{=} m. \tag{10}$$

Finally, the entropy density $s(e, m)$ (i.e. the entropy density associated to the number of sub-tensors with given energy density/sub-tensor average and with given magnetization/size) can be computed by an inverse Legendre transform

$$s(e, m) = \Phi - \beta e - hm. \tag{11}$$

The study of the geometry of the sub-tensor average level sets requires more complicated observables, which will be introduced later in the respective sections. Here we only mention the so-called *complexity* $\Sigma(e, m)$ [38], i.e. the entropy density associated to the number of pure states of the Gibbs measure defined in (4). A non-zero complexity signals a phase where the typical sub-tensors in a given level-set of the sub-tensor average are clustered, i.e. are organized in well separated groups each with internal entropy density

$$s_{\text{int}}(e, m) = s(e, m) - \Sigma(e, m). \tag{12}$$

Finally, it will be useful to introduce the subtensor-average density

$$a(\sigma) = \frac{\sqrt{2} \, \text{avg}(\sigma) \, N^{(p-1)/2}}{\sqrt{p! m^{1-p} \log \frac{1}{m}}} = \text{avg}(\sigma)/T(N, m, p), \tag{13}$$

i.e. the intensive value of the sub-tensor average when we remove the scaling dependency with the size of the tensor $N$, as well as the dependency on the magnetization $m$ in the $m \to 0$ limit (see Section 2.3).

## 1.4 Mapping onto Ising spin models

Looking at (5), it is apparent that the model we are looking at is a close relative of the celebrated Ising $p$-spin model [39], with $E$ being the energy and $m$ being the magnetization of the associated spin glass. The only difference between the usual Ising $p$-spin and the model we are looking at here is the nature of the spins, which are $\{\pm 1\}$ in the Ising model and Boolean $\{0, 1\}$ in the sub-tensor problem, and that we consider the ensemble at fixed magnetization instead of the more usual fixed magnetic field. The non-symmetric model can also be interpreted as an Ising spin model, where now spins are divided in $p$ different "species".

This mapping was already spelled-out in [3] for the matrix $p = 2$ case. As we argued therein, the Ising and Boolean models are not equivalent, as the linear transformation of the spin mapping one into the other introduces a random magnetic field term that is correlated with the main interaction disorder. Nonetheless, we found that the phase diagram is similar between the Ising and Boolean versions of the model for what concerns the level of replica symmetry breaking. On the other hand, the paramagnetic-to-spin-glass crossover in the Boolean model is not sharp, with the overlap parameter analytically increasing as the temperature is lowered (see also [40] for a specific discussion of this point).

We also highlight that, in the $p = 2$ case, there is hysteresis in the $m(h)$ relationship. Indeed, for low enough temperature, the function $h(m)$ is non-invertible, leading to ensemble in-equivalence between the fixed $m$ ensemble and the fixed $h$ ensemble. Thus, due to ensemble in-equivalence, one needs to fix $h$ to obtain a given $m$ at the level of the saddle-point equations, and not *a posteriori* through a usual inverse Legendre transform.

# 2 The equilibrium phase diagram of the $p$-MAS problem

To study the phase diagram of the $p$-MAS problem, we resort to replica theory (see [29] and [41] for a pedagogical treatment). We start by considering the symmetric model, and come back to the non-symmetric one later on.

## 2.1 1-RSB free entropy for the symmetric case

Replica theory involves computing the asymptotic averaged free entropy using the replica trick

$$\Phi = \lim_{N \to \infty} \frac{1}{N} \mathbb{E}_J \log Z = \lim_{N \to \infty} \lim_{n \to 0} \frac{\mathbb{E}_J Z^n - 1}{nN}, \tag{14}$$

then computing $\mathbb{E}_J Z^n$ for integer values of $n$, and finally taking an appropriate analytic continuation of the result to $n \to 0$. While this procedure is to this day a heuristic set of steps, the results derived within replica theory have always been rigorously proved to be correct. Moreover, for the symmetric matrix version of the model at hand a proof originally by Panchenko [42] applies directly [43], guaranteeing the correctness of the replica free entropy for $p = 2$, and suggesting that also for the tensor case replica results will be provable without difficulty. For the non-symmetric model, we are not aware of a proof scheme applying directly, but we suspect that proofs will be more involved due to the non-convexity of the energy covariance (8) w.r.t. the overlaps [44].

We computed the replica free entropy within the 1-RSB scheme, where one assumes that the Gibbs measure is well approximated by a 2-level hierarchy of pure states. This hierarchy is characterized by two overlaps: the average overlap between micro-states belonging to the same pure state $q_1$, and the one between micro-states belonging to different pure states $q_0$. The Parisi parameter $x$ acts as a temperature controlling the trade-off between the free entropy of a single pure state, and the entropy of pure states [29, 38].

More intuitively, one can imagine that pure states at each inverse temperature $\beta$ are in correspondence with clusters of configurations at energy density $e(\beta)$, so that the two-level 1-RSB hierarchy is really describing clustered energy level sets, with overlap between clusters given by $q_0$ and overlap within clusters $q_1$.

In Appendix A we show that

$$\phi_{1-RSB}(m, q_1, q_0, x) = (1 - p)\frac{\beta^2}{2} \left[ m^p + (x - 1)q_1^p - xq_0^p \right] + \frac{1}{x} \int Du \log \int Dv \left( 1 + \exp(\beta H_{u,v}) \right)^x \tag{15}$$

where

$$H_{u,v} = h + \frac{\beta p}{2}(m^{p-1} - q_1^{p-1}) + \sqrt{pq_0^{p-1}} u + \sqrt{p(q_1^{p-1} - q_0^{p-1})} v \tag{16}$$

where $Du$ and $Dv$ denote integration against a standard Gaussian measure. The magnetic field $h$, intra-state overlap $q_1$ and inter-state overlap $q_0$ satisfy the associated saddle-point equations

$$\begin{aligned}
m &= \int Du \frac{\int Dv(1 + \exp(\beta H_{u,v}))^x \ell(\beta H_{u,v})}{\int Dv(1 + \exp(\beta H_{u,v}))^x}, \\
q_1 &= \int Du \frac{\int Dv(1 + \exp(\beta H_{u,v}))^x \ell(\beta H_{u,v})^2}{\int Dv(1 + \exp(\beta H_{u,v}))^x}, \\
q_0 &= \int Du \left[ \frac{\int Dv(1 + \exp(\beta H_{u,v}))^x \ell(\beta H_{u,v})}{\int Dv(1 + \exp(\beta H_{u,v}))^x} \right]^2,
\end{aligned} \tag{17}$$

where $\ell(y) = (1 + e^{-y})^{-1}$ is the logistic sigmoid function. The Parisi parameter $x$ satisfies $x = 1$ if the associated complexity function $\Sigma(x = 1) \geq 0$, and otherwise is set such that $\Sigma(x) = 0$, where the complexity function $\Sigma(x)$ satisfies

$$\Sigma(x) = -x^2 \partial_x \Phi_{1-RSB}(x). \tag{18}$$

The thermodynamic complexity defined in the previous sections is then given by $\Sigma = \max(\Sigma(x = 1), 0)$, computing the entropy density associated to the number of pure states contributing to the Gibbs measure [38].

Within the 1-RSB ansatz, the energy density satisfies

$$e = \partial_\beta \Phi_\beta = \beta \left[ m^p + (x - 1)q_1^p - xq_0^p \right] \tag{19}$$

which is related to the sub-tensor average as defined in (13), and the entropy can be computed as

$$s(e, m) = \Phi - \beta e - hm. \tag{20}$$

For any finite $m$, one could study the phase diagram of the problem by studying the 1-RSB free entropy $\Phi(\beta)$, entropy $s(e)$ and the complexity $\Sigma(e)$, as we did for the case $p = 2$ in [3]. In particular, regions with positive complexity correspond to clustered energy level sets, while the level of replica symmetry breaking of each phase is determined by the local stability of the RSB ansatz [45]. This program involves the numerical solution of (17), which tends to be quite involved mostly due to the fact that the first equation cannot be solved by a fixed point iteration scheme (one needs to solve it for $h$ at fixed $m$). Given that the region of algorithmic interest for this problem in the matrix case $p = 2$ was found to be the limit $m \to 0$ and there dynamic 1-RSB ($x = 1$) was found to be stable in replica space, we limit our attention to this regime, leaving the characterization of the full phase diagram at finite $m$ for future work.

## 2.2 Scaling limit for $m \to 0$

In the limit $m \to 0$, we need to properly rescale our control parameters and observables. Indeed, for small $m$ the infinite temperature entropy (i.e. the number of sub-tensors of size $k = mN$) is of order $\mathcal{O}(Nm \log 1/m)$, while the variance of the energy of a given configuration is of order $\mathcal{O}(Nm^p)$. In order to have a well-defined thermodynamic limit, followed by the $m \to 0$ limit, we need to ensure that the variance of $\beta E$ is of the same order of the infinite-temperature entropy, which can be achieved by rescaling the inverse temperature as

$$\beta = b\sqrt{m^{1-p} \log \frac{1}{m}} \tag{21}$$

where $b$ is the correct inverse temperature in the $m \to 0$ limit. This justifies the scalings of the intensive average given in (13). Additionally, we need to rescale the magnetic field according to the same logic $\mathcal{O}(\beta h m) = \mathcal{O}(m \log 1/m)$, giving

$$h = \eta\sqrt{m^{p-1} \log \frac{1}{m}}. \tag{22}$$

Finally, all overlaps scale as $\mathcal{O}(m)$ (as they are upper bounded by $m$ by definition) and the entropy and complexity scale as $\mathcal{O}(m \log 1/m)$ (the same as the infinite temperature entropy) in the limit.

In the rest of the manuscript, whenever we deal with entropies in the $m \to 0$ limit, we will always rescale them by the natural factor $m \log 1/m$.

## 2.3 The phase diagram for the symmetric case in the $m \to 0$ limit.

Equipped with all these scaling laws, we can solve (17) at leading order in the $m \to 0$ limit, which can be done analytically by computing at leading order in small $m$ all the integrals involved.

Before moving to the subtensor case, let us recall the results obtained for the matrix $p = 2$ case [3]. There, the limiting phase diagram featured three phases:

- For submatrix averages $0 < a < a_{\text{freezing}} = \sqrt{2}$, replica symmetry is not broken, meaning that the Gibbs measure has no hierarchical pure state structure, and typical configurations (under the Gibbs measure) form a connected set. In this phase the complexity is zero, and the total entropy satisfies

$$\frac{s(a)}{m \log 1/m} = 1 - \frac{a^2}{4}. \tag{23}$$

Moreover, the rescaled overlaps satisfy $q_1/m = q_0/m = 0$ in the limit.

- For submatrix averages $a_{\text{freezing}} = \sqrt{2} < a < a_{\text{max}} = 2$, replica symmetry is broken to one step (1-RSB), and we observe a so-called *frozen* 1-RSB phase. The complexity is positive, meaning that the level sets of the submatrix average are composed of exponentially many clusters, and each cluster has vanishing internal entropy, meaning that at leading order each cluster contains $\mathcal{O}(1)$ configurations. The Parisi parameter satisfies $x = 1$. In this phase total entropy and the complexity satisfy

$$\frac{s(a)}{m \log 1/m} = \frac{\Sigma(a)}{m \log 1/m} = 1 - \frac{a^2}{4}. \tag{24}$$

Moreover, the rescaled overlaps satisfy $q_1/m = 1$ and $q_0/m = 0$ in the limit, telling us that equilibrium configurations are isolated (they have intra-cluster average overlap $q_1/m = 0$) and "orthogonal" (they are at the minimal possible inter-cluster overlap $q_0/m = 0$).

- For submatrix averages $a > a_{\max} = 2$, we observe an UNSAT phase, i.e. no submatrix with that average of entries exists.

All the phases above have the characteristic of being described by solutions of (17) with Parisi parameter $x = 1$. In [3] we also verified that no 1-RSB solution with non-trivial Parisi parameter exists $0 < x < 1$ which is stable towards 2-RSB perturbation.

Inspired by the results above, we move to the tensor case $p > 2$ and look for solutions of (17) with Parisi parameter $x = 1$, conjecturing that no dominant solution with $0 < x < 1$ exists. We show in Appendix B that the phase diagram for the $p > 2$ tensor case is qualitatively identical to that of the matrix case, featuring an RS, and frozen 1-RSB, and an UNSAT phase as the subtensor average density $a$ increases. The corresponding phase transitions happen at the thresholds

$$a_{\text{freezing}}(p) = \frac{2}{\sqrt{p}} \quad \text{and} \quad a_{\max}(p) = 2 \,, \tag{25}$$

which reduce to the previous results when $p = 2$. Finally, the total entropy is given by

$$\frac{s(a)}{m \log 1/m} = 1 - \frac{a^2}{4} \,, \tag{26}$$

with zero complexity in the RS phase and complexity $\Sigma(a) = s(a)$ in the frozen 1-RSB phase.

## 2.4   The $p \to +\infty$ limit

When the tensor rank $p$ diverges, the properly rescaled energetic covariance matrix converges to

$$\frac{1}{mN} \mathbb{E}_J E(\sigma) E(\tau) \to \delta(\sigma = \tau) \tag{27}$$

as the diagonal covariances equal one by definition, and the out-diagonal covariances vanish as different configurations $\sigma, \tau$ of size $k$ necessarily have overlap $q(\sigma, \tau)/m < 1$. Thus, it is natural to conjecture that the model behavior will converge to that of a Random Energy Model (REM) [46], as it is the case for the Ising $p$-spin model [39]. Additionally, it was shown in [1] that for the $p = 2$ case the value of $a_{\max}$ coincides at leading order with the maximum of as many independent Gaussians as there are subtensors of size $k$, strengthening the parallel with the REM.

The $p \to +\infty$ limit of the $p$-MAS phase diagram gives

$$a_{\text{freezing}}(p) \to 0 \quad \text{and} \quad a_{\max}(p) \to 2 \,, \tag{28}$$

showing that the freezing transition matches that of the REM model (which is always in a frozen phase) as we expected.

In [3] we remarked that one could naively expect that the REM behavior should hold also for finite $p$ in the $m \to 0$ limit, and we were surprised that this is not the case (the freezing temperature being zero for the REM and non-zero for the finite $p$ and $m \to 0$ $p$-MAS problem). The only difference between the $m \to 0$ limit and the $p \to +\infty$ limit is indeed that in the former case the energy covariance matrix has sparse out-of-diagonal components, while in the latter case the same covariance matrix has uniformly vanishing out-of-diagonal components. We are still puzzled by this behavior, and still have not found an intuitive reason for which structured sparsity in the energy covariance may raise the freezing temperature.

## 2.5   The non-symmetric case

The non-symmetric case can be largely treated within the same formalism as the symmetric case. We report here the main differences and additional assumptions.

The first main difference is at the level of the replica description of the problem. Indeed, the $p$-spin model corresponding to the non-symmetric $p$-MAS problem features $p$ different "species" of spins $\sigma^{(g)} \in \{0,1\}^N$ with $g = 1, \ldots, p$, introducing thus $p$ magnetization $m^{(g)}$ and overlaps $q^{(g)}$. We provide the expression for the associated 1-RSB free entropy and state equations in Appendix C.

One can show that the state equations always admit what we call the *tensor-symmetric* solution, i.e. one where all species-wise magnetization have the same value, as well as the species-wise overlaps. The tensor-symmetric state equations have also the nice property of reducing exactly to the state equation of the symmetric $p$-MAS problem when rescaling the inverse temperature by a factor $\sqrt{p}$. Notice that under this ansatz all non-symmetric entropies are $p$ times their symmetric counterpart. Thus, the phase diagram for the non-symmetric case (under the tensor-symmetric ansatz) is the same as the symmetric one, with the only quantitative difference being a factor $\sqrt{(p-1)!}$ due to the different normalization of the energies (5) and (6), giving the thresholds

$$a_{\text{freezing}}(p) = \frac{2}{\sqrt{p!}}, \quad \text{and} \quad a_{\max}(p) = \frac{2}{\sqrt{(p-1)!}}. \tag{29}$$

In this paper, we assume that the tensor-symmetric solution is the correct one for $m \to 0$, and describe the resulting phase diagram. We leave the assessment of the validity of this ansatz for future work, remarking that in any case all our results on the symmetric case do not depend on this assumption.

## 3  Analytic prediction for the performance of greedy algorithms

We start the analysis of non-equilibrium properties of the $p$-MAS problem by computing analytically the algorithmic thresholds for the tensor problem of two greedy algorithms, and in particular for the algorithm that in the matrix case can enter the frozen 1-RSB phase. In [1, 2], two greedy strategies to find submatrices with large submatrix average have been proposed and analyzed in the thermodynamic limit:

- The Largest Average Submatrix (LAS) iterative algorithm, in which a large-submatrix-average submatrix is found by repeated optimization of the choice of rows and columns defining the submatrix.

- The Greedy Incremental Procedure (IGP), in which a large-submatrix-average submatrix is constructed by successively adding greedily the best row/column pair to a starting $1 \times 1$ submatrix (a single entry).

We anticipate that the LAS algorithm may be applied only in the non-symmetric case, while the IGP can be adapted to work in the symmetric case as well. Both algorithms extend naturally to the tensor case.

The performance of both the LAS and the IGP algorithms for the non-symmetric matrix case $p = 2$ have been characterized asymptotically in [1, 2]. In the following paragraphs we extend these results to the tensor case, and where possible to the symmetric one.

### 3.1  The IGP algorithm

The IGP algorithm is described for both the symmetric and the non-symmetric cases in the following pseudocode blocks. Here we represent subtensors by the set of non-deleted rows (i.e. the set of entries equal to

---

**Algorithm 1** Incremental Greedy Procedure ($\mathcal{IGP}$) - Symmetric case

---

**Input:** $N^{\otimes p}$ symmetric tensor $J$, size of the subtensor $k$
**Output:** $k^{\otimes p}$ principal symmetric subtensor
  $\mathcal{I} = \{i_1\}$                           $\triangleright$ Initialize with a random index $i_1$, forming a $1^{\otimes p}$ sub-tensor
  **while** $2 \le t \le k$ **do**
    $i_* = \arg\max_{i \notin \mathcal{I}} \text{avg}\, J_{\mathcal{I} \cup i}$                $\triangleright$ Find the slice increasing the average the most
    Set $\mathcal{I} = \mathcal{I} \cup \{i_*\}$                       $\triangleright$ Add it to the subtensor
    $t = t + 1$
  **end while**

---

one in the Boolean spin representation). The iteration starts from the subtensor including only the entry

$J_{i_1,i_1,\ldots,i_1}$ for an arbitrary index $i_1$ which can be chosen at random, and at each time step $t$ the current subtensor $\sigma^{t-1}$ gets enlarged into a $t^{\otimes p}$ symmetric tensor $\sigma^t$ by adding along each of the $p$ dimension the same row $i_*$, chosen among the $N - t$ available ones as the one that maximizes the subtensor average of $\sigma^t$. The algorithm requires exactly $k$ enlargement steps, each involving finding the maximum among $\mathcal{O}(N)$ real numbers, giving a total run-time of $\mathcal{O}(kN \log N)$ for $1 \ll k \ll N$[1].

---

**Algorithm 2** Incremental Greedy Procedure ($\mathcal{IGP}$) - Non-symmetric case

---

**Input:** $N^{\otimes p}$ tensor $J$, size of the subtensor $k$
**Output:** $k^{\otimes p}$ subtensor
  $\mathcal{I}^{(a)} = \{i^{(a)}\}$ for all $a$.          $\triangleright$ Initialize by selecting an arbitrary index along each dimension
  **while** $2 \le t \le k$ **do**          $\triangleright$ Enlarge greedily the subtensor by one unit along each dimension
    **while** $1 \le a \le p$ **do**          $\triangleright$ The search is repeated along every dimension
      $i_*^{(a)} = \arg\max_{i^{(a)} \notin \mathcal{I}^{(a)}} \operatorname{avg} J_{\mathcal{I}^{(1)},\ldots,\mathcal{I}^{(a)} \cup i^{(a)},\ldots,\mathcal{I}^{(p)}}$
      Set $\mathcal{I}^{(a)} = \mathcal{I}^{(a)} \cup \{i_*^{(a)}\}$
      $a = a + 1$
    **end while**
    $t = t + 1$
  **end while**

---

The non-symmetric algorithm is the same as the symmetric one, involving just one additional step. Indeed, at each time step $t$ we are free to choose the additional row $i_*^{(g)}$ along each dimension $g = 1, \ldots, p$ sequentially, as we have no symmetry constraint. The complexity of the algorithm is $\mathcal{O}(pkN \log N)$ for $1 \ll k \ll N$.

The IGP can be analyzed easily using the heuristic argument given in [2], which can be made rigorous. The idea is that at each step, the entries added to enlarge the tensor increase its subtensor concentrate to the maximum out of $\mathcal{O}(N)$ independent Gaussian random variables with variance $v_t$ which depends on the current size of the tensor $1 \le t \le k$. This maximum can be computed, and for $1 \ll k \ll N$ the cumulative effect of the enlargement, i.e. the final subtensor average, can be quantified precisely. This argument ignores correlations that build up at each step. This can be rigorously controlled for $1 \ll k \ll N$ by partitioning the tensor $J$ in $k^{\otimes p}$ sub-blocks, and picking each successive index inside a different block, effectively implementing the assumed independence while altering the subtensor average of the output only at sub-leading orders.

In Appendix D we present this argument for both the symmetric and the non-symmetric cases. We obtain the following thresholds

$$a_{\text{IGP}}^{\text{symm}}(p) = \frac{2p}{p+1} \frac{2}{\sqrt{p}} \quad \text{and} \quad a_{\text{IGP}}^{\text{non-symm}}(p) = \frac{1}{\sqrt{(p-1)!}} a_{\text{IGP}}^{\text{symm}}(p) \,, \tag{30}$$

where we also see that the additional factor in the non-symmetric case is the same rescaling that we used to convert the symmetric phase transition points into the non-symmetric phase transition point. Thus,

$$a_{\text{freezing}}(p) < a_{\text{IGP}}(p) < a_{\text{max}}(p) \tag{31}$$

for both the symmetric and non-symmetric cases and for all $p$, highlighting a region of subtensor averages $a \in [a_{\text{freezing}}(p), a_{\text{IGP}}(p)]$ in which equilibrium subtensors are isolated (a hint to algorithmic hardness, at least for equilibrium algorithms) but still an efficient algorithm exists that finds subtensors in that range of subtensor averages.

Notice that for $p \to \infty$ then $a_{\text{IGP}}^{\text{symm}}(p) \to 0$, as it should in the REM limit, where the lack of correlations make it impossible for an algorithm, even out of equilibrium, to reach non-trivial subtensor averages without doing exhaustive search.

## 3.2 The LAS algorithm

The LAS algorithm is described only for the non-symmetric case in the following pseudo-code block. In

---

[1] In providing these run-time estimates we are not being precise. We just want to stress that the algorithms at hand are efficient, i.e. polynomial.

---

**Algorithm 3** Large Average Submatrix ($\mathcal{LAS}$)

---

**Input:** $N^{\otimes p}$ tensor $J$, size of the subtensor $k$
**Output:** $k^{\otimes p}$ subtensor
  Initialize by selecting $k$ indices at random for each dimension.
  **while** convergence is not reached **do**
    **while** $1 \leq a \leq p$ **do**
      Set $\mathcal{I}^{(a)} = \arg\max_{\mathcal{J},|\mathcal{J}|=k} J_{\mathcal{I}^{(1)},...,\mathcal{J},...,\mathcal{I}^{(p)}}$
    **end while**
  **end while**

---

words, the LAS algorithm starts from a random subtensor of size $k$, and proceeds to optimize the choice of rows defining the subtensor dimension by dimension, repeating until convergence. The fixed points of the LAS algorithm are called *local maxima* for the $p$-MAS problem, and are subtensors whose subtensor average cannot be improved by changing its choice of rows along a single dimension (out of $p$). Each iteration of the LAS algorithm requires a polynomial number of operations, as for each dimension $p$ it involves computing $N$ sums over $k^{p-1}$ items, and then computing the $k$ largest sums, involving a total of $\mathcal{O}(pk^{p-1}N\log N)$ operations. In [2], the authors also prove that convergence is fast for the matrix case $p = 2$.

Notice that this algorithm cannot easily be extended to the symmetric case, as dimension-wise optimization breaks the symmetry, which is not restored easily. Skipping the dimension-wise optimization turns the algorithm into exhaustive search. For this reason, we consider the LAS algorithm only in the non-symmetric case.

In [1] the authors proved that all local maxima of the matrix problem $p = 2$ have the same submatrix average with high probability, corresponding to a

$$a_{\mathrm{LAS}}(p=2) = \sqrt{2}\,. \tag{32}$$

In [3] we highlighted that this threshold equals the freezing threshold $a_{\mathrm{freezing}}(p = 2)$, pointing to the fact that the LAS algorithm is subject to the conjectured hardness implied by the frozen phase. Compared with the IGP algorithm, we notice that the LAS is more akin to a local search algorithm like MCMC, while the IGP algorithm constructs directly solutions without doing any local exploration.

In Appendix E we argue that the same proof by [1] extends to the tensor case, modulo minor changes of factors, giving that all fixed points of the LAS algorithm for tensors converge to subtensor average

$$a_{\mathrm{LAS}}(p) = \frac{2}{\sqrt{p!}}\,, \tag{33}$$

which again coincides with the freezing threshold for all $p > 2$, and which goes to zero in the $p \to \infty$ REM limit as it should.

## 4   Overlap gap property

To try to understand better the algorithmic properties of the $p$-MAS problem, and in particular whether the IGP threshold is associated to any geometric property of the subtensor average landscape (such as clustering, etc...) and whether the IGP algorithm is the best algorithm or not, we now study non-equilibrium landscape properties of the associated spin model. We start by considering the Overlap Gap Property (OGP).

### 4.1   Motivation

In the simplest setting, we say that an optimization problem has an overlap gap property for a given energy density if the level set of configurations at that energy density contains no pair of configurations with overlap $q \in (\nu_1, \nu_2)$, but contains pairs of configurations with overlaps both in $0 < q < \nu_1$ and $q > \nu_2$. In other words, having an OGP means that the level set is strictly clustered. By strictly we mean that different clusters are truly disconnected, and not just that either connecting paths are exponentially rare or that connected

subdominant configuration sets may still exist, as is the case with the clustering implied by a frozen 1-RSB phase.

In several problems, it has been shown that if a problem has OGP for a given energy level set, then no sufficiently well-behaved[2] and efficient algorithm can output configurations with that energy density. The proof idea works by contradiction, showing that if a sufficiently well-behaved and efficient algorithm exists that outputs configurations with energy density $e$, then that level set cannot have OGP, as the algorithm can be used to produce configurations with the specified energy density and any arbitrary overlap. In more recent work, more refined statement about OGP were used to argue about hardness. In these generalizations, having an OGP means that a set of $r$ configurations in a energy level set can never have a certain overlap structure (and this reduces to the simple case above for $r = 2$). One can even allow for the $r$ replicas to have a given energy density w.r.t. $r$ different, but correlated, instances of the disorder. We refer the reader to [7] for more details.

It is then natural to ask whether there exists a specific OGP that appears at the IGP threshold in the $p$-MAS problem. If so, that OGP would provide us with a direction towards proving algorithmic hardness at larger subtensor values (answering the question of whether the IGP algorithm is optimal in any sense), and would also provide with a geometric intuition for why this hardness arises.

## 4.2   Previous work

In [2], the authors consider the non-symmetric matrix problem $p = 2$, and provide an upper bound for the threshold at which the simplest OGP ($r = 2$) arises. Translating their work into statistical mechanics terms, they consider the partition function

$$Z_J(a, q) = \sum_{\sigma_1, \sigma_2 : m(\sigma_1) = m(\sigma_2) = k} \delta(q(\sigma_1, \sigma_2) - q) \prod_{a=1,2} \delta(\mathrm{avg}_J(\sigma_a) - aT(N, m, p)), \tag{34}$$

counting the number of pairs of subtensors $\sigma_1, \sigma_2$ (of fixed size $1 \ll k \ll N$) at a fixed overlap $q$ and fixed intensive subtensor average $a$ (recall (13)). With the subscript $J$ we highlight that at his level no disorder average has been performed. They then compute the associated annealed entropy

$$s_{\mathrm{ann}}(a, q) = \frac{1}{N} \log \mathbb{E}_J Z_J(a, q), \tag{35}$$

as through Markov's inequality one has

$$\mathrm{Prob}_J \left( Z_J(a, q) \geq 1 \right) \leq e^{N s_{\mathrm{ann}}(a, q)}. \tag{36}$$

Thus, if $s_{\mathrm{ann}}(a, q) < 0$ in a range of values of $q$ for a given $a$, this inequality implies (but is not a necessary condition) that the problem satisfies the basic OGP $r = 2$ at submatrix average $a$. The authors show that $s_{\mathrm{ann}}(a, q) < 0$ for a range of values of $q$ whenever $a_{\mathrm{OGP}} < a < a_{\max}$, establishing an OGP property for the $p = 2$ non-symmetric problem. Notably, $a_{\mathrm{IGP}} < a_{\mathrm{OGP}}$ strictly, so that while the OGP bound hints at algorithmic hardness around $q_{\max}$, it does not provide more information about the IGP algorithm.

The authors suggest that considering more refined OGP properties (namely, multi-replica OGPs $r > 2$) may close the gap with the IGP algorithm. We explore this question, and others, in the next paragraphs.

Let us notice that proving OGP is just one step towards proving hardness, and problem-specific adaptations of the general technique described in [7] have to be devised. Here we do not address this part of the argument, but limit ourself to the analysis of landscape properties.

## 4.3   Annealed bounds in the symmetric and non-symmetric $p$-MAS problem

We start our analysis of OGP in the $p$-MAS problem by extending the OGP bound of [2] to the tensor case $p > 2$ and to multi-replica OGPs $r > 2$. As above, the non-symmetric case will reduce to the symmetric one under the tensor-symmetric ansatz, so we will focus on the symmetric case and comment on the non-symmetric one when necessary.

---

[2]In this manuscript we will not be precise on this point. What is usually needed is that the algorithm is either Lipschitz w.r.t. the disorder instance, or that it is an online algorithm [47], i.e. an algorithm that sets variables one by one without ever readjusting previously set variables. For e.g. the $\mathcal{IGP}$ algorithm is more akin to an online algorithm in this sense.

Let us define the partition function

$$Z_J^{(p,r)} = \sum_{\sigma_1,\dots,\sigma_r} \prod_{a<b} e^{\gamma q(\sigma_a,\sigma_b)} \prod_a e^{\beta E(\sigma_a) + N\beta h m(\sigma_a)}, \tag{37}$$

governing a system of $r$ replicas. Here $\beta$ is used to set all the energy densities of the replicas to a fixed value $e$, $h$ is used to set all the magnetization of the replicas to a fixed value $m$, and $\gamma$ is used to set all the overlaps between pairs of replicas to a fixed value $0 \leq q \leq m$. This is the canonical version of the OGP partition function (34), which can be though of as a micro-canonical partition function. We are interested in computing the annealed free entropy

$$\phi_{\text{ann}}^{(p,r)} = \frac{1}{N} \log \mathbb{E}_J Z_J^{(p,r)}(q), \tag{38}$$

which is an upper bound to the quenched free entropy. As in the usual equilibrium framework, we will then compute the associated entropy at fixed energy $e$, magnetization $m$ and overlap $q$ as

$$s_{\text{ann}}^{(p,r)} = \phi_{\text{ann}}^{(p,r)} - r\beta e - r\beta h m - \frac{r(r-1)}{2}\gamma q. \tag{39}$$

In Appendix F we present the details of the computations, which resemble closely 1-RSB equilibrium computations with Parisi parameter $x = r$ (modulo the annealed average, setting the number of replicas $n = 1$ instead of $n \to 0$ as usual). As for the equilibrium properties, we focus on the small subtensor limit $m \to 0$. In the limit, the annealed entropy for generic $p$ and $r$ reads

$$\frac{s_{\text{ann}}^{(p,r)}(l,a)}{m \log 1/m} = r - (r-1)l - \frac{r}{4}\frac{a^2}{1+(r-1)l^p}, \tag{40}$$

where we introduce the rescaled overlap parameter $l = q/m$ and the intensive subtensor average $a$ as (13). We notice that for $p = 2$ and $r = 2$ it reduces to the expression derived in [2, Eq. 4, with $y_1 = y_2 = l$, $\alpha = a/\sqrt{2}$] (notice that for $p = 2$ there is no rescaling between symmetric and non-symmetric quantities apart from a factor 2, so that our symmetric result is comparable with the non-symmetric result of [2] divided by 2).

Notice that

$$\frac{s_{\text{ann}}^{(p,r)}(l=0,a)}{m \log 1/m} = r\left(1 - \frac{a^2}{4}\right), \tag{41}$$

which is $r$ times the equilibrium entropy (as it should, as at equilibrium all configurations are at rescaled overlap $l = 0$). On the other hand

$$\frac{s_{\text{ann}}^{(p,r)}(l=1,a)}{m \log 1/m} = 1 - \frac{a^2}{4}, \tag{42}$$

which equals the equilibrium entropy as it should, as for $l = 1$ all $r$ replicas are the same, and the OGP computation reduces to the equilibrium one.

We also observe that all the annealed entropies develop an overlap gap for intensive subtensor averages $a_{\text{OGP}}^{\text{ann}}(p,r) < a < a_{\max}$, where $a_{\text{OGP}}^{\text{ann}}(p,r)$ is the largest value of $a$ such that

$$\forall a \leq a_{\text{OGP}}^{\text{ann}}(p,r) \quad \text{then} \quad s_{\text{ann}}^{(p,r)}(l,a) \geq 0 \quad \forall l \in [0,1], \tag{43}$$

and that for any fixed $p$

$$a_{\text{OGP}}^{\text{ann}}(p) = a_{\text{OGP}}^{\text{ann}}(p, r = p) \leq a_{\text{OGP}}(p, r \neq p), \tag{44}$$

implying that the $r = p$ OGP bound is the best among this class of bounds (contrary to what conjectured in [2]). In Figure 1 we plot $a_{\text{OGP}}^{\text{ann}}(p)$ and compare it with the equilibrium and algorithmic thresholds derived in the previous Sections, and provide an illustration of the observation that the $r = p$ annealed OGP entropy is the first to develop a gap as $a$ is increased.

As it was the case in the $p = 2$ setting, we have that $a_{\text{OGP}}^{\text{ann}}(p) > a_{\text{IGP}}(p)$ strictly also in the tensor case $p > 2$, implying that annealed OGP bounds are not sufficient to shed light on the behavior of the IGP algorithm across all values of $p$. We highlight two possible ways to improve upon this analysis. On the one hand, one could try to compute directly the quenched entropy (and not an annealed bound). It turns out that this is possible but computationally involved, and we discuss this in the following. On the other hand, one could explore more sophisticated OGPs along the lines of [26]. We leave this direction for future work.

As above, the non-symmetric case can be treated along the same lines as the symmetric case under the tensor-symmetric ansatz, modulo the usual $\sqrt{(p-1)!}$ rescaling in $a$ as well as a factor $p$ in the entropy.

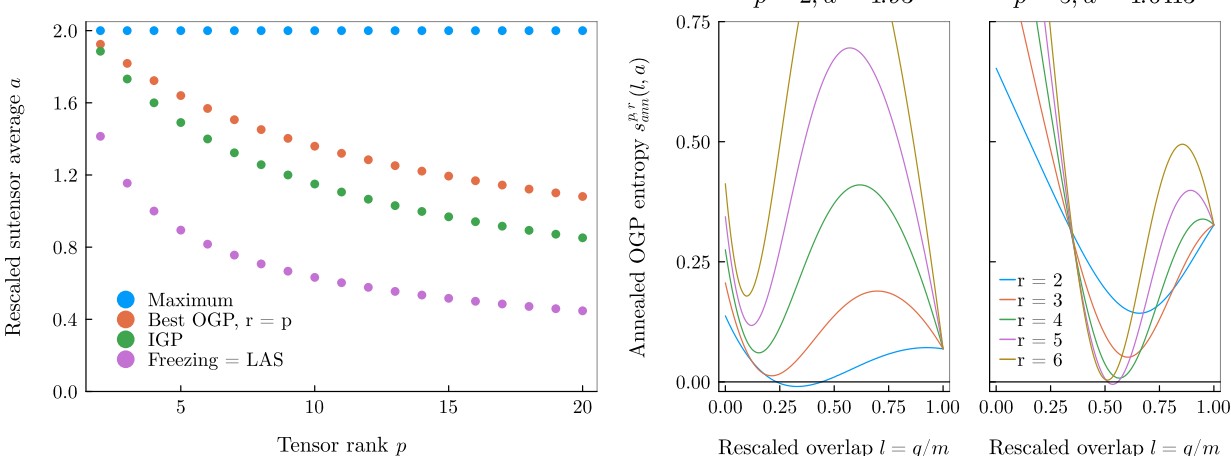

Figure 1: Annealed OGP entropies and associated algorithmic hardness OGP upper bounds in the limit $m \to 0$ for the symmetric $p$-MAS problem. (Left) Equilibrium phase transitions $a_{\max}(p)$ (maximum subtensor average) and $a_{\text{freezing}}(p)$ (freezing) compared with the performance of the greedy LAS and IGP algorithms and with $a_{\text{OGP}}^{\text{ann}}(p)$, i.e. the largest value of $a$ such that the annealed OGP entropy $s_{\text{ann}}^{(p,r=p)}(l,a) \geq 0$ for $l \in [0,1]$. (Right) Plots of the annealed OGP entropies $s_{\text{ann}}^{(p,r)}(l,a)$ for $p = 2, 5$ and $a = 1.93, 1.6413$ respectively, as a function of the rescaled overlap $l = q/m$ and for several values of the number of replicas $r = 2, \ldots, 6$. As we increase $a$, the $r = p$ entropy is the first to develop a gap, i.e. a region of $l$ where $s_{\text{ann}}^{(p,r)}(l,a) < 0$.

### 4.4   A comment on quenched OGP and local entropy

In the previous section we explored the annealed OGP bounds. It is natural to ask whether the quenched OGP entropy associated to the partition function (37) is within analytical reach. Indeed, if we were able to compute the quenched OGP entropy, we would obtain analytical access to sharp OGP thresholds, and not just to an annealed upper bound as computed in the previous section. Moreover, while annealed OGP entropies do not necessarily satisfy any monotonicity property w.r.t. the number of real replicas $r$ (and we saw in the previous section that the $r = 2$ annealed entropy was already providing the best bound for all values of $p$), quenched OGP entropies do. Indeed, one has that[3]

$$\frac{1}{r-1}s_{\text{quench}}^{(p,r-1)} \geq \frac{1}{r}s_{\text{quench}}^{(p,r)}, \tag{45}$$

where $s^{(p,r)}$ is the quenched OGP entropy

$$s_{\text{quench}}^{(p,r)} = \frac{1}{N}\mathbb{E}_J \log Z_J^{(p,r)}(q) - r\beta e - r\beta hm - \frac{r(r-1)}{2}\gamma q. \tag{46}$$

Thus, computing the quenched entropies at larger values of $r$ cannot provide worse OGP thresholds.

It is easy to see (see the very related discussion in [18]) that the quenched OGP entropy computation is equivalent to the 1-RSB equilibrium computation (17) (under a replica symmetric ansatz for the $r$ real replicas), the only differences being an additional field $\gamma$ associated to the intra-cluster overlap $q_1$, the fact that the equations are to be solved for $\gamma$ at fixed $q_1$, and that the Parisi parameter $x = r$ (counting the number of real replicas in the system). We also remark that the $r$-replicas quenched OGP entropy is a close relative of the so-called *local entropy*, at least in the formulation of [48]. Thus computing explicitly the

---

[3]Intuitively, consider the graph whose nodes are subtensors with a given subtensor average which are connected if they are at overlap exactly equal to $q$. Consider then triangles in this graph, i.e. triples of nodes all connected to each other. Then, the number of triangles is bounded by a certain function of the number of edges, as with a given number of edges one can build at most a certain number of triangles. (45) expresses this concept for higher-dimensional triangles at the level of quenched entropies. The same reasoning leads to a less useful monotonicity property for annealed entropies, involving higher moments of the original partition function.

quenched OGP would provide a explicit example, the $p$-MAS problem, in which annealed OGP, quenched OGP and local entropy can be compared between each other, and against analytic algorithmic thresholds, which is a challenging task in the SBP [21].

The asymptotic expansions associated to the quenched computation in the small $m$ limit are severely more involved than in the annealed case. We leave for future work the exploration of the quenched OGP phenomenology.

## 5   Franz-Parisi analysis

A second set of non-equilibrium insights comes for Franz-Parisi-style computations. The idea of Franz and Parisi [36] is to consider a reference equilibrium configuration, and to explore the statistical properties of non-equilibrium configurations (probes) constrained to be at fixed overlap from the reference. Within clustered phases, this allows to study a subset of the subdominant clusters populating the energy level sets.

In the context of constraint satisfaction problems and hardness, this tool has been used in [33, 34, 35] in the context of the binary perceptron model. We perform a similar analysis here, shedding even more light on the geometric properties of the subtensor average level sets, and discussing the apparent absence of a link with algorithmic properties.

In this Section, we focus exclusively on the matrix case $p = 2$ for simplicity. Given that most of the statistical properties of the $p$-MAS problem turned out to be qualitatively equivalent to the matrix case, we believe that this would be the case also for the following analysis.

### 5.1   The Franz-Parisi free entropy

We define the Franz-Parisi (FP) free entropy as

$$\Psi_{\mathrm{FP}} = \mathbb{E}_{J,\sigma_{\mathrm{ref}}} \frac{1}{N} \log \sum_{\sigma} e^{N\beta\beta_{\mathrm{ref}}\gamma q(\sigma_{\mathrm{ref}},\sigma) + \beta E(\sigma) + N\beta h m(\sigma)} \,, \tag{47}$$

where the average over $\sigma_{\mathrm{ref}}$ is performed according to the equilibrium Gibbs distribution (4) at parameters $\beta_{\mathrm{ref}}$ and $h_{\mathrm{ref}}$, fixing respectively the energy and the size of the reference subtensor $\sigma_{\mathrm{ref}}$. The parameters $\beta, m$ fix the energy and size of the probe subtensor $\sigma$, while $\gamma$ fixes the overlap between the reference and the probe subtensors. Notice that the disorder $J$ is the same for both the probe and the reference. Given the Franz-Parisi free entropy, one can compute the entropy of probe subtensors of a given size $k = m/N$ and intensive subtensor average $a$ that are at distance $q$ from an equilibrium configuration of size $k_{\mathrm{ref}} = m_{\mathrm{ref}}/N$ and intensive subtensor average $a_{\mathrm{ref}}$ as

$$s_{\mathrm{FP}} = \Psi_{\mathrm{FP}} - \beta e - \beta h m - \beta\beta_{\mathrm{ref}}\gamma q \,, \tag{48}$$

where $e$ is the energy density associated to $E(\sigma)$ (5) and linked to the subtensor average as given in (13).

We notice that while the FP free entropy is similar to the OGP partition function (37) and the related free entropy, it has an important difference. In the OGP computation, there is no asymmetry between the different subtensors involved. In the FP computation on the other hand, the probe and the reference configuration are clearly playing an asymmetric role, the reference acting as an additional quenched disorder for the probe configuration. We notice again a link with the local entropy formalism, as nicely discussed in [21].

Let us stress here that in the rest of the Section we will work in the small $m$ limit, and under the replica symmetric ansatz for the probe, and the dynamical 1-RSB ansatz for the reference configuration. The first ansatz should hold at least for large enough overlaps, while the second holds as a consequence of the analysis in [3].

### 5.2   Replica symmetric Franz-Parisi potential for small subtensors

In Appendix G, we derive the Franz-Parisi potential for the $p = 2$-MAS problem under a replica symmetric ansatz for the probe, and a dynamic 1-RSB ansatz for the reference. We then perform the same scaling

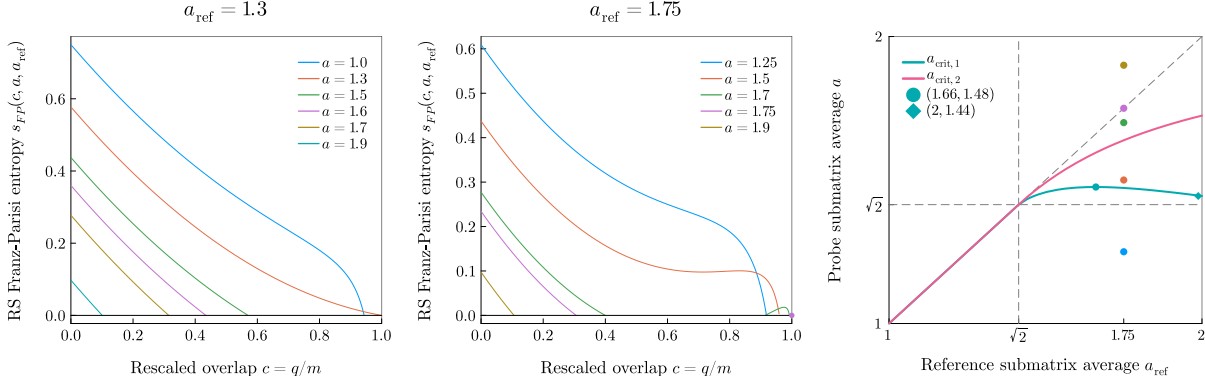

Figure 2: Behavior of the Franz-Parisi (FP) entropy as a function of the rescaled overlap $c = q/m$. (Left) We start considering a reference configuration in the RS phase, $a_{\text{ref}} \leq a_{\text{freezing}} = \sqrt{2}$. We see that in this case the FP entropy is monotonic for all choices of probe submatrix average $a$, and becomes negative at a rescaled overlap $c$ strictly smaller than one, unless $a = a_{\text{ref}}$. (Center) On the other hand, when the reference configuration is in the frozen 1-RSB phase, $a_{\text{ref}} > a_{\text{freezing}} = \sqrt{2}$, we see that the FP entropy becomes non-monotonic for certain values of the probe submatrix average $a$, and may develop a gap of negative entropy at intermediate values of the rescaled overlap $c$. (Right) Summary of the behavior of the FP entropy in the $(a_{\text{ref}}, a)$ plane. The colored dots are color coded with the curves of the central panel. For $a > a_{\text{ref}}$ and $a < a_{\text{crit},1}$ the FP entropy is monotone decreasing, and becomes negative at a value of the overlap $c < 1$. For $a = a_{\text{ref}}$ the behavior is the same, but the entropy is exactly zero at overlap $c = 1$. For $a_{\text{crit},1}(a_{\text{ref}}) < a < a_{\text{crit},2}(a_{\text{ref}})$, the FP entropy develops a local maximum close to $c = 1$, and has no negative entropy gap between $c = 0$ and the location of the local maximum. For $a_{\text{crit},2}(a_{\text{ref}}) < a < a_{\max} = 2$, the behavior is the same, but with an additional negative entropy gap between $c = 0$ and the location of the local maximum. We also highlight with a dot the coordinates of the local maximum of the curve $a_{\text{crit},1}(a_{\text{ref}})$, and with a diamond the coordinates of $a_{\text{crit},1}(2)$.

analysis for small-size submatrices $m \to 0$ as discussed in the previous sections, setting the size of both reference and probes to $k = mN$. We find that in the limit

$$\frac{s_{\text{FP}}(c, a, a_{\text{ref}})}{m \log \frac{1}{m}} = 1 - c - \frac{\left(a - a_{\text{ref}}c^2\right)^2}{4(1 - c^2)} \tag{49}$$

where we introduce the rescaled overlap parameter $c = q/m$ and the intensive subtensor averages $a, a_{\text{ref}}$ as defined in (13) for respectively the probe and reference configurations. In the rest of the Section we drop the $m \log 1/m$ scaling for simplicity.

Notice that at $c = 0$ (the global maximizer for any $a, a_{\text{ref}}$) the entropy equals the equilibrium total entropy

$$s_{\text{FP}}(c = 0; a, a_{\text{ref}}) = 1 - \frac{a^2}{4} \tag{50}$$

irrespectively of $a_{\text{ref}}$, confirming that at equilibrium the overlap $q_0 \ll m$ (both in the RS and frozen 1-RSB phases), i.e. equilibrium configurations are all at zero rescaled overlap between each other, while at $c = 1$ it equals

$$s_{\text{FP}}(c = 1; a, a_{\text{ref}}) = \begin{cases} 0 & a = a_{\text{ref}} \\ -\infty & a \neq a_{\text{ref}} \end{cases} \tag{51}$$

confirming the observation that at $c = 1$ probe and reference are the same microstate, so they must have the same energy (hence $s = -\infty$ if $a \neq a_{\text{ref}}$), and when they do the probe is stuck on a single configuration, thus zero entropy.

The form of the entropy is simple and allows for a detailed analysis. We discuss the main features below. See also Figure 2.

### 5.2.1 Probes with same submatrix average than the reference $a = a_{\text{ref}}$

We start by considering the case $a = a_{\text{ref}}$, i.e. probing the submatrix average level sets around equilibrium configurations. We have

$$s_{\text{FP}}(c, a = a_{\text{ref}}) = 1 - c - (1 - c^2)(a_{\text{ref}}/2)^2 \,. \tag{52}$$

One can easily check that for $0 < a_{\text{ref}} < a_{\text{freezing}} = \sqrt{2}$ the Franz-Parisi entropy is non-negative and monotone decreasing for all $0 \leq c \leq 1$, as one would expect in an RS phase. For $\sqrt{2} = a_{\text{freezing}} < a_{\text{ref}} < a_{\text{max}} = 2$ instead the Franz-Parisi entropy becomes negative for

$$c_{\text{zero}}(a = a_{\text{ref}}) = \left(\frac{2}{a_{\text{ref}}}\right)^2 - 1 < 1 \,, \tag{53}$$

while $s_{\text{FP}}(c = 1, a = a_{\text{ref}}) = 0$. This means that level sets are dominated by isolated point-like clusters, in accordance with the frozen 1-RSB prediction given in [3]. $c_{\text{zero}}(a = a_{\text{ref}})$ can be interpreted as the maximum overlap that any non-exponentially-rare configuration can achieve with an equilibrium cluster when they both have the same submatrix average.

### 5.2.2 Probes with larger submatrix average than the reference $a > a_{\text{ref}}$

In the case $a > a_{\text{ref}}$, we are looking at probes with larger submatrix average than the reference equilibrium configuration. In this case, the qualitative behavior of the Franz-Parisi entropy is simple, with $s_{\text{FP}}(c, a, a_{\text{ref}})$ being a decreasing function of $c$ that reaches zero at a value $c_{\text{zero}}(a, a_{\text{ref}}) < 1$, and then remains negative for $c > c_{\text{zero}}(a, a_{\text{ref}})$. This means that in the immediate surroundings of equilibrium configurations, there are no configurations with larger submatrix average. The closest configuration with larger submatrix average is at overlap $c_{\text{zero}}(a, a_{\text{ref}})$.

### 5.2.3 Probes with smaller submatrix average than the reference $a < a_{\text{ref}}$

We now consider the case $a < a_{\text{ref}}$, i.e. look at probes with smaller submatrix average than the reference equilibrium configuration. In this case, we observe a more varied phenomenology.

- There exists a first critical value of the submatrix average $a_{\text{crit},1}(a_{\text{ref}})$ such that the Franz-Parisi entropy for all $a_{\text{ref}}$ and for $a < a_{\text{crit},1}(a_{\text{ref}})$ is monotone decreasing, and becomes negative at a value $c_{\text{zero}}(a, a_{\text{ref}}) < 1$. This is the same behavior as in the $a > a_{\text{ref}}$ case. Notice that $a_{\text{crit},1}(a)$, satisfies $a_{\text{crit},1}(a) = a$ for $a \leq \sqrt{2}$ and $a_{\text{crit},1}(a) < a$ for $\sqrt{2} < a \leq 2$.

- There exists a second critical value of the submatrix average $a_{\text{crit},2}(a_{\text{ref}})$ such that the Franz-Parisi entropy for all $a_{\text{ref}}$ and for $a_{\text{crit},1}(a_{\text{ref}}) < a < a_{\text{crit},2}(a_{\text{ref}})$ is not monotonic anymore, featuring a single local maximum at a non-trivial value $0 < c_{\text{local}}(a, a_{\text{ref}}) < 1$, and becoming negative at $c_{\text{zero}}(a, a_{\text{ref}})$ such that $c_{\text{local}}(a, a_{\text{ref}}) < c_{\text{zero}}(a, a_{\text{ref}}) < 1$. Notice that $a_{\text{crit},2}(a)$, satisfies $a_{\text{crit},2}(a) = a$ for $a \leq \sqrt{2}$ and $a_{\text{crit},1}(a) < a_{\text{crit},2}(a) < a$ for $\sqrt{2} < a \leq 2$.

- Finally, for all $a_{\text{ref}}$ and for $a_{\text{crit},2}(a_{\text{ref}}) < a < a_{\text{max}}$, the qualitative behavior of the Franz-Parisi entropy is the same as in the previous case, with the only difference that the entropy is negative in a range of values between the global maximum at $c = 0$ and the local maximum at $c_{\text{local}}(a, a_{\text{ref}})$.

Figure 2 shows the qualitative behavior of $s$ for a set of representative values of $(a, a_{\text{ref}})$, as well as the curves $a_{\text{crit},1}$ and $a_{\text{crit},2}$. Notice that $a_{\text{crit},1}(a_{\text{ref}})$ is non-monotonic.

These results point towards the interpretation that for $a > a_{\text{crit},1}(a_{\text{ref}})$ equilibrium configurations are surrounded by clusters of configurations with smaller submatrix average, which are either strictly separated from other configurations, or a separated through entropic barriers. To provide a more precise interpretation of these different behaviors it is best to consider super-level sets of the submatrix average.

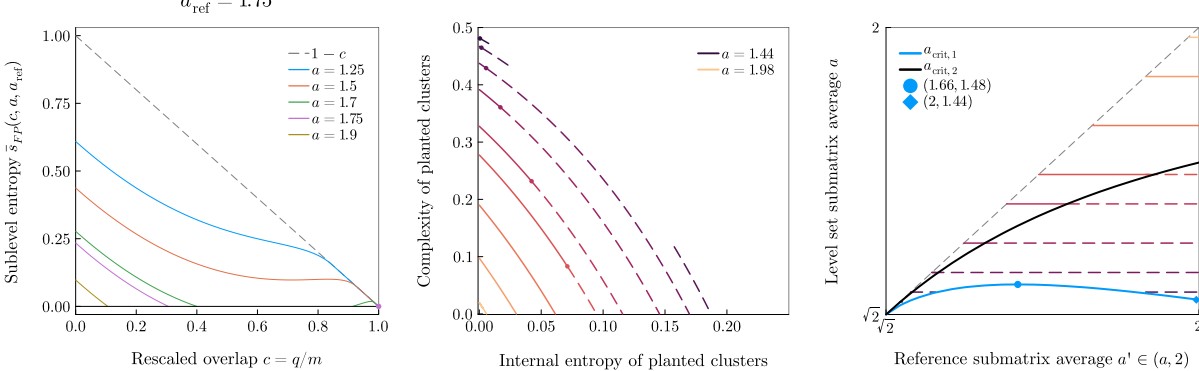

Figure 3: Structure of level and sub-level sets of the submatrix average. (Left) Sub-level FP entropy $\bar{s}_{\mathrm{FP}}$ for reference submatrix average $a_{\mathrm{ref}} = 1.75 > a_{\mathrm{freeze}}$ and several values of cutoff submatrix average $a_{\mathrm{cut}}$ as a function of the rescaled overlap $c = q/m$. We observe a similar phenomenology as for the standard FP entropy, see Figure 2 center. (Center) Complexity/internal entropy curves for planted clusters in submatrix average level sets $a = 1.44, 1.46, 1.5, 1.56, 1.64, 1.7, 1.8, 1.9, 1.98$. We see that in each energy level sets there are frozen clusters (the equilibrium configurations) as well as planted clusters (with non-zero internal entropy) surrounding equilibrium configurations at larger submatrix averages $a' \in (a, 2)$. Notice that the horizontal axis scale is quite stretched w.r.t. the vertical axis scale, so that in particular the slope of all the curves is well below $-1$ at all points. (Right) Reproduction of Figure 2 right, in the context of planted clusters. For each value of $a$ considered in the central panel, we plot here the values of $a' \in (a, 2)$ from which the curves in the central panel are constructed parametrically (same color means same value of $a$). Notice that level sets $1.44 \lessapprox a \lessapprox 1.48$ present planted clusters only of either very small or very large internal entropy, with a gap in between, due to the non-monotonicity of $a_{\mathrm{crit},1}$ (blue line). For $\sqrt{2} \approx 1.41 < a \lessapprox 1.44$ instead, only clusters with very small internal entropy are present.

### 5.2.4 Super-level sets analysis

Fix $a_{\mathrm{cut}} \in [0, 2]$, and consider the super-level set

$$\mathcal{S}(a_{\mathrm{cut}}) = \{\text{configurations } \sigma \text{ s.t. } a(\sigma) \geq a_{\mathrm{cut}}\} . \tag{54}$$

We would like to compute the Franz-Parisi entropy associated to probes $\sigma \in \mathcal{S}(a_{\mathrm{cut}})$, i.e. probes with submatrix average larger than $a_{\mathrm{cut}}$, while previously we considered probes with a fixed submatrix average. This can be derived by the usual Franz-Parisi entropy by defining

$$\bar{s}_{\mathrm{FP}}(c, a_{\mathrm{cut}}, a_{\mathrm{ref}}) = \max_{a \geq a_{\mathrm{cut}}} \left(1 - c - \frac{\left(a - a_{\mathrm{ref}} c^2\right)^2}{4(1 - c^2)}\right) = \begin{cases} s_{\mathrm{FP}}(c; a_{\mathrm{cut}}, a_{\mathrm{ref}}) & \text{if} \quad c^2 < a_{\mathrm{cut}}/a_{\mathrm{ref}} \\ 1 - c & \text{if} \quad a_{\mathrm{cut}}/a_{\mathrm{ref}} \leq c^2 \leq 1 \end{cases} , \tag{55}$$

where the second equality comes from solving the maximization in $a$ at fixed $c, a_{\mathrm{ref}}$, which is just a constrained maximization of a quadratic function. Figure 3 shows the qualitative behavior of $\bar{s}_{\mathrm{FP}}$ for a set of representative values of $(a_{\mathrm{cut}}, a_{\mathrm{ref}})$.

We notice immediately that for $a_{\mathrm{cut}} \geq a_{\mathrm{ref}}$, then no modification in the entropy occurs, $\bar{s}_{\mathrm{FP}}(c, a_{\mathrm{cut}}, a_{\mathrm{ref}}) = s_{\mathrm{FP}}(c, a_{\mathrm{cut}}, a_{\mathrm{ref}})$. The qualitative picture is as described above: equilibrium configuration have no configurations with larger or equal submatrix average around them.

Instead, when $a_{\mathrm{cut}} < a_{\mathrm{ref}}$ we have that $\bar{s}_{\mathrm{FP}}(c, a_{\mathrm{cut}}, a_{\mathrm{ref}}) \neq s_{\mathrm{FP}}(c, a_{\mathrm{cut}}, a_{\mathrm{ref}})$ for values of $c$ close to 1. In particular, there the two entropies differ we have

$$\bar{s}_{\mathrm{FP}}(c, a_{\mathrm{cut}}, a_{\mathrm{ref}}) = 1 - c , \tag{56}$$

which equals the $N \to \infty$ and $m \to 0$ limit of the rescaled entropy of configuration of size $k = mN$ of any submatrix average, constrained to have overlap $c = q/m$ with a given configuration. In other words, the super-level entropy saturates its natural upper-bound, meaning that at leading order all configurations at

overlap $c^2 \geq a_{\text{cut}}/a_{\text{ref}}$ from an equilibrium configuration of submatrix average $a_{\text{ref}}$ have submatrix average at least $a_{\text{cut}}$.

We also check numerically that the change in behavior for $c^2 \geq a_{\text{cut}}/a_{\text{ref}}$ always happens at values of the overlap larger than the position of the local maximum $c_{\text{local}}(a_{\text{cut}}, a_{\text{ref}})$, implying that the overall behavior of the super-level entropy $\bar{s}_{\text{FP}}$ is the same as that of the usual Franz-Parisi entropy. Thus, the overall phenomenology described for the Franz-Parisi entropy $s_{\text{FP}}$ holds for the super-level entropy $\bar{s}_{\text{FP}}$, with the only difference being the behavior close to $c = 1$, where the Franz-Parisi entropy $s_{\text{FP}}$ may be negative, but the super-level entropy $\bar{s}_{\text{FP}}$ is not.

Thus, we propose the following qualitative interpretation of the transitions $a_{\text{crit},1}(a_{\text{ref}})$ and $a_{\text{crit},2}(a_{\text{ref}})$.

- If $a_{\text{cut}} \leq a_{\text{crit},1}(a_{\text{ref}})$, then the super-level potential is monotone decreasing and positive for all values of $c$. No detectable cluster of configuration is surrounding the equilibrium configuration inside the super-level set $\mathcal{S}(a_{\text{cut}})$.

- If $a_{\text{crit},1}(a_{\text{ref}}) < a_{\text{cut}} \leq a_{\text{crit},2}(a_{\text{ref}})$, then the super-level potential is non-negative but non-monotonic, featuring a local maximum at $c_{\text{local}}(a_{\text{cut}}, a_{\text{ref}})$. Inside the super-level set $\mathcal{S}(a_{\text{cut}})$, equilibrium configurations are surrounded by a well-defined cluster of nearby configurations, separated from further configurations by entropic barriers. The size (in rescaled overlap terms) of these clusters is given by the value $c_{\text{local}}(a, a_{\text{ref}})$, and the internal entropy of these clusters is given by $s_{\text{FP}}(c_{\text{local}}(a_{\text{cut}}, a_{\text{ref}}); a_{\text{cut}}, a_{\text{ref}})$.

- If $a_{\text{crit},2}(a_{\text{ref}}) < a_{\text{cut}} \leq a_{\text{ref}}$, then the super-level potential is non-monotonic, featuring a local maximum at $c_{\text{local}}(a_{\text{cut}}, a_{\text{ref}})$, and is negative in a region strictly contained between $c = 0$ and $c = c_{\text{local}}(a_{\text{cut}}, a_{\text{ref}})$. Inside the super-level set $\mathcal{S}(a_{\text{cut}})$, equilibrium configurations are surrounded by a well-defined cluster of nearby configurations, strictly separated from further configurations, with size and internal entropy as in the previous point.

- If $a_{\text{ref}} < a_{\text{cut}} \leq a_{\text{max}}$, then the super-level sets contain no configuration surrounding the reference equilibrium configuration.

Thus, we see that the level sets of the $p = 2$-MAS problem feature a rich phenomenology for $a_{\text{ref}} \in [\sqrt{2}, 2]$ (i.e. in the frozen 1-RSB phase). Each submatrix average level set $a \in [\sqrt{2}, 2]$ contains isolated equilibrium configurations, but also the boundaries of "inflated" clusters surrounding equilibrium configurations at larger submatrix averages. We will call these clusters *planted* clusters, as they surround a planted equilibrium configuration.

We can further characterize such clusters by computing how many there are and how large they are in entropic terms. To do that, fix $a \in [\sqrt{2}, 2]$. Then, in the level set at $a$ one can find a complexity $\Sigma(a)$ of isolated configurations with zero internal entropy and with submatrix average $a$ (the equilibrium configurations at $a$), as well as a complexity of $\Sigma(a')$ of planted clusters (one per each equilibrium configuration at $a'$), each with internal entropy $s_{\text{int}} = s_{\text{FP}}(c_{\text{local}}(a, a'), a, a')$, for all $a < a' \leq a_{\text{max}} = 2$.

Figure 3 shows the associated curves $(s_{\text{FP}}(c_{\text{local}}(a, a'), a, a'), \Sigma(a'))$ plotted parametrically in $a' \in (a, 2)$ for several values of $a \in [\sqrt{2}, 2]$. In the plot, solid lines denote the region in which the planted clusters are strictly isolated, while dashed denotes the region in which the planted clusters are separated by entropic barriers. Notice that the level sets at $a$ slightly above $a_{\text{freezing}} = \sqrt{2}$ present a coexistence of many small clusters (leftmost part of the curves) and a few large clusters (rightmost part of the curves), with a gap in between, due to the non-monotonicity of the threshold $a_{\text{crit},1}(a')$. The gap develops for level sets $a$ such that $\sqrt{2} = a_{\text{freezing}} < a < \max a_{\text{crit},1} \approx 1.48$ (see Figure 2 right).

Notice that the level sets at $a \in [\sqrt{2}, 2]$ are dominated by the frozen clusters. Indeed, the dominant clusters are those with maximal sum of complexity and internal entropy $\Sigma + s_{\text{int}}$, which is always achieved in Figure 3 at $s_{\text{int}} = 0$, as the curves are concave and their slope is strictly lesser than $-1$, and decreasing as $s_{\text{int}}$ increases. This is compatible with the fact that frozen clusters were identified as the dominant contribution at equilibrium [3].

### 5.2.5 Lack of algorithmic insights

The analysis presented in this whole Section describes a very rich phenomenology for the level sets of the $p = 2$-MAS problem, in particular in the frozen 1-RSB phase. Indeed, we found that among the exponentially

rare configurations contributing to level sets, one can describe precisely a set of clusters that are planted around equilibrium configurations at larger submatrix average. Of course, this does not account for all the exponentially rare configurations contributing to level sets, missing all such configurations that are either subdominant in the Franz-Parisi analysis, or that at zero overlap with equilibrium configurations.

A natural question is now whether this analysis sheds some light on the algorithmic properties of this problem, and in particular on the behavior of the $\mathcal{IGP}$ algorithm entering the frozen phase. We find that the thresholds $a_{\text{crit},1}, a_{\text{crit},2}$, as well as the overall qualitative behavior of the planted clusters do not seem to correlate/behave specially at the $\mathcal{IGP}$ algorithmic thresholds $a_{\text{IGP}}$. We conclude that the $\mathcal{IGP}$ algorithm may be exploring some of the exponentially rare configurations that are missed both by the equilibrium configuration and by the Franz-Parisi analysis.

# 6    Acknowledgments

We thank Damien Barbier and David Gamarnik for insightful discussions. We acknowledge funding from the Swiss National Science Foundation grant TMPFP2_210012. R.P.O. was co-funded by the European Union (MSCA FutureData4Eu, Grant Agreement n. 101126733). Views and opinions expressed are however those of the authors only and do not necessarily reflect those of the European Union or REA. Neither the European Union nor the granting authority can be held responsible for them.

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

# Appendix

The code to reproduce the plots is available at https://github.com/SPOC-group/maximum-avg-subtensors.

## A Equilibrium 1-RSB computation for the symmetric $p$-MAS problem

In order to compute the averaged free entropy (see (4) and (14)), we first need to compute the averaged replicated partition function

$$
\begin{aligned}
\mathbb{E}_J[Z^n] &= \mathbb{E}_J\left[\mathrm{Tr}\, e^{\beta \sum_{\alpha=1}^n (E_J(\sigma^\alpha) + Nhm(\sigma^\alpha))}\right] \\
&= \int \prod_{i_1 < \cdots < i_p} DJ_{i_1,\ldots,i_p} \,\mathrm{Tr}\exp(\beta\sqrt{\frac{p!}{N^{p-1}}} \sum_{i_1 < \cdots < i_p} J_{i_1,\ldots,i_p} \sum_{\alpha=1}^n \sigma_{i_1}^\alpha \ldots \sigma_{i_p}^\alpha + \beta h \sum_{i=1}^N \sum_{\alpha=1}^n \sigma_i^\alpha),
\end{aligned}
\tag{57}
$$

where we introduce $n$ replicas labelled with the index $\alpha$. We integrate over each $J_{i_1,\ldots,i_p}$ independently explicitly

$$
\mathbb{E}_J[Z^n] = \mathrm{Tr}\exp(\frac{\beta^2 p!}{2N^{p-1}} \sum_{\alpha,\beta=1}^n \sum_{i_1 < \cdots < i_p} \sigma_{i_1}^\alpha \sigma_{i_1}^\beta \ldots \sigma_{i_p}^\alpha \sigma_{i_p}^\beta + \beta h \sum_{\alpha=1}^n \sum_{i=1}^N \sigma_i^\alpha).
\tag{58}
$$

We can split the sums over the replicas into diagonal and non-diagonal terms

$$
\sum_{\alpha,\beta=1}^n \sum_{i_1 < \cdots < i_p} \sigma_{i_1}^\alpha \sigma_{i_1}^\beta \ldots \sigma_{i_p}^\alpha \sigma_{i_p}^\beta = \left(\sum_{\alpha=\beta=1}^n + 2\sum_{\alpha<\beta}\right) \sum_{i_1 < \cdots < i_p} \sigma_{i_1}^\alpha \sigma_{i_1}^\beta \ldots \sigma_{i_p}^\alpha \sigma_{i_p}^\beta,
$$

with

$$
\begin{aligned}
\sum_{\alpha=\beta=1}^n \sum_{i_1 < \ldots < i_p} \sigma_{i_1}^\alpha \sigma_{i_1}^b \ldots \sigma_{i_p}^\alpha \sigma_{i_p}^\beta &= \sum_{\alpha=1}^n \sum_{i_1 < \ldots < i_p} (\sigma_{i_1}^\alpha)^2 \ldots (\sigma_{i_p}^\alpha)^2 = \sum_{\alpha=1}^n \sum_{i_1 < \ldots < i_p} \sigma_{i_1}^\alpha \ldots \sigma_{i_p}^\alpha \\
&\approx \frac{1}{p!} \sum_{\alpha=1}^n \sum_{i_1,\ldots,i_p} \sigma_{i_1}^\alpha \ldots \sigma_{i_p}^\alpha = \frac{1}{p!} \sum_{\alpha=1}^n \left(\sum_i \sigma_i^\alpha\right)^p,
\end{aligned}
$$

where we used that $\sigma_i^2 = \sigma_i$ for a boolean spin; and with

$$
\begin{aligned}
\sum_{\alpha<\beta} \sum_{i_1 < \ldots < i_p} \sigma_{i_1}^\alpha \sigma_{i_1}^\beta \ldots \sigma_{i_p}^\alpha \sigma_{i_p}^\beta &\approx \frac{1}{p!} \sum_{\alpha<\beta} \sum_{i_1,\ldots,i_p} \sigma_{i_1}^\alpha \sigma_{i_1}^\beta \ldots \sigma_{i_p}^\alpha \sigma_{i_p}^\beta \\
&= \frac{1}{p!} \sum_{\alpha<\beta} \left(\sum_i \sigma_i^\alpha \sigma_i^\beta\right)^p.
\end{aligned}
$$

We notice that the approximations become exact in the thermodynamic limit, as the diagonal terms become negligible.

After splitting the sums, the averaged partition function is

$$
\mathbb{E}_J[Z^n] = \mathrm{Tr}\exp(N\beta^2 \sum_{\alpha<\beta} \left(\frac{1}{N} \sum_i \sigma_i^\alpha \sigma_i^\beta\right)^p + \frac{N\beta^2}{2} \sum_\alpha \left(\frac{1}{N} \sum_i \sigma_i^\alpha\right)^p + \beta h \sum_\alpha \sum_i \sigma_i^\alpha).
\tag{59}
$$

We now introduce the variables

$$
q_{\alpha\beta} = \frac{1}{N} \sum_i \sigma_i^\alpha \sigma_i^\beta, \quad m_\alpha = \frac{1}{N} \sum_i \sigma_i^\alpha
\tag{60}
$$

and we enforce their definition with Fourier transform of Dirac deltas:

$$\mathbb{E}_J[Z^n] = \int \prod_{\alpha<\beta} dq_{\alpha\beta} \int \prod_\alpha dm_\alpha \exp(N\beta^2 \sum_{\alpha<\beta} q^p_{\alpha\beta} + \frac{N\beta^2}{2} \sum_\alpha m^p_\alpha)$$

$$\operatorname{Tr}\exp(\beta h \sum_\alpha \sum_i \sigma_i^\alpha) \prod_{\alpha<\beta} \delta(\sum_i \sigma_i^\alpha \sigma_i^\beta - Nq_{\alpha\beta}) \prod_\alpha \delta(\sum_i \sigma_i^\alpha - Nm_\alpha)$$

$$= \int \prod_{\alpha<\beta} dq_{\alpha\beta} d\hat{q}_{\alpha\beta} \prod_\alpha dm_\alpha d\hat{m}_\alpha \exp(N\beta^2 \sum_{\alpha<\beta} q^p_{\alpha\beta} + \frac{N\beta^2}{2} \sum_\alpha m^p_\alpha)$$

$$\operatorname{Tr}\exp(\beta h \sum_\alpha \sum_i \sigma_i^\alpha + \sum_{\alpha<\beta} \hat{q}_{\alpha\beta}(\sum_i \sigma_i^\alpha \sigma_i^\beta - Nq_{\alpha\beta}) + \sum_\alpha \hat{m}_\alpha(\sum_i \sigma_i^\alpha - Nm_\alpha)). \tag{61}$$

Spins have now become decoupled with respect to the site variable $i$, so we can factorize the trace:

$$\mathbb{E}[Z^n] = \int \prod_{\alpha<\beta} dq_{\alpha\beta} d\hat{q}_{\alpha\beta} \prod_a dm_\alpha d\hat{m}_\alpha \exp(N\left[\beta^2 \sum_{\alpha<\beta} q^p_{\alpha\beta} + \frac{\beta^2}{2} \sum_\alpha m^p_\alpha - \sum_{\alpha<\beta} \hat{q}_{\alpha\beta} q_{\alpha\beta} - \sum_\alpha \hat{m}_\alpha m_\alpha\right])$$

$$\left(\operatorname{Tr}\exp(\beta h \sum_\alpha \sigma^\alpha + \sum_{\alpha<\beta} \hat{q}_{\alpha\beta}\sigma^\alpha\sigma^\beta + \sum_\alpha \hat{m}_\alpha\sigma^\alpha)\right)^N$$

$$= \int \prod_{\alpha<\beta} dq_{\alpha\beta} d\hat{q}_{\alpha\beta} \prod_a dm_\alpha d\hat{m}_\alpha \exp\left\{N\left[\beta^2 \sum_{\alpha<\beta} q^p_{\alpha\beta} + \frac{\beta^2}{2} \sum_\alpha m^p_\alpha - \sum_{\alpha<\beta} \hat{q}_{\alpha\beta} q_{\alpha\beta} - \sum_\alpha \hat{m}_\alpha m_\alpha\right.\right. \tag{62}$$

$$\left.\left. + \log\left(\operatorname{Tr}\exp(\beta h \sum_\alpha \sigma^\alpha + \sum_{\alpha<\beta} \hat{q}_{\alpha\beta}\sigma^\alpha\sigma^\beta + \sum_\alpha \hat{m}_\alpha\sigma^\alpha)\right)\right]\right\}.$$

Under the 1-RSB ansatz, the terms in (62) are

$$\sum_{\alpha<\beta} q^p_{\alpha\beta} = \frac{1}{2}\left(n^2 q^p_0 + \frac{n}{x}x^2(q^p_1 - q^p_0) - nq^p_1\right)$$

$$\sum_{\alpha<\beta} q_{\alpha\beta}\hat{q}_{\alpha\beta} = \frac{1}{2}\left(n^2 q_0\hat{q}_0 + \frac{n}{x}x^2(q_1\hat{q}_1 - q_0\hat{q}_0) - nq_1\hat{q}_1\right)$$

$$\sum_{\alpha<\beta} \hat{q}_{\alpha\beta}\sigma^\alpha\sigma^\beta = \frac{1}{2}\left(\hat{q}_0\left(\sum_{\alpha=1}^n \sigma^\alpha\right)^2 + (\hat{q}_1 - \hat{q}_0)\sum_{\text{block}}^{n/x}\left(\sum_{\alpha\in\text{block}}^x \sigma^\alpha\right)^2 - \hat{q}_1\sum_{\alpha=1}^n \sigma^\alpha\right).$$

The magnetization $m$ is still assumed to be replica symmetric. In order to evaluate the log term, we need to introduce one Gaussian integral to deal with the sum $(\sum_{\alpha=1}^n \sigma^\alpha)^2$ and $n/x$ Gaussian integrals to deal with

the sums $\left(\sum_{a\in\text{block}}^{x}\sigma^{\alpha}\right)^{2}$:

$$\text{Tr}_{\sigma}\bullet\exp(\frac{\hat{q}_0}{2}\left(\sum_{\alpha=1}^{n}\sigma^{\alpha}\right)^{2}+(\hat{m}+\beta h-\frac{\hat{q}_1}{2})\sum_{\alpha=1}^{n}\sigma^{\alpha}+\frac{\hat{q}_1-\hat{q}_0}{2}\sum_{\text{block}}^{n/x}\left(\sum_{\alpha\in\text{block}}^{x}\sigma^{\alpha}\right)^{2})$$

$$=\text{Tr}_{\sigma}\bullet\int Du\exp(\left(\sqrt{\hat{q}_0}u+\hat{m}+\beta h-\frac{\hat{q}_1}{2}\right)\sum_{\alpha=1}^{n}\sigma^{\alpha})\prod_{\text{block}}^{n/x}\int Dv_{\text{block}}\exp(\sqrt{\hat{q}_1-\hat{q}_0}v_{\text{block}}\sum_{\alpha\in\text{block}}^{x}\sigma^{\alpha})$$

$$=\text{Tr}_{\sigma}\bullet\int Du\prod_{\text{block}}^{n/x}\int Dv_{\text{block}}\exp(\left(\hat{m}+\beta h-\frac{\hat{q}_1}{2}+\sqrt{\hat{q}_0}u+\sqrt{\hat{q}_1-\hat{q}_0}v_{\text{block}}\right)\sum_{\alpha\in\text{block}}^{x}\sigma^{\alpha})$$

$$=\int Du\prod_{\text{block}}^{n/x}\int Dv_{\text{block}}\left(\text{Tr}_{\sigma}\exp((\hat{m}+\beta h-\frac{\hat{q}_1}{2}+\sqrt{\hat{q}_0}u+\sqrt{\hat{q}_1-\hat{q}_0}v_{\text{block}})\sigma)\right)^{x}$$

$$=\int Du\left[\int Dv\left(1+\exp(\hat{m}+\beta h-\frac{\hat{q}_1}{2}+\sqrt{\hat{q}_0}u+\sqrt{\hat{q}_1-\hat{q}_0}v)\right)^{x}\right]^{n/x}.$$

We take the $n\to 0$ limit to obtain the 1-RSB free entropy density:

$$\phi_{1-RSB}=\text{Extr}_{q_0,\hat{q}_0,q_1,\hat{q}_1,m,\hat{m},x}\left\{\frac{\beta^2}{2}m^p+\frac{\beta^2}{2}x(q_1^p-q_0^p)-\frac{\beta^2}{2}q_1^p-\frac{x}{2}(q_1\hat{q}_1-q_0\hat{q}_0)+\frac{1}{2}q_1\hat{q}_1-m\hat{m}\right.$$

$$\left.+\frac{1}{x}\int Du\log\int Dy\left(1+\exp(\hat{m}+\beta h-\frac{\hat{q}_1}{2}+\sqrt{\hat{q}_0}u+\sqrt{\hat{q}_1-\hat{q}_0}v)\right)^{x}\right\} \tag{63}$$

The extremization of the 1-RSB free entropy with respect to $m$, $q_0$ and $q_1$ gives the relations:

$$\hat{m}=\frac{\beta^2}{2}pm^{p-1},\qquad \hat{q}_0=\beta^2 pq_0^{p-1},\qquad \hat{q}_1=\beta^2 pq_1^{p-1}.$$

Substituting the hat variables into (63) gives the expression of the 1-RSB free entropy density

$$\phi_{1-RSB}=(1-p)\frac{\beta^2}{2}[m^p+(x-1)q_1^p-xq_0^p]+\frac{1}{x}\int Du\log\int Dv\left(1+\exp(\beta H_{u,v})\right)^{x}, \tag{64}$$

where
$$H_{u,v}=h+\frac{\beta p}{2}(m^{p-1}-q_1^{p-1})+\sqrt{pq_0^{p-1}}u+\sqrt{p(q_1^{p-1}-q_0^{p-1})}v. \tag{65}$$

The saddle-point equations then follow similarly an in the usual $p$-spin model. The expression for the energy and magnetization, as well as for the complexity, then follow by taking the respective partial derivatives w.r.t. $\beta,m,x$.

# B The small magnetization limit of the equilibrium behavior of the $p$-MAS

## B.1 RS solution

We consider the RS equations (i.e. (17) with $q_0=q_1=q$):

$$m=\int Du\log(\beta H_u),$$

$$q=\int Dul^{2}(\beta H_u), \tag{66}$$

$$H_u=h+\frac{\beta p}{2}(m^{p-1}-q^{p-1})+\sqrt{pq^{p-1}}u$$

in the scaling limit:

$$\beta = b\sqrt{m^{1-p}\log\frac{1}{m}}$$

$$h = \eta\sqrt{m^{p-1}\log\frac{1}{m}}. \tag{67}$$

Plugging the scalings in the RS saddle-point equations gives

$$\beta H_u = b\left(\eta + \frac{pb}{2} - \frac{pb}{2}(\frac{q}{m})^{p-1}\right)\log\frac{1}{m} + b\sqrt{p(\frac{q}{m})^{p-1}}\sqrt{\log\frac{1}{m}}u. \tag{68}$$

As $m^2 \le q \le m$, we assume $q = cm^{1+\alpha}$ with $c > 0$ and $\alpha \ge 0$. For $\alpha > 0$, the second term goes to 0 and the dependence on $u$ drops in $H_u$, implying that

$$q = l^2(H) = m^2 \implies c = \alpha = 1, \tag{69}$$

consistently with our assumption for $q$. The equation for $m$ becomes

$$m = l(b(\eta + pb/2)\log\frac{1}{m}) \implies \log\frac{m}{1-m} = b(\eta + pb/2)\log\frac{1}{m}. \tag{70}$$

Solving for $\eta$:

$$\eta = -\frac{1}{b} - \frac{pb}{2}. \tag{71}$$

We can also compute the energy and entropy density:

$$e_{\mathrm{RS}} = \beta(m^p - q^p) \sim \beta m^p = b\sqrt{m^{p+1}\log\frac{1}{m}}$$

$$s_{\mathrm{RS}} = \phi_{\mathrm{RS}} - \beta e - \beta hm \sim (1-p)\frac{b^2}{2}m\log\frac{1}{m} - b^2 m\log\frac{1}{m} - b(-\frac{1}{b} - \frac{pb}{2})m\log\frac{1}{m} \tag{72}$$

$$= (1 - \frac{b^2}{2})m\log\frac{1}{m}.$$

We can recover the subtensor average from the energy:

$$\mathrm{avg} = \frac{\sqrt{p!N^{1-p}}}{m^p}e = \sqrt{(mN)^{1-p}\log\frac{1}{m}}\sqrt{p!}b$$

$$= b\sqrt{p!}\sqrt{k^{1-p}\log N}. \tag{73}$$

## B.2   1-RSB solution

Consider now (17) in the 1-RSB dynamic case, i.e. $x = 1$. We use the scaling

$$\beta = b\sqrt{m^{1-p}\log\frac{1}{m}}$$

$$h = \eta\sqrt{m^{p-1}\log\frac{1}{m}}$$

$$q_0 = m^2 \tag{74}$$

$$q_1 = c^2 m,$$

with $c = \mathcal{O}_m(1)$ to keep the dependence on $v$ in $H_{u,v}$. We have

$$\beta H_{u,v} = \left(b\eta + \frac{pb^2}{2}(1 - c^{2(p-1)})\right)\log\frac{1}{m} + b\sqrt{pm^{p-1}\log\frac{1}{m}}u + b\sqrt{p(c^{2(p-1)} - m^{p-1})\log\frac{1}{m}}v. \tag{75}$$

We see that the $u$ term is going to 0, and the dependence on $u$ drops, giving $q_0 = m^2$ and

$$H_v = \left(b\eta + \frac{pb^2}{2}(1 - c^{2(p-1)})\right)\log\frac{1}{m} + b\sqrt{pc^{2(p-1)}}\sqrt{\log\frac{1}{m}}v \equiv AN + B\sqrt{N}v, \tag{76}$$

where we defined $A = b\eta + \frac{pb^2}{2}(1 - c^{2(p-1)})$, $B = b\sqrt{p}c^{p-1}$ and $N = \log\frac{1}{m}$. We now want to solve the saddle-point equations in the region of positive complexity where $x = 1$. To do so, we need to compute integrals of the form

$$I_s = \int Dv(1 + e^{\beta H_v})l^s(\beta H_v). \tag{77}$$

We begin by computing $I_2$:

$$I_2 = \int dv \frac{e^{-v^2/2}}{\sqrt{2\pi}}(1 + e^{\beta H_v})l^2(\beta H_v) = \int dv \frac{e^{-v^2/2}}{\sqrt{2\pi}}(1 + e^{AN+B\sqrt{N}v})^{-1}e^{2(AN+B\sqrt{N}v)}$$

$$= \int \frac{dv}{\sqrt{2\pi}}e^{-v^2/2+2AN+2B\sqrt{N}v-\log 1\mathrm{pexp}(AN+B\sqrt{N}v)}, \tag{78}$$

where $\log 1\mathrm{pexp}(x) = \log(1 + \exp(x))$. We make the change of variable $v = \sqrt{N}w$:

$$I_2 = \sqrt{N}\int \frac{dw}{\sqrt{2\pi}}e^{N\left(-w^2/2+2A+2Bw-\frac{1}{N}\log 1\mathrm{pexp}(N(A+Bw))\right)} \equiv \sqrt{N}\int \frac{dw}{\sqrt{2\pi}}e^{N\mathcal{H}} \tag{79}$$

to put the integral in a form amenable to the saddle-point method. The logpexp term can be approximated by

$$\frac{1}{N}\log 1\mathrm{pexp}(N(A + Bw)) \approx \max(0, A + Bw) + \mathcal{O}(e^{-N\|A+Bw\|}) \tag{80}$$

when $\|A + Bw\|$ is sufficiently far from 0. We compute the derivative of the exponent:

$$\partial_w \mathcal{H}w = \begin{cases} -w + B & w > -\frac{A}{B} \\ -w + 2B & w < -\frac{A}{B} \end{cases} = 0, \tag{81}$$

giving two solutions:

$$\begin{cases} w_1^* = B & w > -\frac{A}{B} \\ w_2^* = 2B & w < -\frac{A}{B} \end{cases}. \tag{82}$$

We now distinguish different cases:

- if $2B > -\frac{A}{B}$ and $B > -\frac{A}{B}$, i.e. $-\frac{A}{B^2} < 1$, then $w^* = B$ and

$$I_2 = \sqrt{\frac{N}{2\pi}}e^{N(-B^2/2+2A+2B^2-(A+B^2))} = \sqrt{\frac{N}{2\pi}}e^{N(B^2/2+A)} \tag{83}$$

- if $B < -\frac{A}{B}$ and $2B < \frac{A}{B}$, i.e. $2 < -\frac{A}{B^2}$, then $w^* = 2B$ and

$$I_2 = \sqrt{\frac{N}{2\pi}}e^{N(-2B^2+2A+4B^2)} = \sqrt{\frac{N}{2\pi}}e^{N(2A+2B^2)} \tag{84}$$

- if $B > -\frac{A}{B}$ and $2B < -\frac{A}{B}$, we have $1 > 2$ which is impossible

- if $B < -\frac{A}{B}$ and $2B > -\frac{A}{B}$, i.e. $1 < -\frac{A}{B^2} < 2$, the derivative is never equal to 0 far from $w = -\frac{A}{B}$. We see that the derivative is $> 0$ for $w < -\frac{A}{B}$ and $< 0$ for $w > -\frac{A}{B}$, meaning we must look for the maximum at $-\frac{A}{B}$, which is where the approximation of the log1pexp breaks down.

We then compute $I_1$ and $I_0$:

$$I_1 = \int Dv(1 + e^{\beta H_v})l(\beta H_v) = \int Dv e^{\beta H_v}$$

$$= \sqrt{N} \int \frac{dw}{\sqrt{2\pi}} e^{N(-w^2/2+A+Bw)} = \sqrt{\frac{N}{2\pi}} e^{N(B^2/2+A)}, \tag{85}$$

$$I_0 = \int Dy(1 + e^{\beta H_v}) = 1 + I_1.$$

The 1-RSB equations in the small $m$ limit are:

$$m = \frac{I_1}{I_0}$$

$$q_1 = c^2 m = \frac{I_2}{I_0}. \tag{86}$$

If $B^2/2 + A > 0$, $I_1 \approx I_0$ which is not compatible with $m \to 0$, further constraining $-\frac{A}{B^2} > \frac{1}{2}$. We are then left with the following cases :

- $\frac{1}{2} < -\frac{A}{B^2} < 1$: the equation for $q_1$ gives

$$I_1 = I_2 \implies c^2 = 1 \implies A = b\eta, \quad B = b\sqrt{p} \tag{87}$$

and the equation for $m$ gives

$$m = \frac{I_1}{I_0} = \frac{I_1}{1 + I_1} = \frac{\sqrt{N/2\pi}e^{N(\frac{B^2}{2}+A)}}{1 + \sqrt{N/2\pi}e^{N(\frac{B^2}{2}+A)}} = \frac{e^{N(\frac{B^2}{2}+A)}}{\sqrt{2\pi/N} + e^{N(\frac{B^2}{2}+A)}}. \tag{88}$$

Taking the log on both sides we get

$$\log m = N(\frac{B^2}{2} + A) - \underbrace{\log(\sqrt{2\pi/N} + e^{N(\frac{B^2}{2}+A)})}_{\sim \log N} \approx N(\frac{B^2}{2} + A). \tag{89}$$

Remembering that $N = \log\frac{1}{m}$, we have $\frac{B^2}{2} + A = -1$, giving us

$$\eta = -\frac{1}{b} - \frac{pb}{2}. \tag{90}$$

- $2 < -\frac{A}{B^2}$: $\implies A + 2B^2 < 0$, therefore we have

$$c^2 = \frac{I_2}{I_1} = e^{N(A+\frac{3}{2}B^2)} \to 0. \tag{91}$$

The dependence on $v$ drops in Eq. (76) and we get $q_1 = m^2 = q_0$, recovering the RS solution.

- $1 < -\frac{A}{B^2} < 2$: following [3] we also recover the RS solution.

To sum up, for $-\frac{A}{B^2} < \frac{1}{2}$ there is no solution, for $\frac{1}{2} < -\frac{A}{B^2} < 1$ we find a 1-RSB solution different from the RS one while for $-\frac{A}{B^2} > 1$ we recover the RS solution. Recalling that $A = b\eta = -1 - \frac{pb^2}{2}$ and $B = b\sqrt{p}$ for $c = 1$, the condition for the existence of the 1-RSB solution is:

$$\frac{1}{2} < -\frac{A}{B^2} < 1 \implies \frac{1}{2} < \frac{1 + pb^2/2}{pb^2} = \frac{1}{pb^2} + \frac{1}{2} < 1 \implies b > \sqrt{2/p}. \tag{92}$$

In the region with positive complexity, we fix $x = 1$ and the 1-RSB energy is equal to the RS one. To find the complexity (18), we first compute:

$$-x^2 \partial_x \phi_{1-\text{RSB}}(x) = -x^2(1-p)\frac{b^2}{2}(q_1^p - q_0^p) + \overbrace{\int Du \log \int Dv(1 + \exp(\beta H))^x}^{D}$$
$$- x \underbrace{\int Du \frac{\int Dv \log(1 + e^{\beta H})(1 + e^{\beta H})^x}{\int Dv(1 + e^{\beta H})}}_{E} \tag{93}$$

In the small $m$ limit, the dependence on $u$ drops, and setting $x = 1$ we have

$$D = \log(I_0) = \log(1 + I_1) \approx I_1 \sim m$$

$$E = \int Dv \log(1 + e^{\beta H})(1 + e^{\beta H}) \approx \sqrt{\frac{\log \frac{1}{m}}{2\pi}} e^{\log \frac{1}{m}(B^2/2 + A)}(A \log \frac{1}{m} + B^2 \log \frac{1}{m}) \tag{94}$$

$$\approx (A + B^2) m \log \frac{1}{m} = (-1 + \frac{pb^2}{2}) m \log \frac{1}{m},$$

where we used that $A + B^2/2 = -1$. Collecting the terms, we are left with

$$\Sigma = -x^2 \partial_x \phi_{1-\text{RSB}}(x)|_{x=1} = (1 - \frac{b^2}{2}) m \log \frac{1}{m}. \tag{95}$$

## C  The non-symmetric case

We start from (6). Then $(a, b$ are replica indices)

$$\mathbb{E}_J Z^n = \text{Tr} \prod_{i_1 \dots i_p} \int DJ \exp \left( J \beta N^{\frac{1-p}{2}} \sum_a \prod_{\alpha=1}^p \sigma_{i_\alpha}^{(\alpha),a} \right)$$

$$= \text{Tr} \prod_{i_1 \dots i_p} \exp \left( \frac{1}{2} \beta^2 N^{1-p} \sum_{a,b} \prod_{\alpha=1}^p \sigma_{i_\alpha}^{(\alpha),a} \sigma_{i_\alpha}^{(\alpha),b} \right)$$

$$= \text{Tr} \exp \left( \frac{1}{2} \beta^2 N^{1-p} \sum_{a,b} \prod_{\alpha=1}^p \sum_{i_\alpha} \sigma_{i_\alpha}^{(\alpha),a} \sigma_{i_\alpha}^{(\alpha),b} \right)$$

$$= \text{Tr} \exp \left( \frac{1}{2} \beta^2 N \sum_{a,b} \prod_{\alpha=1}^p q(\sigma_{i_\alpha}^{(\alpha),a}, \sigma_{i_\alpha}^{(\alpha),b}) \right) \tag{96}$$

$$= \int dm dq \exp \left( \beta^2 N \sum_{a<b} \prod_{\alpha=1}^p q_\alpha^{ab} + \frac{1}{2} \beta^2 N \sum_a \prod_{\alpha=1}^p m_\alpha^a \right)$$

$$\times \text{Tr} \, \delta \left( N q_\alpha^{ab} - \sum_i \sigma_i^{(\alpha),a} \sigma_i^{(\alpha),b} \right) \delta \left( N m_\alpha^a - \sum_i \sigma_i^{(\alpha),a} \right)$$

Now assume the "tensor symmetric" solution $q_\alpha^{ab} = q^{ab}$ and $m_\alpha^a = m^a$, then

$$\mathbb{E}_J Z^n = \int dm dq \exp \left( \beta^2 N \sum_{a<b} q_{ab}^p + \frac{1}{2} \beta^2 N \sum_a m_a^p \right)$$

$$\times \left[ \text{Tr} \, \delta \left( N q^{ab} - \sum_i \sigma_i^a \sigma_i^b \right) \delta \left( N m^a - \sum_i \sigma_i^a \right) \right]^p \tag{97}$$

Then we see that the trace term will contribute a term proportional to $p$ to the free entropy, as it is a product over $p$ of identical terms, while the first term has not such factor of $p$. We also see that by setting $\beta = \sqrt{p}\beta_s$ this free entropy is just $p$ times the free entropy of the symmetric model. Thus the saddle-point equations are the same as those for the symmetric model at any level of RSB.

# D  Asymptotics of the IGP algorithm

We first recall the following two facts:

- the maximum of $N$ Gaussian random variables $X_i \sim \mathcal{N}(0, \sigma^2)$ is asymptotically $\sqrt{2\sigma^2 \log N}$,

- the sum of $n$ i.i.d. standard Gaussian random variables is also Gaussian, with $\sigma^2 = n$.

We begin by recalling the argument laid out in [2] in the non-symmetric matrix case to fix the notation and ideas, before generalizing to arbitrary $p$ and adapting the argument to the symmetric case.

## D.1  Non-symmetric case

### D.1.1  $p = 2$ case

We denote by $\mathcal{I}$ the set of row indices belonging to the submatrix and by $\mathcal{J}$ the set of column indices belonging to the submatrix. In the initial step of the algorithm, we pick a random index $i_1$ such that $\mathcal{I} = \{i_1\}$ and $\mathcal{J} = \varnothing$. In the row indexed by $i_1$, denoted $J_{i_1,[N]}$, we find the largest entry $J_{i_1,j_1}$, adding its column index to $\mathcal{J}$ to form an initial $1 \times 1$ submatrix. It is the maximum out of $N$ i.i.d. standard Gaussians and is asymptotically $\sqrt{2 \log N}$. We then find the largest entry $J_{i_2,j_1}$ in the column $J_{[N] \setminus \mathcal{I}, j_1}$, which is the column of $J_{i_1,j_1}$ with $J_{i_1,j_1}$ removed. It is also $\sqrt{2 \log N}$ asymptotically as the maximum of $N-1$ standard Gaussians (in general, the maximum out of $N - l$ Gaussians with $l \ll N$ is also $\sqrt{2 \log N}$). Adding its row index to our submatrix, $\mathcal{I} = \{i_1, i_2\}$, we now have a $2 \times 1$ submatrix. To form a $2 \times 2$ submatrix, we find the index $j_2$ such that sum of the two entries in the $2 \times 1$ column $J_{\mathcal{I}, j_2}$ is maximal, among the pairs in the $2 \times (N - 1)$ submatrix $J_{\mathcal{I}, [N] \setminus \mathcal{J}}$, which is the submatrix formed of the two rows indexed by $\mathcal{I}$, with the two entries already in our submatrix removed. The sum of two standard Gaussians is also a Gaussian with a variance of 2, so the maximum of the sum of two entries among $N - 1$ pairs is asymptotically $\sqrt{2 \cdot 2 \log N}$.

Let us take stock of the steps taken so far:

1. we start by finding the largest entry in an arbitrary row, with an asymptotic value of $\sqrt{2 \log N}$, to form a $1 \times 1$ submatrix,

2. in the corresponding column, we find the next largest entry, also with an asymptotic value of $\sqrt{2 \log N}$, forming a $2 \times 1$ submatrix,

3. in the two rows corresponding to the two entries we selected so far, we find the pair of entries with the second largest sum, asymptotically $\sqrt{4 \log N}$, and form a $2 \times 2$ submatrix.

We then repeat steps 2 and 3, increasing each time the size of the submatrix from $r \times r$ to $(r+1) \times (r+1)$ and contributing $\sqrt{2(r) \log N} + \sqrt{2(r+1) \log N}$ to the total of the final submatrix. An illustration of two of those steps is shown in Figure 4. Once the submatrix is of size $k \times k$, the sum of all its entries is

$$T = \sqrt{2 \log N} + \sum_{r=1}^{k-1} (\sqrt{2r \log N} + \sqrt{2(r+1) \log N}) \approx 2 \int_1^k \sqrt{x}\, dx \sqrt{2 \log N} \approx \frac{4}{3} k^{3/2} \sqrt{2 \log N}. \qquad (98)$$

Diving by the number of entries in the submatrix, $k^2$, we find the average

$$A_{IGP,p=2}^{non-sym} = \frac{4}{3}\sqrt{2k^{-1} \log N}. \qquad (99)$$

We assumed that the maximizations over Gaussian variables were independent even though they are not, but the algorithm can be slightly adjusted with no change in the asymptotic values, as shown in [2].

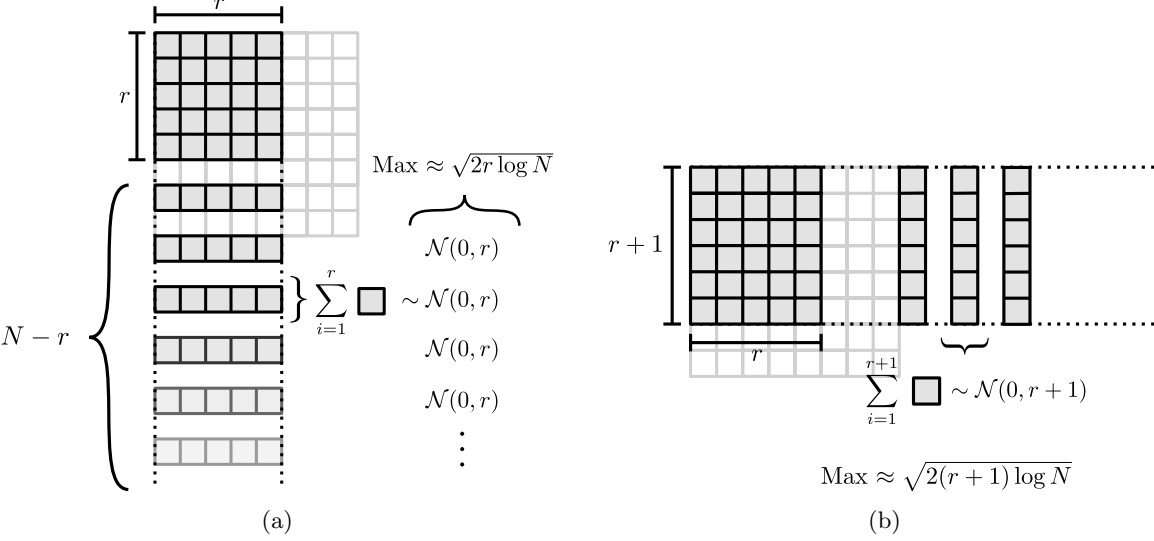

(a)                                                           (b)

Figure 4: Illustration of the two steps taken by the original $p = 2$ IGP algorithm to increase the size of the subtensor from $r$ to $r + 1$. For the purpose of the illustration, we draw a submatrix with contiguous rows and columns. In general, this need not be the case, but we can relabel the rows/columns to make it so.

### D.1.2   Generic $p$

In the tensorial case, the difference is that $p$ steps are needed to increase the size of the subtensor from $r^{\otimes p}$ to $(r + 1)^{\otimes p}$. In the first of those $p$ steps, we find a $(p - 1)-$dimensional slice along the first dimension such that the sum of its entries is larger than the sum of other possible slices. This slice has $r^{p-1}$ entries, and therefore contributes $\sqrt{2r^{p-1}\log N}$ to the total of the subtensor. Having added this $r^{\otimes(p-1)}$ slice to the subtensor, it is now of size $(r + 1) \times r^{\otimes(p-1)}$. We then repeat this search along each dimension, adding the highest sum slice to the subtensor, with a contribution of $\sqrt{2(r + 1)^{d-1}r^{p-d}\log N}$, where $d$ is the index of dimension along which we're currently enlarging the subtensor. We illustrate the first two steps needed to increase a 3-subtensor from $4 \times 4 \times 4$ to $5 \times 5 \times 5$ in Figure 5. The total contribution of the $p$ steps needed to increase the size of the subtensor from $r^{\otimes p}$ to $(r + 1)^{\otimes p}$ is therefore

$$T_r = \sum_{d=1}^{p} \sqrt{(r+1)^{d-1}r^{p-d}}\sqrt{2\log N}. \tag{100}$$

The final total is

$$T = \sum_{r=1}^{k-1} T_r = \sum_{r=1}^{k-1}\sum_{d=1}^{p} \sqrt{(r+1)^{d-1}r^{p-d}}\sqrt{2\log N}. \tag{101}$$

We can once again approximate the sum over $r$ with integrals:

$$\sum_{r=1}^{k-1}\sum_{d=1}^{p} \sqrt{(r+1)^{d-1}r^{p-d}} \approx \sum_{d=1}^{p} \int_{2}^{k} x^{\frac{d-1}{2}}(x+1)^{\frac{p-d}{2}}dx. \tag{102}$$

In the large $k$ limit, we have

$$\int_{2}^{k} x^{\frac{d-1}{2}}(x+1)^{\frac{p-d}{2}}dx \approx \frac{2}{p+1}k^{\frac{p+1}{2}}. \tag{103}$$

We see that the dependence on $d$ drops, leaving us with

$$T = \frac{2p}{p+1}k^{\frac{p+1}{2}}\sqrt{2\log N}. \tag{104}$$

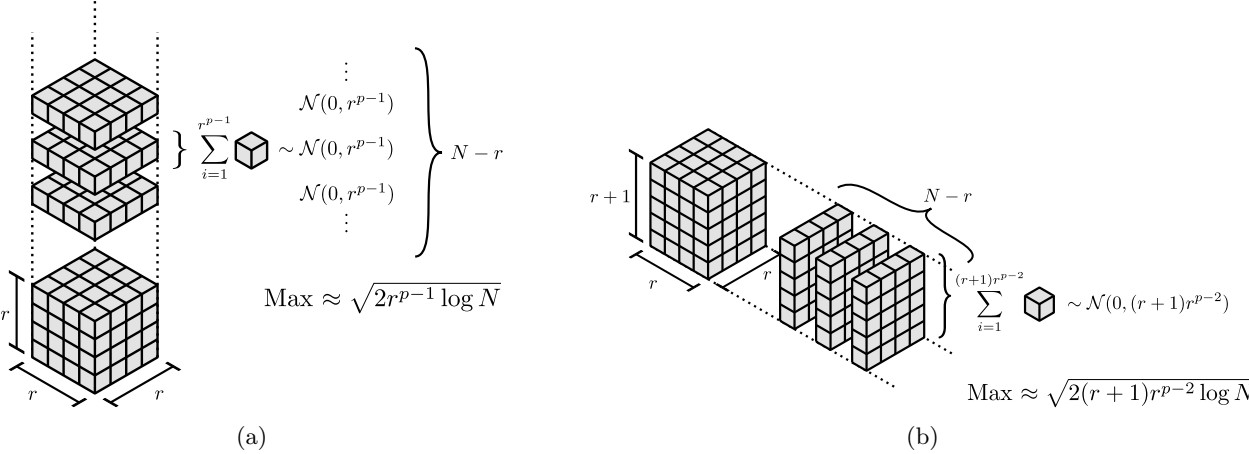

(a)                                                                        (b)

Figure 5: Illustration of the first two steps needed to increase the size of the subtensor from $r$ to $r+1$ for the non-symmetric IGP algorithm. Here, $r = 4$ and $p = 3$.

We divide by $k^p$ to find the average

$$A_{IGP,p}^{non-sym} = \frac{2p}{p+1}\sqrt{2k^{1-p}\log N}. \tag{105}$$

## D.2   Symmetric case

In the symmetric case, a single step increases the size of the subtensor by one along every dimension. We choose an arbitrary dimension along which we will search for a maximal slice (as the tensor is symmetric we can fix this dimension to be the first one). Recall that we are looking at principal subtensors, so we denote our subtensor with a single subset of indices $\mathcal{I}$ and we are looking for a maximal slice among the slices $J_{i,\mathcal{I},...,\mathcal{I}}$. Unlike the non-symmetric case were all the entries of a slice were i.i.d, due to symmetry a slice only has $\binom{r}{p-1}$ independent entries, ignoring the diagonal entries (independent entries can be labelled by a strictly increasing sequence of integers $1 \le i_1 < \cdots < i_{p-1} \le r$, and there are $\binom{r}{p-1}$ such sequences). We are then finding the maximum among $N - r$ Gaussians of variance $\binom{r}{p-1}$. Each of those entries appears $(p-1)!$ times in the slice, and the maximal slice will therefore contribute $(p-1)!\sqrt{2\binom{r}{p-1}\log N}$ to the total. Finally, adding this slice along the first dimension will automatically add the same slice along all other dimensions, giving a total contribution for each step of

$$p \cdot (p-1)!\sqrt{2\binom{r}{p-1}\log N}. \tag{106}$$

A step will also add other entries that we have not taken into account, as illustrated in Figure 6, but they are all diagonal and therefore negligible for $k \gg 1$. Summing the contributions of all the steps, we get

$$T = p!\sum_{r=r_0}^{k-1}\sqrt{2\binom{r}{p-1}\log N}, \tag{107}$$

where we neglect the steps before $p \ll r_0 \ll k$ as their contribution is negligible and to ensure that we can safely ignore the diagonal entries. In the large $k$ limit

$$\sum_{r=r_0}^{(k-1)}\sqrt{\binom{r}{p-1}} = \sum_{r=r_0}^{k-1}\sqrt{\frac{r(r-1)\ldots(r-(p-2))}{(p-1)!}} \approx \frac{1}{\sqrt{(p-1)!}}\int_{r_0}^{k}\sqrt{x^{p-1}}dx. \tag{108}$$

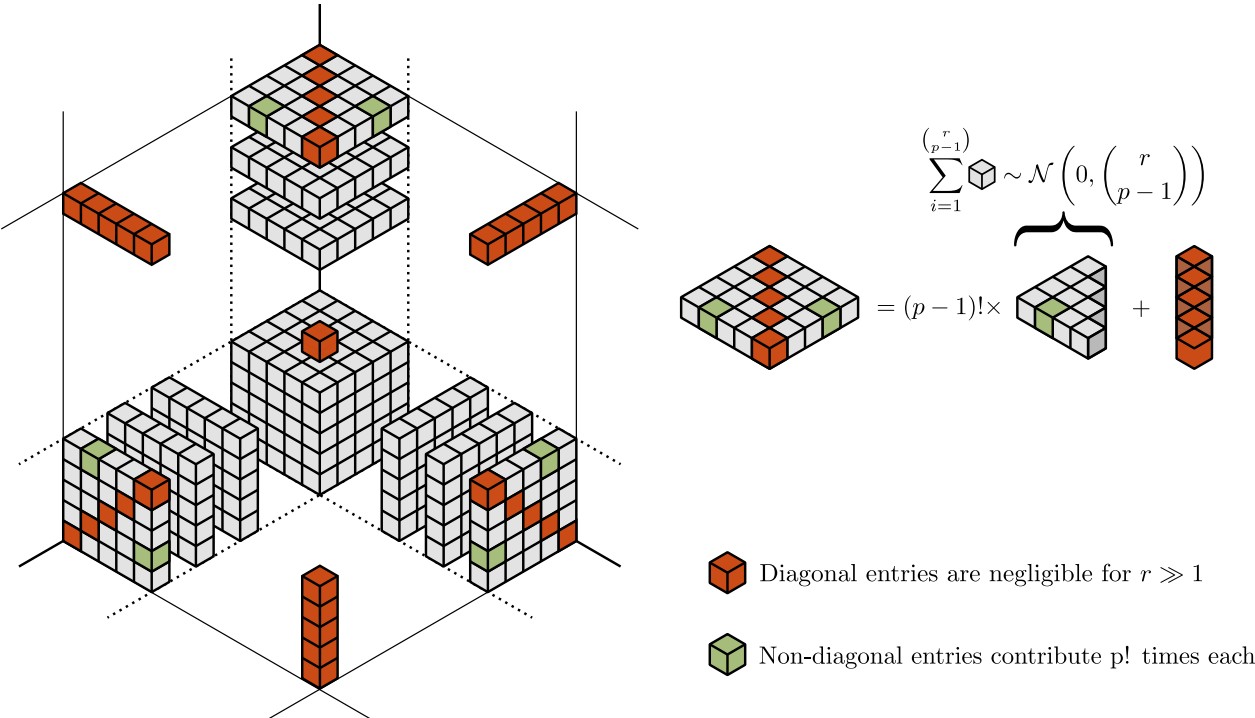

Figure 6: Due to symmetry, a single step of the symmetric $\mathcal{IGP}$ increases the size of the tensor by one unit. We show here the entries added by such a step, and in particular in orange the diagonal terms, which can be ignored for $r$ large enough, and we illustrate the contribution of a single entry, in green, from the selected slice. This entry appears $(p-1)!$ times in the slice, given that it is $(p-1)$-dimensional, and the slice appears $p$ times, once along each dimension.

The total is

$$T = \frac{p!}{\sqrt{(p-1)!}} \frac{2}{p+1} k^{\frac{p+1}{2}} \sqrt{2 \log N} \tag{109}$$

and dividing by $k^p$ we have

$$A_{IGP}^{sym} = \frac{2p}{p+1} \sqrt{2(p-1)! k^{1-p} \log N}. \tag{110}$$

# E  Asymptotics of the LAS algorithm

The structure theorem (theorem 2.5 in [1]) implies that all entries in a typical locally optimal submatrix are concentrated around $\sqrt{2k^{-1} \log N}$, which coincides with the freezing threshold. We sketch here the adjustments required to extend the proof to subtensors of arbitrary dimension.

The definitions of row and column dominant are generalized by considering a subtensor to be slice dominant if its average cannot be improved by swapping one of its index along a dimension to another not yet included. We can then define the events $\mathcal{R}_k^{(\alpha)}$ as the subtensor being slice optimal along dimension $\alpha$, and we are interested in the event $\mathcal{I}_{k,n} = \cap_{\alpha=1}^p \mathcal{R}_k^{(\alpha)}$ corresponding to locally optimal subtensors.

As the ANOVA decomposition plays an important role in the original proof, we introduce the tensor-ANOVA decomposition of a subtensor $\mathbf{C}$ as:

$$C_{i_1,\ldots,i_p} = \tilde{c}_{i_1,\ldots,i_p} + (c_{\bullet,\ldots,\bullet,i_p} - c_{\bullet,\ldots,\bullet}) + (c_{\bullet,\ldots,\bullet,i_{p-1},\bullet} - c_{\bullet,\ldots,\bullet}) + \ldots + (c_{i_1,\bullet,\ldots,\bullet} - c_{\bullet,\ldots,\bullet}) + c_{\bullet,\ldots,\bullet}, \tag{111}$$

where $c_{\bullet,\ldots,\bullet}$ is the average of the entries of $\mathbf{C}$ and $c_{\bullet,\ldots,i_\alpha,\ldots,\bullet} = k^{-(p-1)} \sum_{i_1,\ldots,i_{\alpha-1},i_{\alpha+1},\ldots,i_p=1}^k W_{i_1,\ldots,i_p}$ is the average of the first $k \times^{(p-1)}$ entries in the $i_\alpha$th slice along dimension $\alpha$. One can check that this decomposition

has the same independence properties as the matrix one. The rest of the proof follows the matrix case closely, leading to the same conclusion as for submatrices, that is the average of locally optimal subtensors is identic to the freezing threshold.

# F    Annealed OGP computation for the $p$-MAS problem

We start from the partition function

$$Z_{p,r} = \sum_{\sigma^1,\dots,\sigma^r} \prod_{\alpha<\beta} e^{\gamma q(\sigma_\alpha,\sigma_\beta)} \prod_\alpha e^{\beta E(\sigma_\alpha)+N\beta h m(\sigma_\alpha)}, \tag{112}$$

with $r$ real replicas. We compute its average

$$\mathbb{E}_J[Z_{p,r}]$$

$$= \mathbb{E}_J\left[\mathrm{Tr}\exp(\beta\sum_\alpha\sqrt{\frac{p!}{N^{p-1}}}\sum_{i_1<\dots<i_p}J_{i_1,\dots,i_p}\sigma_{i_1}^\alpha\dots\sigma_{i_p}^\alpha + N\beta h\sum_\alpha m(\sigma_\alpha) + N\gamma\sum_{\alpha<\beta}q(\sigma_\alpha,\sigma_\beta))\right]$$

$$= \mathrm{Tr}\exp(\frac{p!}{2N^{p-1}}\beta^2\sum_{\alpha,\beta}\sum_{i_1<\dots<i_p}\sigma_{i_1}^\alpha\sigma_{i_1}^\beta\dots\sigma_{i_p}^\alpha\sigma_{i_p}^\beta + N\beta h\sum_\alpha m(\sigma^\alpha) + N\gamma\sum_{\alpha<\beta}q(\sigma^\alpha,\sigma^\beta)) \tag{113}$$

$$= \mathrm{Tr}\exp(N\beta^2\sum_{\alpha<\beta}q(\sigma^\alpha,\sigma^\beta)^p + \frac{N}{2}\beta^2\sum_\alpha m(\sigma^\alpha)^p + N\beta h\sum_\alpha m(\sigma^\alpha) + N\gamma\sum_{\alpha<\beta}q(\sigma^\alpha,\sigma^\beta)).$$

To take the trace, we define $q_{\alpha\beta} = q(\sigma^\alpha,\sigma^\beta)$ and $m_\alpha = m(\sigma^\alpha)$ and enforce the equalities with Diracs, that we write in Fourier:

$$\mathbb{E}_J[Z_{p,r}]$$

$$= \exp(N\beta^2\sum_{\alpha<\beta}q_{\alpha\beta}^p + \frac{N}{2}\beta^2\sum_\alpha m_\alpha^p)$$

$$\times \mathrm{Tr}\int\prod_\alpha dm_\alpha d\hat{m}_\alpha\prod_{\alpha<\beta}dq_{\alpha\beta}d\hat{q}_{\alpha\beta}\exp(\sum_\alpha\hat{m}_\alpha\left(\sum_i\sigma_i^\alpha - Nm_\alpha\right) + \sum_{\alpha<\beta}\hat{q}_{\alpha\beta}\left(\sum_i\sigma_i^\alpha\sigma_i^\beta - Nq_{\alpha\beta}\right))$$

$$\times \exp(N\beta h\sum_\alpha m(\sigma^\alpha) + N\gamma\sum_{\alpha<\beta}q(\sigma^\alpha,\sigma^\beta)) \tag{114}$$

$$= \int\prod_\alpha dm_\alpha d\hat{m}_\alpha\prod_{\alpha<\beta}dq_{\alpha\beta}d\hat{q}_{\alpha\beta}\exp(N\beta^2\sum_{\alpha<\beta}q_{\alpha\beta}^p + \frac{N}{2}\beta^2\sum_\alpha m_\alpha^p - N\sum_\alpha\hat{m}_\alpha m_\alpha - N\sum_{\alpha<\beta}\hat{q}_{\alpha\beta}q_{\alpha\beta})$$

$$\times \mathrm{Tr}\exp(\sum_\alpha(\hat{m}_\alpha + \beta h)\sum_i\sigma_i^\alpha + \sum_{\alpha<\beta}(\hat{q}_{\alpha\beta} + \gamma)\sum_i\sigma_i^\alpha\sigma_i^\beta).$$

We can factorize the trace over the site index $i$ to obtain

$$\mathbb{E}_J[Z_{p,r}] = \int\prod_\alpha dm_\alpha d\hat{m}_\alpha\prod_{\alpha<\beta}dq_{\alpha\beta}d\hat{q}_{\alpha\beta}\exp\left(N\left[\beta^2\sum_{\alpha<\beta}q_{\alpha\beta}^p + \frac{\beta^2}{2}\sum_\alpha m_\alpha^p\right.\right.$$

$$\left.\left. - \sum_\alpha\hat{m}_\alpha m_\alpha - \sum_{\alpha<\beta}\hat{q}_{\alpha\beta}q_{\alpha\beta} + \log\mathrm{Tr}_{\sigma^\bullet}\exp L\right]\right), \tag{115}$$

where

$$L = \sum_\alpha(\hat{m}_\alpha + \beta h)\sigma^\alpha + \sum_{\alpha<\beta}(\hat{q}_{\alpha\beta} + \gamma)\sigma^\alpha\sigma^\beta. \tag{116}$$

The saddle-point equations are

$$m_\alpha = \frac{\mathrm{Tr}\,\sigma^\alpha e^L}{\mathrm{Tr}\,e^L} = \langle\sigma^\alpha\rangle_L, \quad q_{\alpha\beta} = \langle\sigma^\alpha\sigma^\beta\rangle_L$$

$$\hat{m}_\alpha = p\frac{\beta^2}{2}m_\alpha^{p-1}, \quad \hat{q}_{\alpha\beta} = p\beta^2 q_{\alpha\beta}^{p-1} \tag{117}$$

and we can eliminate the hat variables to get

$$\mathbb{E}_J[Z_{p,r}] = \int \prod_\alpha dm_\alpha \prod_{\alpha<\beta} dq_{\alpha\beta} \exp N \left( (1-p)\beta^2 \sum_{\alpha<\beta} q_{\alpha\beta}^p + (1-p)\frac{\beta^2}{2} \sum_\alpha m_\alpha^p + \log \operatorname{Tr} \exp L \right). \qquad (118)$$

We consider the real replica symmetric ansatz, i.e. $m_\alpha = m$ and $q_{\alpha\beta} = q$ for all the real replicas:

$$\mathbb{E}_J[Z_{p,r}] = \int dmdq \exp(N\left[ r(1-p)\frac{\beta^2}{2}m^p + \frac{r(r-1)}{2}(1-p)\beta^2 q^p + \log \operatorname{Tr} \exp L \right]), \qquad (119)$$

with

$$L = (p\frac{\beta^2}{2}m^{p-1} + \beta h) \sum_\alpha \sigma^\alpha + (p\beta^2 q^{p-1} + \gamma) \sum_{\alpha<\beta} \sigma^\alpha \sigma^\beta. \qquad (120)$$

We can factorize with respect to the replica index:

$$\mathbb{E}_J[Z_{p,r}] = \int dmdq \exp N \left[ r(1-p)\frac{\beta^2}{2}m^p + \frac{r(r-1)}{2}(1-p)\beta^2 q^p + \log(\int Dz(1+e^{H_z})^r) \right], \qquad (121)$$

with

$$H_z = \beta h - \frac{\gamma}{2} + p\frac{\beta^2}{2}(m^{p-1} - q^{p-1}) + \sqrt{p\beta^2 q^{p-1} + \gamma} z = A + Bz. \qquad (122)$$

The saddle-point equations are

$$m = \frac{\int Dz(1+\exp(H_z))^r l(H_z)}{\int D_z(1+\exp(H_z))^r}$$

$$q = \frac{\int Dz(1+\exp(H_z))^r l^2(H_z)}{\int D_z(1+\exp(H_z))^r}. \qquad (123)$$

Noticing that

$$(1+\exp(H_z))^r = \sum_{k=0}^r \binom{r}{k} \exp(kH_z), \qquad (124)$$

we can write

$$\int Dz(1+\exp(H_z))^r = \sum_{k=0}^r \binom{r}{k} \exp(kA + \frac{1}{2}k^2B^2)$$

$$\int Dz(1+\exp(H_z))^{r-1} \exp(H_z) = \sum_{k=1}^r \binom{r-1}{k-1} \exp(kA + \frac{1}{2}k^2B^2) \qquad (125)$$

$$\int Dz(1+\exp(H_z))^{r-2} \exp(H_z) = \sum_{k=2}^p \binom{r-2}{k-2} \exp(kA + \frac{1}{2}k^2B^2).$$

We use the scalings:

$$\beta = b\sqrt{m^{1-p} \log \frac{1}{m}}$$

$$h = \eta\sqrt{m^{p-1} \log \frac{1}{m}} \qquad (126)$$

$$\gamma = g \log \frac{1}{m} + 2g_0$$

where we add a subleading term in $\gamma$ to ensure that $l \in (0,1)$, having parametrized $q = lm$. We have

$$A = \left[ b\eta - \frac{g}{2} + p\frac{b^2}{2}(1-l^{p-1}) \right] \log \frac{1}{m} = -\mathcal{A} \log \frac{1}{m}$$

$$B^2 = \left[ pb^2 l^{p-1} + g \right] \log \frac{1}{m} + 2g_0 = -2\mathcal{B} \log \frac{1}{m} + 2g_0 \qquad (127)$$

and the saddle-point equations are

$$m = \frac{\sum_{k=1}^{r} \binom{r-1}{k-1} m^{f(k)} e^{g_0 k^2}}{\sum_{k=0}^{r} \binom{r}{k} m^{f(k)}}$$

$$q = \frac{\sum_{k=2}^{r} \binom{r-2}{k-2} m^{f(k)} e^{g_0 k^2}}{\sum_{k=0}^{r} \binom{r}{k} m^{f(k)}},$$

(128)

with $f(k) = \mathcal{A}k + \mathcal{B}k^2$. As $f(0) = 0$, all other terms need to be of order $m$ or $o(m)$, therefore $f(k) \geq 1$ for all $k \geq 1$ and the denominator equals 1 in the limit. The SP equations reduce to

$$m = \sum_{k=1}^{r} \binom{r-1}{k-1} m^{f(k)} e^{g_0 k^2}$$

$$q = \sum_{k=2}^{r} \binom{r-2}{k-2} m^{f(k)} e^{g_0 k^2},$$

(129)

and each sum will be dominated by the term with the smallest $f(k)$. We first consider the case where $\mathcal{B} < 0$, as in this case $f(k)$ is concave down and is minimized on the border of the domain, i.e. $k = 1, 2$, $k = r$, or both, depending on conditions on $\mathcal{A}$ and $\mathcal{B}$ that we detail now.

For the $m$ equation:

- $f(1) < f(r)$ if $\mathcal{A} > -(1+r)\mathcal{B}$, then $m^{f(1)}$ dominates,

- $f(1) = f(r)$ if $\mathcal{A} = -(1+r)\mathcal{B}$, then $m^{f(1)}$ and $m^{f(r)}$ both dominate,

- $f(1) > f(r)$ if $\mathcal{A} < -(1+r)\mathcal{B}$, then $m^{f(r)}$ dominates.

For the $q$ equation:

- $f(2) < f(r)$ if $\mathcal{A} > -(2+r)\mathcal{B}$, then $m^{f(2)}$ dominates,

- $f(2) = f(r)$ if $\mathcal{A} = -(2+r)\mathcal{B}$, then $m^{f(2)}$ and $m^{f(r)}$ both dominate,

- $f(2) > f(r)$ if $\mathcal{A} < -(2+r)\mathcal{B}$, then $m^{f(r)}$ dominates.

Combining the cases and taking into account that $-(2+r)\mathcal{B} > -(1+r)\mathcal{B}$ as $\mathcal{B} < 0$, we have:

- $\mathcal{A} < -(1+r)\mathcal{B}$, $m^{f(r)}$ dominates in both equations, implying that $l = 1$. We discard this case as we want $l \in (0,1)$.

- $\mathcal{A} = -(1+r)\mathcal{B}$, the $m$ equation is dominated by $m^{f(1)}$ and $m^{f(r)}$ and the $q$ equation by $m^{f(r)}$. Imposing that $f(1) = f(p) = 1$ implies $\mathcal{A} = 1 + 1/p$ and $\mathcal{B} = -1/p$ which is consistent with the condition. We also have $1 = e^{g_0} + e^{p^2 g_0}$ and $l = e^{p^2 g_0}$, leaving $l$ free to take any value in $(0,1)$ by tuning the subleading parameters.

- $\mathcal{A} > -(1+r)\mathcal{B}$, the $m$ equation imposes that only $f(1) = 1$ while the $q$ equation implies that another value of $k$ dominates, which is a contradiction.

Keeping the only valid option, we have $\mathcal{A} = 1 + 1/p$ and $\mathcal{B} = -1/p$, giving

$$b\eta - \frac{g}{2} + p\frac{b^2}{2}(1 - l^{p-1}) = -1 - 1/p, \quad pb^2 l^{p-1} + g = 2/p,$$

(130)

that is

$$b\eta = -1 - p\frac{b^2}{2}, \quad g = \frac{2}{r} - pb^2 l^{p-1}.$$

(131)

The free entropy density is obtained from (119):

$$\phi_{\text{ann}}^{(p,r)} = (1-p)r\frac{\beta^2}{2}[m^p + (r-1)q^p] + \log \text{Tr} \exp L,$$

(132)

where the log term is subleading in the small magnetization limit and can be dropped. We can obtain the entropy via a Legendre transform:

$$s_{\mathrm{ann}}^{(p,r)} = \phi_{\mathrm{ann}}^{(p,r)} - r\beta e - r\beta hm - \frac{r(r-1)}{2}\gamma q, \tag{133}$$

where the energy is $e = \beta(m^p + (r-1)q^p)$. In the limit, we have

$$
\begin{aligned}
\frac{s_{\mathrm{ann}}^{(p,r)}}{m\log\frac{1}{m}} &= (1-p)r\frac{b^2}{2}(1+(r-1)l^p) \\
&\quad - rb^2(1+(r-1)l^p) + r(1+p\frac{b^2}{2}) - \frac{r(r-1)}{2}(\frac{2}{r} - pb^2l^{p-1})l \\
&= r - (r-1)l - r\frac{b^2}{2}(1+(r-1)l^p).
\end{aligned}
\tag{134}
$$

The rescaled average can be obtained from the expression for the energy and is

$$a = \sqrt{2}b(1+(r-1)l^p). \tag{135}$$

We can therefore rewrite the entropy as

$$\frac{s_{\mathrm{ann}}^{(p,r)}(a,l)}{m\log\frac{1}{m}} = r - (r-1)l - \frac{r}{4}\frac{a^2}{1+(r-1)l^p}. \tag{136}$$

# G   Franz-Parisi analysis

To gain a more comprehensive understanding of the configuration space surrounding a typical configuration, we employ a free entropy landscape analysis. This approach constrains a spin configuration, the probe, to maintain a fixed overlap with a reference configuration sampled from the Gibbs measure. The framework was originally introduced by Franz and Parisi [36] as a potential that characterizes meta-stable state structures in mean-field spin glasses. This potential provides a characterization of the free entropy associated with maintaining a system at given temperature while conditioning a fixed overlap with an equilibrium configuration sampled at a different temperature. We should recall that in the MAS problem, there is a mapping between the average value of the submatrix and the temperature associated with the Gibbs measure.

To compute this potential, we introduce a field $\gamma$, that enforces a fixed overlap $\rho_{\mathrm{ref}} = q(\boldsymbol{\sigma}^{\mathrm{ref}}, \boldsymbol{\sigma}) \overset{!}{=} q_c$ between a system probe configuration $\boldsymbol{\sigma}$ and a quenched reference configuration $\boldsymbol{\sigma}^{\mathrm{ref}}$. Then we defined the Franz-Parisi partition function at fixed $\gamma$ as:

$$Z_c(\beta, J, q_{\mathrm{ref}}, \boldsymbol{\sigma}^{\mathrm{ref}}) = \sum_{\boldsymbol{\sigma}} e^{\beta E_J(\boldsymbol{\sigma}) + N\beta hm(\boldsymbol{\sigma}) + N\beta\beta_{\mathrm{ref}}\gamma q(\sigma_{\mathrm{ref}},\sigma)} \tag{137}$$

Therefore, the free entropy of the probe configuration $\boldsymbol{\sigma}$, the so-called Franz-Parisi potential, is:

$$\Psi_{FP} := \frac{1}{N}\mathbb{E}_{\boldsymbol{\sigma}^{\mathrm{ref}},J}\left[\log Z_c\right] = \mathbb{E}_J\left[\sum_{\boldsymbol{\sigma}^{\mathrm{ref}}} e^{\beta_{\mathrm{ref}} E(\boldsymbol{\sigma}^{\mathrm{ref}}) + N\beta_{\mathrm{ref}} h_{\mathrm{ref}} m(\boldsymbol{\sigma}^{\mathrm{ref}})} Z^{-1}\log Z_c\right]. \tag{138}$$

where the partition function of the Gibbs measure associated to the reference configuration at inverse temperature $\beta_{\mathrm{ref}}$ and magnetic field $h_{\mathrm{ref}}$ is

$$Z(\beta_{\mathrm{ref}}, J) = \sum_{\bar{\boldsymbol{\sigma}}} e^{\beta_{\mathrm{ref}} E(\bar{\boldsymbol{\sigma}}) + N\beta_{\mathrm{ref}} h_{\mathrm{ref}} m(\bar{\boldsymbol{\sigma}})}. \tag{139}$$

To compute the average with respect to the quenched variable, we make use of the replica trick, which allows us to bypass the difficulty of computing the expected value of the product of $Z^{-1}$ and the logarthm of the coupled partition function $Z_c$, in Eq.(138). This is achieve by introducing $n'$ integer replicas of the reference

system and $n$ integer replicas of the coupled system; to finally proceed to perform the analytical continuation of both functions:

$$N\Psi_{FP} = \lim_{n,n'\to 0} \mathbb{E}_J \left[ \sum_{\boldsymbol{\sigma}^{\text{ref}}} e^{\beta_{\text{ref}}E(\boldsymbol{\sigma}^{\text{ref}})+N\beta_{\text{ref}}h_{\text{ref}}m(\boldsymbol{\sigma}^{\text{ref}})} Z^{n'-1} \left( \frac{Z_c^n - 1}{n} \right) \right]. \tag{140}$$

The replicated partition functions are

$$Z_c^n = \sum_{\alpha} \sum_{\boldsymbol{\sigma}^{\alpha}} e^{\beta \sum_{\alpha} E(\boldsymbol{\sigma}^{\alpha})+N\beta h \sum_{\alpha} m(\boldsymbol{\sigma}^{\alpha})+N\beta\beta_{\text{ref}}\gamma q(\sigma_{\text{ref}},\sigma^{\alpha})}, \tag{141}$$

$$Z^{n'-1} = \sum_{a} \sum_{\bar{\boldsymbol{\sigma}}^a} e^{\beta_{\text{ref}} \sum_{\alpha} E(\bar{\boldsymbol{\sigma}}^a)+N\beta_{\text{ref}}h_{\text{ref}} \sum_a m(\bar{\boldsymbol{\sigma}}^a)}. \tag{142}$$

Here we use the upper indices $\alpha \in [n]$ and $a \in [n'-1]$ to indicate the replica of the system, and the lower indices $i \in \mathcal{N}$ to indicate the sites.

We introduce the replicated partition function of the FP potential, which is the central object of interest for its computation:

$$Z^{(n',n)} = \sum_{\boldsymbol{\sigma}^{\text{ref}}} e^{\beta_{\text{ref}}E(\boldsymbol{\sigma}^{\text{ref}})+N\beta_{\text{ref}}h_{\text{ref}}m(\boldsymbol{\sigma}^{\text{ref}})} Z^{n'-1} Z^n \tag{143}$$

## G.1  Average of the replicated partition function

The explicit form of the replicated partition function is

$$Z^{(n',n)} = \sum_{\boldsymbol{\sigma}^{\text{ref}}} \left\{ \exp \left( \frac{\beta_{\text{ref}}}{\sqrt{N}} \sum_{i<j} J_{ij}\sigma_i^{\text{ref}}\sigma_j^{\text{ref}} + \beta_{\text{ref}}h_{\text{ref}} \sum_i \sigma_i^{\text{ref}} \right) \right.$$
$$\cdot \text{Tr}_{\{\bar{\boldsymbol{\sigma}}^a\}_{a=1}^{n'-1}} \exp \left( \frac{\beta_{\text{ref}}}{\sqrt{N}} \sum_{i<j} J_{ij} \sum_{a=1}^{n'-1} \bar{\sigma}_i^a \bar{\sigma}_j^a + \beta_{\text{ref}}h_{\text{ref}} \sum_i \sum_{a=1}^{n'-1} \bar{\sigma}_i^a \right) \tag{144}$$
$$\left. \cdot \text{Tr}_{\{\boldsymbol{\sigma}^{\alpha}\}_{\alpha=1}^{n}} \exp \left( \frac{\beta}{\sqrt{N}} \sum_{i<j} J_{ij} \sum_{\alpha=1}^{n} \sigma_i^{\alpha}\sigma_j^{\alpha} + \beta h \sum_i \sum_{\alpha=1}^{n} \sigma_i^{\alpha} + \beta\beta_{\text{ref}}\gamma \sum_i \sum_{\alpha=1}^{n} \sigma_i^{\text{ref}}\sigma_i^{\alpha} \right) \right\}$$

Notice that the sum over $\boldsymbol{\sigma}^{\text{ref}}$ is equivalent to add an extra replica to the $(n'-1)$th power of the partition function $Z$; consequently, we can trace over $n'$ different replicas in the second line of the previous equation, but constraining only the replica $\bar{\boldsymbol{\sigma}}^1$ to have a fixed overlap with $\boldsymbol{\sigma}^{\alpha}$. Then, we can rewrite the replicated partition function as:

$$Z^{(n',n)} = \sum_{\{\bar{\boldsymbol{\sigma}}^a, \boldsymbol{\sigma}^{\alpha}\}} \exp \left[ \sum_{i<j} \frac{J_{ij}}{\sqrt{N}} \left( \beta_{\text{ref}} \sum_a \bar{\sigma}_i^a \bar{\sigma}_j^a + \beta \sum_{\alpha} \sigma_i^{\alpha}\sigma_j^{\alpha} \right) + \beta_{\text{ref}}h_{\text{ref}} \sum_{i,a} \bar{\sigma}_i^a + \beta h \sum_{i,\alpha} \sigma_i^{\alpha} \right.$$
$$\left. + \beta\beta_{\text{ref}}\gamma \sum_{i,\alpha} \sigma_i^{\text{ref}}\sigma_i^{\alpha} \right] \tag{145}$$

where now $a \in [n']$ and $\alpha \in [n]$.

Following the prescription of the replica method, after taking the expectation value with respect to $J_{ij}$, we have:

$$\mathbb{E}_J \left[ Z^{(n',n)} \right] = \sum_{\{\boldsymbol{\sigma}^a \boldsymbol{\sigma}^{\alpha}\}} \exp \left[ \frac{1}{2N} \sum_{i<j} \left( \beta_{\text{ref}} \sum_a \bar{\sigma}_i^a \bar{\sigma}_j^a + \beta \sum_{\alpha} \sigma_i^{\alpha}\sigma_j^{\alpha} \right)^2 + \right.$$
$$\left. \sum_i \left( \beta_{\text{ref}}h_{\text{ref}} \sum_a \bar{\sigma}_i^a + \beta h \sum_{\alpha} \sigma_i^{\alpha} + \beta\beta_{\text{ref}}\gamma \sum_{\alpha} \sigma_i^{\text{ref}}\sigma_i^{\alpha} \right) \right] \tag{146}$$

We conveniently reorganize the replicated partition function as

$$
\mathbb{E}[Z^{(n',n)}] = \sum_{\{\bar{\boldsymbol{\sigma}}^a\boldsymbol{\sigma}^\alpha\}} \exp\left[ \frac{\beta^2}{4N}\sum_\alpha \left(\sum_i \sigma_i^\alpha\right)^2 + \frac{\beta^2}{2N}\sum_{\alpha<\beta}\left(\sum_i \sigma_i^\alpha\sigma_i^\beta\right)^2 + \beta h\sum_\alpha\left(\sum_i \sigma_i^\alpha\right) \right.
$$

$$
+ \frac{(\beta_{\text{ref}})^2}{4N}\sum_a\left(\sum_i \bar{\sigma}_i^a\right)^2 + \frac{(\beta_{\text{ref}})^2}{2N}\sum_{a<b}\left(\sum_i \bar{\sigma}_i^a\bar{\sigma}_i^b\right)^2 + \beta_{\text{ref}}h_{\text{ref}}\sum_a\left(\sum_i \bar{\sigma}_i^a\right) \quad (147)
$$

$$
\left. + \frac{\beta\beta_{\text{ref}}}{2N}\sum_{a,\alpha}\left(\sum_i \bar{\sigma}^a\sigma^\alpha\right)^2 + \beta\beta_{\text{ref}}\gamma\sum_\alpha\sum_i \bar{\sigma}^1\sigma_i^\alpha \right]
$$

Here, we expressed each Dirac delta function in terms of its Fourier transform to extract the terms within the constraints. In this form, we identify five squared terms per replica, each of which can be linearized using a Hubbard-Stratonovich transformation. This transformation naturally introduces the usual overlaps, $m_\alpha$ and $q_{\alpha,\beta}$, for the constrained system, as well as the overlaps, $m_a^{\text{ref}}$ and $q_{a,b}^{\text{ref}}$ for the reference systems. Additionally, it introduces the overlap $\rho_{a,\alpha}$, which indicates the overlap between the replicas of the constrained system and the reference one. Thus the replicated partition function can be written as:

$$
\mathbb{E}[Z^{(n',n)}] = \int \prod_\alpha \mathrm{d}m_\alpha \prod_{\alpha<\beta}\mathrm{d}q_{\alpha,\beta}\prod_a \mathrm{d}m_a^{\text{ref}}\prod_{a<b}\mathrm{d}q_{a,b}^{\text{ref}}\prod_{a,\alpha}\mathrm{d}p_{a,\alpha}
$$

$$
\cdot \exp\left( -\frac{N\beta^2}{4}\sum_\alpha m_\alpha^2 + \frac{\beta^2}{2}\sum_\alpha m_\alpha \sum_i \sigma_i^\alpha - \frac{N\beta^2}{2}\sum_{\alpha<\beta}q_{\alpha,\beta}^2 + \beta^2\sum_{\alpha<\beta}q_{\alpha,\beta}\sum_i \sigma_i^\alpha\sigma_i^\beta \right)
$$

$$
\cdot \exp\left( -\frac{N(\beta_{\text{ref}})^2}{4}\sum_a (m_a^{\text{ref}})^2 + \frac{(\beta_{\text{ref}})^2}{2}\sum_a m_a^{\text{ref}}\sum_i \bar{\sigma}_i^a - \frac{N(\beta_{\text{ref}})^2}{2}\sum_{a<b}(q_{a,b}^{\text{ref}})^2 + (\beta_{\text{ref}})^2\sum_{a<b}q_{a,b}^{\text{ref}}\sum_i \sigma_i^a\sigma_i^b \right)
$$

$$
\cdot \exp\left( -\frac{N\beta\beta_{\text{ref}}}{2}\sum_{a,\alpha}\rho_{a,\alpha} + \beta\beta_{\text{ref}}\sum_{a,\alpha}\sum_i \bar{\sigma}_i^a\sigma_i^\alpha + h_{\text{ref}}\beta_{\text{ref}}\sum_a\sum_i \bar{\sigma}^a \right)
$$

$$
\cdot \exp\left( \beta\sum_\alpha\sum_i (h + \beta_{\text{ref}}\gamma\bar{\sigma}_i^1)\sigma_i^\alpha + \mathcal{O}(\log N) \right)
$$

$$(148)$$

After tracing over each independent site $i$, the expression becomes

$$
\mathbb{E}[Z^{(n',n)}] = \int \prod_\alpha \mathrm{d}m_\alpha \prod_{\alpha<\beta}\mathrm{d}q_{\alpha,\beta}\prod_a \mathrm{d}m_a^{\text{ref}}\prod_{a<b}\mathrm{d}q_{a,b}^{\text{ref}}\prod_{a,\alpha}\mathrm{d}p_{a,\alpha}
$$

$$
\cdot \exp\left( -\frac{\beta^2}{4}\sum_\alpha m_\alpha^2 - \frac{\beta^2}{2}\sum_{\alpha<\beta}q_{\alpha,\beta}^2 - \frac{(\beta_{\text{ref}})^2}{4}\sum_a (m_a^{\text{ref}})^2 - \frac{(\beta_{\text{ref}})^2}{2}\sum_{a<b}(q_{a,b}^{\text{ref}})^2 - \frac{\beta\beta_{\text{ref}}}{2}\sum_{a,\alpha}p_{a,\alpha}^2 \right)^N \quad (149)
$$

$$
\cdot \exp\left( N\log(\psi_e) \right)
$$

where we had defined

$$
\psi_s := \mathrm{Tr}_{\{\bar{\sigma}^a,\sigma^\alpha\}}\exp\left( \frac{\beta^2}{2}\sum_\alpha m_\alpha\sigma^\alpha + \beta^2\sum_{\alpha<\beta}q_{\alpha,\beta}\sigma^\alpha\sigma^\beta + \beta\sum_\alpha (h + \beta_{\text{ref}}\bar{\sigma}^1\gamma)\sigma^\alpha \right.
$$

$$(150)$$

$$
\left. + \frac{(\beta_{\text{ref}})^2}{2}\sum_a m_a^{\text{ref}}\bar{\sigma}^a + (\beta_{\text{ref}})^2\sum_{a<b}q_{a,b}^{\text{ref}}\bar{\sigma}^a\bar{\sigma}^b + \beta_{\text{ref}}h_{\text{ref}}\sum_a \bar{\sigma}^a + \beta\beta_{\text{ref}}\sum_{a,\alpha}\rho_{a,\alpha}\bar{\sigma}^a\sigma^\alpha \right)
$$

We use the replica symmetry assumption to study the Franz-Parisi potential, given our interest in the frozen 1-RSB phase where the Parisi parameter is equal to one, which makes the free entropy well described

by the replica symmetric solution. However, we recall that the 1-RSB assumption and the Parisi parameter also allow us to compute the number of typical clusters in the system, as explained in [3]. Therefore, after imposing the RS ansatz for the overlap matrices and taking the limits $n', n \to 0$ as prescribed by the replica trick:

$$\Psi_{FP} = \Psi_s + \Psi_e \tag{151}$$

where we define the RS energetic potential

$$\Psi_e = -\frac{\beta^2}{4}(m^2 - q^2) - \frac{\beta\beta_{\text{ref}}}{2}(\rho_{\text{ref}}^2 - \rho^2) \tag{152}$$

And the entropic potential

$$\Psi_s = \lim_{n',n\to 0} \frac{1}{n} \log\left(\psi_s\right) = \lim_{n'\to 0}(\partial_n \psi_s|_{n=0} + n' \partial_{n'} \psi_s|_{n=0})$$

$$\psi_s = \text{Tr}_{\{\bar\sigma^a, \sigma^\alpha\}} \exp\left(\frac{\beta^2(m-q)}{2}\sum_\alpha \sigma^\alpha + \frac{\beta^2 q}{2}\left(\sum_\alpha \sigma^\alpha\right)^2 + (h + \beta_{\text{ref}}\bar\sigma^1 \gamma)\beta \sum_\alpha \sigma^\alpha\right.$$

$$+ \frac{(\beta_{\text{ref}})^2(m_{\text{ref}} - q_{\text{ref}})}{2}\sum_a \bar\sigma^a + \frac{(\beta_{\text{ref}})^2 q_{\text{ref}}}{2}\left(\sum_a \bar\sigma^a\right)^2 + \beta_{\text{ref}} h_{\text{ref}}\sum_a \bar\sigma^a + \beta\beta_{\text{ref}}\rho_{\text{ref}}\sum_\alpha \bar\sigma^1 \sigma^\alpha$$

$$\left.+ \beta\beta_{\text{ref}}\rho \sum_{a\neq 1,\alpha} \bar\sigma^a \sigma^\alpha\right)$$

To compute the trace over each replica, we need the replicas to be decoupled; that is, the exponent must be linear in each replicated spin $\sigma^\alpha$ and $\bar\sigma^a$. To achieve this, we use a similar algebraical trick to the one employed to linearize the replicas of the same system, which allows us to apply a Hubbard-Stratonovich transformation.

$$\sum_{a\neq 1,\alpha} \bar\sigma^a \sigma^\alpha = -\sum_\alpha \bar\sigma^1 \sigma^\alpha + \sum_{a,\alpha} \bar\sigma^a \sigma^\alpha$$

$$= -\sum_\alpha \bar\sigma^1 \sigma^\alpha + \frac{1}{2}\left(\sum_a \bar\sigma^a + \sum_\alpha \sigma^\alpha\right)^2 - \frac{1}{2}\left(\sum_a \bar\sigma^a\right)^2 - \frac{1}{2}\left(\sum_\alpha \sigma^\alpha\right)^2 \tag{153}$$

Applying this transformation on the entropic potential, we obtain

$$\psi_s = \text{Tr}_{\{\bar\sigma^a, \sigma^\alpha\}} \exp\left(\frac{\beta^2(m-q)}{2}\sum_\alpha \sigma^\alpha + \frac{\beta^2 q - \beta\beta_{\text{ref}}\rho}{2}\left(\sum_\alpha \sigma^\alpha\right)^2 + (h + \beta_{\text{ref}}\bar\sigma^1 \gamma)\beta \sum_\alpha \sigma^\alpha\right.$$

$$+ \frac{(\beta_{\text{ref}})^2(m_{\text{ref}} - q_{\text{ref}})}{2}\sum_a \bar\sigma^a + \frac{(\beta_{\text{ref}})^2 q_{\text{ref}} - \beta\beta_{\text{ref}}\rho}{2}\left(\sum_a \bar\sigma^a\right)^2 + \beta_{\text{ref}} h'\sum_a \bar\sigma^a$$

$$\left.+ \beta\beta_{\text{ref}}(\rho_{\text{ref}} - \rho)\sum_\alpha \bar\sigma^1 \sigma^\alpha + \frac{\beta\beta_{\text{ref}}\rho}{2}\left(\sum_a \bar\sigma^a + \sum_\alpha \sigma^\alpha\right)^2\right)$$

$$= \text{Tr}_{\{\bar\sigma^a, \sigma^\alpha\}} \int Dx_1 Dx_2 Dx_3 \exp\left(\frac{\beta^2(m-q)}{2}\sum_\alpha \sigma^\alpha + x_1\sqrt{\beta^2 q - \beta\beta_{\text{ref}}\rho}\sum_\alpha \sigma^\alpha + (h + \beta_{\text{ref}}\bar\sigma^1 \gamma)\beta \sum_\alpha \sigma^\alpha\right.$$

$$+ \frac{(\beta_{\text{ref}})^2(m_{\text{ref}} - q_{\text{ref}})}{2}\sum_a \bar\sigma^a + x_2\sqrt{(\beta_{\text{ref}})^2 q_{\text{ref}} - \beta\beta_{\text{ref}}\rho}\sum_a \bar\sigma^a + \beta_{\text{ref}} h_{\text{ref}}\sum_a \bar\sigma^a + \beta\beta_{\text{ref}}(\rho_{\text{ref}} - \rho)\sum_\alpha \bar\sigma^1 \sigma^\alpha$$

$$\left.+ x_3\sqrt{\beta\beta_{\text{ref}}\rho}\left(\sum_a \bar\sigma^a + \sum_\alpha \sigma^\alpha\right)\right) \tag{154}$$

We reorder the trace terms by isolating those proportional to $\bar{\sigma}^1$, for which we will take the trace after *tracing* over the rest of the replicas.

$$
\psi_s = \text{Tr}_{\bar{\sigma}^1} \int Dx_1 Dx_2 Dx_3 \exp \left( \frac{(\beta_{\text{ref}})^2 (m_{\text{ref}} - q_{\text{ref}})}{2} \bar{\sigma}^1 + x_2 \sqrt{(\beta_{\text{ref}})^2 q_{\text{ref}} - \beta\beta_{\text{ref}}\rho} \bar{\sigma}^1 + (\beta_{\text{ref}} h_{\text{ref}} + x_3 \sqrt{\beta\beta_{\text{ref}}\rho}) \bar{\sigma}^1 \right)
$$

$$
\cdot \left[ \text{Tr}_{\sigma^\alpha} \exp \left( \frac{\beta^2 (m-q)}{2} \sigma^\alpha + x_1 \sqrt{\beta^2 q - \beta\beta_{\text{ref}}\rho} \sigma^\alpha + (\beta h + \beta\beta_{\text{ref}}\bar{\sigma}^1\gamma + x_3 \sqrt{\beta\beta_{\text{ref}}\rho})\sigma^\alpha + \beta\beta_{\text{ref}}(\rho_{\text{ref}} - \rho)\bar{\sigma}^1\sigma^\alpha \right) \right]^n
$$

$$
\cdot \left[ \text{Tr}_{\bar{\sigma}^a} \exp \left( \frac{(\beta_{\text{ref}})^2 (m_{\text{ref}} - q_{\text{ref}})}{2} \bar{\sigma}^a + x_2 \sqrt{(\beta_{\text{ref}})^2 q_{\text{ref}} - \beta\beta_{\text{ref}}\rho} \bar{\sigma}^a + (\beta_{\text{ref}} h_{\text{ref}} + x_3 \sqrt{\beta\beta_{\text{ref}}\rho}) \bar{\sigma}^a \right) \right]^{n'-1}
$$

$$(155)$$

$$
\psi_s = \text{Tr}_{\bar{\sigma}^1} \int Dx_1 Dx_2 Dx_3 \exp \left( \frac{(\beta_{\text{ref}})^2 (m_{\text{ref}} - q_{\text{ref}})}{2} \bar{\sigma}^1 + x_2 \sqrt{(\beta_{\text{ref}})^2 q_{\text{ref}} - \beta\beta_{\text{ref}}\rho} \bar{\sigma}^1 + (\beta_{\text{ref}} h_{\text{ref}} + x_3 \sqrt{\beta\beta_{\text{ref}}\rho}) \bar{\sigma}^1 \right)
$$

$$
\cdot \exp \left[ n \log 1\text{pexp} \left( \frac{\beta^2 (m-q)}{2} + x_1 \sqrt{\beta^2 q - \beta\beta_{\text{ref}}\rho} + (\beta h + \beta\beta_{\text{ref}}\bar{\sigma}^1\gamma + x_3 \sqrt{\beta\beta_{\text{ref}}\rho}) + \beta\beta_{\text{ref}}(\rho_{\text{ref}} - \rho)\bar{\sigma}^1 \right) \right]
$$

$$
\cdot \exp \left[ (n'-1) \log 1\text{pexp} \left( \frac{(\beta_{\text{ref}})^2 (m_{\text{ref}} - q_{\text{ref}})}{2} + x_2 \sqrt{(\beta_{\text{ref}})^2 q_{\text{ref}} - \beta\beta_{\text{ref}}\rho} + (\beta_{\text{ref}} h_{\text{ref}} + x_3 \sqrt{\beta\beta_{\text{ref}}\rho}) \right) \right]
$$

$$(156)$$

To simplify the integrals, let's consider the following unitary transformations:

$$
\begin{pmatrix} y_2 \\ y_3 \end{pmatrix} = \frac{1}{\beta_{\text{ref}} \sqrt{q_{\text{ref}}}} \begin{pmatrix} \sqrt{(\beta_{\text{ref}})^2 q_{\text{ref}} - \beta\beta_{\text{ref}}\rho} & \sqrt{\beta\beta_{\text{ref}}\rho} \\ -\sqrt{\beta\beta_{\text{ref}}\rho} & \sqrt{(\beta_{\text{ref}})^2 q_{\text{ref}} - \beta\beta_{\text{ref}}\rho} \end{pmatrix} \begin{pmatrix} x_2 \\ x_3 \end{pmatrix}
$$

$$
\implies x_3 \sqrt{\beta\beta_{\text{ref}}\rho} = y_2 \frac{\beta\rho}{\sqrt{q_{\text{ref}}}} + y_3 \sqrt{\frac{\beta\beta_{\text{ref}} q_{\text{ref}}\rho - \beta^2\rho^2}{q_{\text{ref}}}}
$$

$$(157)$$

then

$$
\psi_s = \text{Tr}_{\bar{\sigma}^1} \int Dy_1 Dx_2 Dy_3 \exp \left( \frac{(\beta_{\text{ref}})^2 (m_{\text{ref}} - q_{\text{ref}})}{2} + \beta_{\text{ref}} h_{\text{ref}} + y_2 \beta_{\text{ref}} \sqrt{q_{\text{ref}}} \right)^{\bar{\sigma}^1}
$$

$$
\cdot \exp \left[ n \log 1\text{pexp} \left( \frac{\beta^2 (m-q)}{2} + \beta h + \beta\beta_{\text{ref}}\bar{\sigma}^1\gamma + \beta\beta_{\text{ref}}(\rho_{\text{ref}} - \rho)\bar{\sigma}^1 \right. \right.
$$

$$
\left. \left. + x_1 \sqrt{\beta^2 q - \beta\beta_{\text{ref}}\rho} + y_2 \frac{\beta\rho}{\sqrt{q_{\text{ref}}}} + y_3 \sqrt{\frac{\beta\beta_{\text{ref}} q_{\text{ref}}\rho - \beta^2\rho^2}{q_{\text{ref}}}} \right) \right]
$$

$$
\cdot \exp \left[ (n'-1) \log 1\text{pexp} \left( \frac{(\beta_{\text{ref}})^2 (m_{\text{ref}} - q_{\text{ref}})}{2} + \beta_{\text{ref}} h_{\text{ref}} + y_2 \beta_{\text{ref}} \sqrt{q_{\text{ref}}} \right) \right]
$$

$$(158)$$

and one extra unitary transformation:

$$
\begin{pmatrix} y_1 \\ y_3' \end{pmatrix} = \frac{1}{\beta \sqrt{(q - \rho^2/q_{\text{ref}})}} \begin{pmatrix} \sqrt{\beta^2 q - \beta\beta_{\text{ref}}\rho} & \sqrt{\frac{\beta\beta_{\text{ref}} q_{\text{ref}}\rho - \beta^2\rho^2}{q_{\text{ref}}}} \\ -\sqrt{\frac{\beta\beta_{\text{ref}} q_{\text{ref}}\rho - \beta^2\rho^2}{q_{\text{ref}}}} & \sqrt{\beta^2 q - \beta\beta_{\text{ref}}\rho} \end{pmatrix} \begin{pmatrix} x_2 \\ y_3 \end{pmatrix}
$$

$$(159)$$

We manage to reduce the number of gaussian integrals:

$$
\psi_s = \text{Tr}_{\bar{\sigma}^1} \int Dy_1 Dy_2 \exp\left(\frac{(\beta_{\text{ref}})^2(m_{\text{ref}} - q_{\text{ref}})}{2} + \beta_{\text{ref}}h_{\text{ref}} + y_2\beta_{\text{ref}}\sqrt{q_{\text{ref}}}\right)^{\bar{\sigma}^1}
$$

$$
\cdot \exp\left[n\log 1\text{pexp}\left(\frac{\beta^2(m-q)}{2} + \beta h + \beta\beta_{\text{ref}}\bar{\sigma}^1\gamma + \beta\beta_{\text{ref}}(\rho_{\text{ref}} - \rho)\bar{\sigma}^1 + y_1\beta\sqrt{q - \rho^2/q_{\text{ref}}} + y_2\beta\rho/\sqrt{q_{\text{ref}}}\right)\right]
$$

$$
\cdot \exp\left[(n' - 1)\log 1\text{pexp}\left(\frac{(\beta_{\text{ref}})^2(m_{\text{ref}} - q_{\text{ref}})}{2} + \beta_{\text{ref}}h_{\text{ref}} + y_2\beta_{\text{ref}}\sqrt{q_{\text{ref}}}\right)\right]
\tag{160}
$$

After taking the replica limit

$$
\Psi_s = \int Dy_1 Dy_2 \ell\left(-\beta_{\text{ref}}H_{\text{RS}}^{\text{ref}}(y_2)\right)\log 1\text{pexp}\left(\beta H_C(y_1, y_2)\right)
$$
$$
+ \int Dy_1 Dy_2 \ell\left(\beta_{\text{ref}}H_{\text{RS}}^{\text{ref}}(y_2)\right)\log 1\text{pexp}\left(\beta H_C(y_1, y_2) + \beta\beta_{\text{ref}}(\gamma + \rho_{\text{ref}} - \rho)\right)
\tag{161}
$$

where

$$
H_{\text{RS}}^{\text{ref}}(y_2) = \frac{\beta_{\text{ref}}}{2}(m_{\text{ref}} - q_{\text{ref}}) + h_{\text{ref}} + y_2\sqrt{q_{\text{ref}}}
\tag{162}
$$

$$
H_C(y_1, y_2) = \frac{\beta}{2}(m - q) + h + y_1\sqrt{q - \rho^2/q_{\text{ref}}} + y_2\rho/\sqrt{q_{\text{ref}}}
\tag{163}
$$

Finally, we obtain the free entropy of the probe system at fixed $h$:

$$
\Psi_{FP} = \Psi_e + \Psi_s
$$
$$
\Psi_e = -\frac{\beta^2}{4}(m^2 - q^2) - \frac{\beta\beta_{\text{ref}}}{2}(\rho_{\text{ref}}^2 - \rho^2)
$$
$$
\Psi_s = \int Dy_1 Dy_2 \, \ell\left(-\beta_{\text{ref}}H_{\text{RS}}^{\text{ref}}(y_2)\right)\log 1\text{pexp}\left(\beta H_C(y_1, y_2)\right)
$$
$$
+ \int Dy_1 Dy_2 \, \ell\left(\beta_{\text{ref}}H_{\text{RS}}^{\text{ref}}(y_2)\right)\log 1\text{pexp}\left(\beta H_C(y_1, y_2) + \beta\beta_{\text{ref}}(\gamma + \rho_{\text{ref}} - \rho)\right)
\tag{164}
$$

Then the Franz-Parisi potential which is defined as the free entropy of the probe system at fixed magnetization $m$ and overlap with the reference system $\rho \overset{!}{=} q_c$, it is provided by the Legendre transform:

$$
\Phi_{FP} = \Psi_{FP}(m, m_{\text{ref}}, \beta, \beta_{\text{ref}}) - \beta h m - \beta\beta_{\text{ref}}\gamma q_c
\tag{165}
$$

## G.2 Saddle-point equations

Before computing the SP, we recall that the reference configuration is sampled independently from the Gibbs measure. Consequently, the typical values of the order parameters, $q_{\text{ref}} = q_{RS}$ and $h' = h_{RS}$, correspond to the values that extremize the RS variational free entropy. This result can be recovered by taking the limit $n \to 0$ and extremizing with respect to $q_{\text{ref}}$ and $m_{\text{ref}}$ the terms proportional to $n'$, being $n'$ non-zero. From this, we obtain the variational free entropy of the reference configuration at equilibrium, along with the corresponding saddle-point equations:

$$
m_{\text{ref}} = \int Dy \ell(\beta_{\text{ref}}H_{\text{RS}}^{\text{ref}}(y))
$$
$$
q_{\text{ref}} = \int Dy \ell^2(\beta_{\text{ref}}H_{\text{RS}}^{\text{ref}}(y))
\tag{166}
$$

Then we only need to solve the SP - equations for the constrained system :

$$\rho = \int \mathrm{D}y_1 \mathrm{D}y_2 \left[ \ell(\beta_{\text{ref}} H_{\text{RS}}^{\text{ref}}) \ell(\beta H_C) - \ell^2(\beta_{\text{ref}} H_{\text{RS}}^{\text{ref}}) \left[ \ell(\beta H_C) - \ell(\beta H_C + \beta\beta_{\text{ref}}(\gamma + \rho_{\text{ref}} - \rho)) \right] \right]$$

$$m = \int \mathrm{D}y_1 \mathrm{D}y_2 \left[ \ell(\beta H_C) - \ell(\beta_{\text{ref}} H_{\text{RS}}^{\text{ref}}) \left[ \ell(\beta H_C) - \ell(\beta H_C + \beta\beta_{\text{ref}}(\gamma + \rho_{\text{ref}} - \rho)) \right] \right]$$

$$q = \int \mathrm{D}y_1 \mathrm{D}y_2 \left[ \ell^2(\beta H_C) - \ell(\beta_{\text{ref}} H_{\text{RS}}^{\text{ref}}) \left[ \ell^2(\beta H_C) - \ell^2(\beta H_C + \beta\beta_{\text{ref}}(\gamma + \rho_{\text{ref}} - \rho)) \right] \right]$$

$$\rho_{\text{ref}} = \int \mathrm{D}x_1 \, \mathrm{D}x_2 \, \ell(\beta_{\text{ref}} H_{\text{RS}}^{\text{ref}}) \ell\left(\beta H_C + \beta\beta_{\text{ref}}(\gamma + \rho_{\text{ref}} - \rho)\right)$$

(167)

## G.3  The small magnetization limit – Saddle point equations

We now turn of attention to the study of te FP potential in the limit where $m_{\text{ref}} = m \to 0$, which corresponds to the regime where $k \ll N$. By using the same combinatorial argument to re-scale the temperature and thus the energy of the system, comparable to the entropy; we impose the leading order terms for the following parameters:

$$\beta_{\text{ref}} = b_{\text{ref}} \sqrt{\frac{1}{m} \log \frac{1}{m}} \qquad\qquad \beta = b \sqrt{\left( \log \frac{1}{m} \right) \frac{1}{m}}$$

$$q_{\text{ref}} = m^2 \qquad\qquad q = r^2 m$$

$$h_{\text{ref}} = -\eta_{\text{ref}} \sqrt{m \log \frac{1}{m}} \qquad\qquad h = -\eta \sqrt{m \left( \log \frac{1}{m} \right)}$$

(168)

With this scaling, we see that the dependency on $y_1$ in $H_{\text{RS}}^{\text{ref}}(y_1)$ and $H_C(y_1, y_2)$ drops, and from the SP equations of the reference system (166), we have

$$m = \ell(\beta_{\text{ref}} H_{\text{RS}}^{\text{ref}})$$

$$\implies \eta_{\text{ref}} = -\frac{1}{2} \left( \frac{1}{b_{\text{ref}}} + \frac{b_{\text{ref}}}{2} \right)$$

(169)

The SP equation (167) of the overlap $\rho$ between replica of the reference system and the probe becomes:

$$\rho = \ell(\beta_{\text{ref}} H_{\text{RS}}^{\text{ref}}) \underbrace{\int \mathrm{D}y_1 \mathrm{D}y_2 \left[ \ell(\beta H_C) - \ell(\beta_{\text{ref}} H_{RS}) \left[ \ell(\beta H_C) - \ell(\beta H_C + \beta\beta_{\text{ref}}(\gamma + \rho_{\text{ref}} - \rho)) \right] \right]}_{=m, \text{ from SP (167) w.r.t. } m} \implies \rho = m^2$$

(170)

Additionally, we propose the scaling $\rho_{\text{ref}} \overset{!}{=} q_{\text{ref}} = mc$, where $0 \leq c \leq 1$, and $\gamma = m\nu$. Then, in the limit $m \to 0$, the set of saddle point equations for the FP potential are

$$m = \int \mathrm{D}v \, \ell(\beta H_C) + cm$$

$$r^2 m = \int \mathrm{D}v \, \ell^2(\beta H_C) + m \int \mathrm{D}v \, \ell^2(\beta H_C + \beta\beta_{\text{ref}}(\gamma + q_{\text{ref}}))$$

(171)

$$c = \int \mathrm{D}v \ell(\beta H_C + \beta\beta_{\text{ref}}(\gamma + q_{\text{ref}}))$$

where

$$H_C(v) = h + \frac{\beta}{2}(m - q) + v\sqrt{q}$$

(172)

Note that we have retained only the terms of order $\mathcal{O}(m)$ in the Taylor series expansion, so the equations should be understood as equal only to leading order. We have three equations for the three parameters $\eta$, $r$ and $\nu$, while the remaining variables $c$, $b$ and $b_{\text{ref}}$ serve as control parameters.

As mentioned in section 1.2, given the symmetry of the distribution of the matrix $J$, the inverse temperature $\beta$ can be negative, and so $b$. However, the asymptotic expansion we have are always for positive $\beta$, since the two cases can be mapped one onto each other by simply performing the change of variables $v \mapsto -v$ in the Gaussian integral and $h \mapsto -h$. The case $b = 0$ will be treated separately. Similar as it was done before, it is convenient to write the auxiliary term $\beta H_C(v) = AN + B\sqrt{N}$ to asymptotically compute the integrals of the saddle-point equations. Thus, in the scaling limit we we have:

$$
\begin{aligned}
A &= -b\eta + \frac{b^2}{2}(1 - r^2) \\
B &= br \\
C &= bb_{\mathrm{ref}}(c + \nu) \\
N &= \log\frac{1}{m}
\end{aligned}
\tag{173}
$$

The saddle-point equations are

$$
\begin{aligned}
(1 - c)m &= I_1(A, B) \\
c &= I_1(A + C, B) \\
r^2 m &= I_2(A, B) + m I_2(A + C, B)
\end{aligned}
\tag{174}
$$

and the rescaled Franz-Parisi potential is

$$
\begin{aligned}
\phi_{\mathrm{FP}} &:= \frac{\Phi_{\mathrm{FP}}}{m\log(1/m)} = -\frac{b^2}{4}(1 - r^4) - \frac{bb_{\mathrm{ref}}}{2}c^2 + b\eta - bb_{\mathrm{ref}}\nu c + S \\
S &:= \mathcal{G}(A, B) + m\mathcal{G}(A + C, B).
\end{aligned}
\tag{175}
$$

Here we had defined the Gaussian integral

$$
\begin{aligned}
I_l(A, B) &:= \int \mathrm{D}v\, \ell^l(AN + B\sqrt{N}) \\
\mathcal{G}(A, B) &:= \frac{1}{N} \int \mathrm{D}v\, \log(1 + e^{AN + B\sqrt{N}})
\end{aligned}
\tag{176}
$$

The asymptotics for the integrals at the leading order[3] are

$$
I_l(A, B) \approx
\begin{cases}
G\left(\frac{A}{B}\sqrt{N}\right) & -\frac{A}{B^2} \leq 0 \\
\frac{\exp\left(-\frac{A^2}{2B^2}N\right)}{\sqrt{2\pi N}} K(0, l, A, B) & 0 < -\frac{A}{B^2} < l \\
e^{N f_{A,B}(l)} G\left(-\frac{A}{B}\sqrt{N} - lB\sqrt{N}\right) & -\frac{A}{B^2} \geq l
\end{cases}
\tag{177}
$$

$$
\mathcal{G}(A, B) \approx
\begin{cases}
A\, G\left(\frac{A}{B}\sqrt{N}\right) & -\frac{A}{B^2} \leq 0 \\
\frac{B}{\sqrt{2\pi N}} e^{-\frac{A^2}{2B^2}N} & 0 < -\frac{A}{B^2} < 1 \\
e^{AN + \frac{1}{2}B^2 N} G\left(-\frac{A}{B}\sqrt{N} - B\sqrt{N}\right) & -\frac{A}{B^2} \geq 1
\end{cases}
\tag{178}
$$

with

$$
\begin{aligned}
f_{A,B}(k) &= kA + \frac{1}{2}B^2 k^2 + \mathcal{O}(N^{-1}) \\
K(0, 1, x, y) &= -\frac{\pi \csc\left(\frac{\pi x}{y^2}\right)}{y} \\
K(0, 2, x, y) &= -\frac{\pi(x + y^2)\csc\left(\frac{\pi x}{y^2}\right)}{y^3} = \frac{x + y^2}{y^2} K(0, 1, x, y) \\
G(z) &= \frac{1}{2}\left(1 + \mathrm{erf}\left(\frac{z}{\sqrt{2}}\right)\right)
\end{aligned}
\tag{179}
$$

### G.3.1 Overlap with the reference configuration equation

We begin computing the **equation for c**, which is related to the overlap between the probe and the reference system. We observe that by imposing the condition that the integral $I_1(A + C, B) \overset{!}{=} \mathcal{O}(1)$, we require $A + C \geq 0$. In that case:

$$c = G\left(\frac{A+C}{B}\sqrt{N}\right) \tag{180}$$

If instead $A + C < 0$, we have two cases

- if $-B^2 < A + C < 0$ we have the equation

$$c = \frac{\exp\left(-\frac{(A+C)^2}{2B^2}N\right)}{\sqrt{2\pi N}} K(0, 1, A_2 + C, B) \tag{181}$$

  implying

$$\frac{\exp\left(-\frac{(A+C)^2}{2B^2}N\right)}{\sqrt{2\pi N}} = \mathcal{O}(1) \implies A + C = 0 \tag{182}$$

  which is a contradiction.

- if $A + C < -B^2$ we have the equation

$$c = e^{N f_{A+C,B}(1)} G\left(-\frac{A}{B}\sqrt{N} - B\sqrt{N}\right) \tag{183}$$

  implying

$$f_{A+C,B}(1) = 0 \implies A + C = -\frac{1}{2}B^2 \tag{184}$$

  under the condition that

$$A + C = -\frac{1}{2}B^2 < -B^2 \implies \frac{1}{2}B^2 < 0 \tag{185}$$

  which is a contradiction.

Thus, the only possible choice is $A + C \geq 0$ and

$$c = G\left(\frac{A+C}{B}\sqrt{N}\right) \tag{186}$$

In the case where $c = 1$, this is equivalent to setting $\rho_{\text{ref}} = m$ and requiring $A + C > 0$. This is because $0 \leq \ell(\cdot) \leq 1$, which implies that $I_2(A, B) \leq I_1(A, B)$, and by using the magnetization saddle point equation of the probe, we obtain $I_1(A, B) = 0 \implies I_2(A, B) = 0$. Thus, from the overlap saddle-point equation, we find $r^2 = c = 1$. The asymptotic expansion at which $I_2(A, B) = 0$ corresponds to the case where $-A/B^2 < 0$, implying that the leading-order expression for the entropic potential is:

$$S = bb_{\text{ref}}(1 + \nu) - b\eta \tag{187}$$

In this case, we observe that the terms $b\eta$ and $bb_{\text{ref}}\nu$ cancel out in the Franz-Parisi potential (175), meaning they are not needed to compute the observables. Therefore, the relevant Franz-Parisi potential, the probe sub-matrix average, and the probe entropy can be computed directly from the remaining terms. Specifically, we obtain for $c = 1$:

$$s_{\text{FP}} = \frac{bb_{\text{ref}}}{2}mN^2$$
$$a = b_{\text{ref}} \tag{188}$$
$$s_{\text{probe}} = 0.$$

For this reason, we will focus only on the case $0 \le c < 1$ to study the rest of saddle point equations, where we require at least that the leading order of the argument of $G\left(\frac{A+C}{B}\sqrt{N}\right)$ is $\mathcal{O}(1)$, which leads to the condition $A + C = 0$, i.e., an equation for $\eta$ of the form $\eta = \eta(\nu, r; b, b_{\text{ref}}, c)$.

$$bb_{\text{ref}}(c + \nu) = b\left[\eta - \frac{b}{2}(1 - r^2)\right] \implies \eta = b_{\text{ref}}(c + \nu) + \frac{b}{2}(1 - r^2). \tag{189}$$

Moreover, under these assumptions, the overlap equation simplifies to

$$I_2(A + C, B) \sim c \implies (r^2 - c)m \sim I_2(A, B). \tag{190}$$

Finally, the entropic potential, at the leading order, is given by $S = \mathcal{G}(A, B)$.

### G.3.2   Magnetization equation

We now study the saddle-point equation related to the magnetization of the probe system:

$$(1 - c)m = I_1(A, B), \tag{191}$$

given the condition $A + C = 0$. We need that both sides of the equations are of the same order in $m$, meaning that $I_1(A, B)$ must at least $\mathcal{O}(m)$. Thus we analyze the three possible cases of this integral:

- If $A \ge 0$ then $I_1 = \mathcal{O}(1)$ making the equation inconsistent.

- If $-B^2 < A < 0$, then the equation is

$$(1 - c)m = \frac{\exp\left(-\frac{A^2}{2B^2}N\right)}{\sqrt{2\pi N}}K(0, 1, A_2, B_1) \tag{192}$$

  implying

$$\frac{\exp\left(-\frac{A^2}{2B^2}N\right)}{\sqrt{2\pi N}} = \kappa m \implies \frac{A^2}{2B^2} = 1 \implies A = -\sqrt{2}B \implies \eta - \frac{b}{2}(1 - r^2) = \sqrt{2}r \tag{193}$$

  for some positive $\kappa$, giving

$$\eta(r) = \sqrt{2}r + \frac{b}{2}(1 - r^2)$$
$$\nu(r) = \frac{\sqrt{2}r}{b_{\text{ref}}} - c, \tag{194}$$

  where the last expression comes from the assumption $A + C = 0$. Then, the integral condition becomes

$$-B^2 < A \implies |b|r > \sqrt{2}, \tag{195}$$

  and the equiation to be solved becomes

$$1 - c = \kappa K(0, 1, -\sqrt{2}|b|r, |b|r), \tag{196}$$

  which gives $\kappa$ as function of $r$. In this case the entropic potential is

$$S = \frac{B}{\sqrt{2\pi N}}e^{-\frac{A^2}{2B^2}N} = \frac{B}{\sqrt{2\pi N}}m \ll mN \tag{197}$$

- If $A < -B^2$, the equation is:

$$(1 - c)m = e^{f_{A,B}(1)}G\left(-\frac{A + B^2}{B}\sqrt{N}\right) \tag{198}$$

implying

$$e^{Nf_{A,B}(1)} = m^{f_{A,B}(1)} = \kappa m \implies A = -1 - \frac{1}{2} \implies \eta = \frac{b^2 + 2}{2b} \tag{199}$$

for some positive $\kappa$, giving an equation

$$\frac{b^2 + 2}{2b} = b_{\text{ref}}(c + \eta) + \frac{b}{2}(1 - r^2) \implies \nu(r) = \frac{b}{2b_{\text{ref}}}r^2 + \frac{1}{bb_{\text{ref}}} - c \tag{200}$$

The integral condition becomes $A < -B^2$ becomes

$$-1 - \frac{1}{2}B^2 + B^2 \leq 0 \implies |b|r \leq \sqrt{2} \tag{201}$$

and we can rewrite the saddle-point equation as

$$1 - c = \kappa G\left(-\frac{A + B^2}{B}\sqrt{N}\right), \tag{202}$$

where $\kappa$ comes from the sub-leading terms of the integral asymptotics that should be included to solve the saddle-point equations. However, we notice that in this case

$$S = \mathcal{G}(A, B) \approx m\kappa G\left(-\frac{A + B^2}{B}\sqrt{N}\right) \ll m \log \frac{1}{m} \tag{203}$$

Notice that in all solutions where $0 < c < 1$, the entropic potential $S \ll m \log \frac{1}{m}$ and does not contribute significantly to the FP potential $\phi_{\text{FP}}$ (175).

### G.3.3 Overlap equation

The equation related to the overlap of the probe system is

$$(r^2 - c)m = I_2(A, B) \tag{204}$$

Again, we study the three possible approximations of $I_2$:

- If $A \geq 0$, the magnetization is inconsistent since $I_2(A, B) = \mathcal{O}(1)$

- If $-B^2 < A < 0$, we have the equation

$$(r^2 - c)m = \frac{\exp\left(-\frac{A^2}{2B^2}N\right)}{\sqrt{2\pi N}}K(0, 2, A, B) = m(1 - c)\frac{K(0, 2, -\sqrt{2}|b|r, br)}{K(0, 1, -\sqrt{2}|b|r, |b|r)}, \tag{205}$$

where we replaced the exponent term with the expression founded for the magnetization equation (192) in this case. This yields :

$$\frac{r^2 - c}{1 - c} = \frac{K(0, 2, -\sqrt{2}|b|r, br)}{K(0, 1, -\sqrt{2}|b|r, |b|r)} = 1 - \frac{\sqrt{2}}{|b|r} \implies r^3 - r + \frac{1 - c}{|b|}\sqrt{2} = 0 \tag{206}$$

to be solved for $r \in [0, 1[$ and $|b|r > \sqrt{2}$.

- if $-2B^2 < A \leq -B^2$, we still having equation (205) but we need to consider the magnetization equation (198), the relation $A = -1 + \frac{1}{2}B$ and $\eta(r), \nu(r)$ from Eq. (194). Therefore, we have two sub cases:

– If $r = \sqrt{c}$, the equation (205) is consistent only if

$$\frac{\exp\left(-\frac{A^2}{2B^2}N\right)}{\sqrt{2\pi N}} \ll m \implies \frac{A^2}{2B^2} > 1 \implies A < -\sqrt{2}B. \tag{207}$$

Using the relation between $A$ and $B$:

$$1 - \sqrt{2}B + \frac{1}{2}B^2 = (1 - \frac{B}{\sqrt{2}})^2 > 0 \implies |b|r \neq \sqrt{2}. \tag{208}$$

Moreover, from the condition for $I_2$:

$$-2B^2 < -1 - \frac{1}{2}B^2 \implies b^2r^2 > \frac{2}{3}. \tag{209}$$

Gathering all the information:

$$r = \sqrt{c}; \qquad \frac{2}{3} < b^2c < 2; \qquad \nu = \frac{b}{2b_{\text{ref}}}c + \frac{1}{bb_{\text{ref}}} - c; \qquad \eta = \frac{b^2 + 2}{2b}. \tag{210}$$

– If $r > \sqrt{c}$, equation (205) is consistent only if

$$\frac{\exp\left(-\frac{A^2}{2B^2}N\right)}{\sqrt{2\pi N}} = \kappa_2 m \implies \frac{A^2}{2B^2} = 1 \implies A = -\sqrt{2}B \implies |b|r = \sqrt{2}, \tag{211}$$

where in the last line we use $A = -1 - \frac{1}{2}B^2$. Then the equation becomes

$$\frac{2}{b^2} - c = \kappa_2 K(0, 2, -2, \sqrt{2}), \tag{212}$$

to be solved for $\kappa_2$ through fixing of the sub-leading orders of the integral asymptotics. Additionally, we need to impose the integral condition

$$-2B^2 < A \implies -2B^2 < -\sqrt{2}B \implies \sqrt{2}B(\sqrt{2}B - 1) > 0 \implies |b|r > \frac{1}{\sqrt{2}}, \tag{213}$$

which is automatically satisfied given that $|b|r = \sqrt{2}$. Then, from the condition $r > \sqrt{c}$, we obtain

$$\frac{\sqrt{2}}{|b|} > \sqrt{c} \implies b^2c < 2. \tag{214}$$

Gathering all the information:

$$r = \frac{\sqrt{2}}{|b|} \qquad b^2c < 2; \qquad \nu = \frac{2}{bb_{\text{ref}}}; \qquad \eta = \frac{b^2 + 2}{2b}. \tag{215}$$

• If $A \leq -2B^2$, the equation is

$$(r^2 - c)m = m^{f_{A,B}(2)}G\left(-\frac{A + 2B^2}{B}N\right). \tag{216}$$

Again, we have two cases compatible with equation (198):

– If $r = \sqrt{c}$, equation (205) is consistent only if

$$m^{f_{A,B}(2)} \ll m \implies f_{A,B} < -1 \implies 2A + 2B^2 < -1 \implies B_1^2 < 1 \implies b^2c < 1, \tag{217}$$

where we use $A = -1 - \frac{1}{2}B$. Moreover, we have the condition $b^2c \leq 2$ and the integral condition

$$A \leq -2B^2 \implies -1 - \frac{1}{2}B^2 \leq -2B^2 \implies b^2c \leq \frac{2}{3} \tag{218}$$

Gathering all the information:

$$r = \sqrt{c} \qquad b^2c < \frac{2}{3}; \qquad \nu = \frac{b}{2b_{\text{ref}}}c + \frac{1}{bb_{\text{ref}}} - c; \qquad \eta = \frac{b^2 + 2}{2b}. \tag{219}$$

| Quantity | Solution 1 | Solution 2 | Solution 3 |
|---|---|---|---|
| Condition | $b^2 c < 2$ | $b^2 c < 2$ | $\lvert b\rvert r > \sqrt{2},\ r \geq \sqrt{c}$ |
| $r$ | $\sqrt{c}$ | $\frac{\sqrt{2}}{\lvert b\rvert}$ | Root: $r^3 - r + \frac{1-c}{\lvert b\rvert}\sqrt{2} = 0$ |
| $\nu$ | $\frac{b}{2b_{\rm ref}}c + \frac{1}{bb_{\rm ref}} - c$ | $\frac{2}{bb_{\rm ref}} - c$ | $\mathrm{sign}(b)\frac{\sqrt{2}r}{b_{\rm ref}} - c$ |
| $\eta$ | $\frac{b^2+2}{2b}$ | $\frac{b^2+2}{2b}$ | $\mathrm{sign}(b)\sqrt{2}r + \frac{b}{2}(1-r^2)$ |
| $\phi_{FP}$ | $\frac{b^2}{4}(1-c^2) + \frac{bb_{\rm ref}}{2}c^2 + 1 - c$ | $\frac{b^2}{4} + \frac{1}{b^2} + \frac{bb_{\rm ref}}{2}c^2 + 1 - 2c$ | $\frac{b^2}{4}(1-r^2)^2 + \frac{bb_{\rm ref}}{2}c^2 + \lvert b\rvert(1-c)\sqrt{2}r$ |
| $e$ | $\frac{b}{2}(1-c^2) + \frac{b_{\rm ref}}{2}c^2$ | $\frac{b}{2} - \frac{2}{b^3} + \frac{b_{\rm ref}}{2}c^2$ | $\frac{b}{2} + \frac{b'}{2}c^2 - \frac{b}{2}r^4$ |
| $a$ | $b(1-c^2) + b_{\rm ref}c^2$ | $b - \frac{4}{b^3} + b_{\rm ref}c^2$ | $b(1-r^4) + b_{\rm ref}c^2$ |
| $s$ | $1 - c - \frac{\left(a - a_{\rm ref}c^2\right)^2}{4(1-c^2)}$ | $1 - 2c - \frac{b^2}{4} + \frac{3}{b^2}$ | $-\frac{b^2}{4}(1-r^2)^2$ |

Table 2: Summary of the three solutions with their corresponding potentials, energy, and entropy densities.

– If $r > \sqrt{c}$, the equation is consistent if

$$m^{f_{A,B}(2)} = \kappa_3 m \implies 2A + 2B^2 = -1 \implies \lvert b\rvert r = 1. \tag{220}$$

This implies that the condition $A < -2B^2$ is not satisfied, and we should ignore this case.

## G.4 The small magnetization limit – Observables

Given the solutions of the saddle-point equations from the previous section, we can now compute the rescaled observables:

$$\text{rescaled Franz-Parisi: } \phi_{\rm FP} := \frac{\Phi_{FP}}{m \log(1/m)} = -\frac{b^2}{4}(1-r^4) - \frac{bb_{\rm ref}}{2}c^2 + b\eta - bb_{\rm ref}\nu c$$

$$\text{rescaled energy density: } e := \partial_b \phi_{\rm FP} \tag{221}$$

$$\text{rescaled submat average: } a := 2e$$

$$\text{rescaled entropy density: } s_{\rm FP} := \phi_{\rm FP} - be$$

We emphasize that the derivative with respect to $b$, used to compute the rescaled energy density, should be taken only after evaluating $\phi_{\rm FP}$ at the saddle-point solutions obtained in the previous section.

Notice that for $b^2 c < 2$, two solutions coexists in the same $(b,c)$ plane, and for $c = 1$, this region reduces to $b < \sqrt{2}$. Moreover, at $c = 1$, the probe configuration coincides with the reference configuration, and we expect the entropy to vanish, $s_{\rm FP} = 0$, and the energy to equal the equilibrium energy of the reference system, $e = \frac{1}{2}b_{\rm ref}$. This is indeed the case for solution 1, not for solution 2, and it holds for solution 3 in the regime where the branch has $r = 1$ branch. Additionally, at $c = 1$, we know that the Franz-Parisi potential should equal $b$ times the average energy of the reference system, i.e., $\phi_{\rm FP} = \frac{bb_{\rm ref}}{2} = be_{\rm ref}$, which again occurs only for solution 1 and for solution 3 when $r \approx 1$. On the other hand, at $c = 0$, we expect the probe system to recover the RS equilibrium solution at inverse temperature $b$ i.e. $e = \frac{b}{2}$, with non-zero entropy. This behavior is found in solution 1, but not in solution 2. Thus, we discard solution 2 in favor of solution 1.

We conjecture that solution 1 and solution 3 must be connected at some intermediate value of $c$. If this is the case, the observables should vary continuously across the transition. This imposes a continuity condition on the submatrix average, which can also be derived by requiring continuity of the overlap $r$:

$$b(1-c^2) + b_{\rm ref}c^2 = b(1-r^4) + b_{\rm ref}c^2 \implies r = \sqrt{c}. \tag{222}$$

Substituting this into the equation fo $r$ in solution 3, we obtain

$$0 = -(1-c)\sqrt{c} + \frac{1-c}{\lvert b\rvert}\sqrt{2} \implies c = 1 \text{ or } c = \frac{2}{b^2} \tag{223}$$

The condition $c = \frac{2}{b^2}$ identifies precisely the point where solution 1 ceases to be admissible.

### G.4.1 Observables in function of submatrix average and $c$

We want to study the FP potential as a function of $a$, $a_{\text{ref}}$ and $c$. In solution 1, this reads:

$$b = \frac{a - a_{\text{ref}}c^2}{1 - c^2}; \qquad s_{\text{FP}} = 1 - c\frac{(a - a_{\text{ref}}c^2)^2}{4(1 - c^2)}; \qquad b^2 c < 2 \implies \frac{|a - a_{\text{ref}}c^2|}{1 - c^2} < \sqrt{\frac{2}{c}} \tag{224}$$

While in solution 3:

$$b = \frac{a - a_{\text{ref}}c^2}{1 - r^4}; \qquad s_{\text{FP}} = -\frac{1}{4}\frac{(a - a_{\text{ref}}c^2)^2}{(1 + r^2)^2}; \qquad |b|r > \sqrt{2} \implies r\frac{|a - a_{\text{ref}}c^2|}{1 - r^4} > 2 \tag{225}$$

$$0 = -r(1 - r^2) + \frac{1 - c}{|a - ac^2|}(1 + r^2)(1 - r^2)\sqrt{2}; \qquad r \geq \sqrt{c}$$

Notice that $r = 1$ is a problematic point for inverting the relation $a \leftrightarrow b$, unless this occurs simultaneously with $a \to a_{\text{ref}}c^2$. At $r = 1$, we necessarily have $c = 1$, which gives $a = a_{\text{ref}}$ and $s_{\text{FP}} = 0$, as expected. Therefore, in the following, we consider the case $c \neq 1 \implies r \neq 1$. The equation for $r$ then becomes:

$$-r + \sqrt{2}\frac{1 - c}{|a - a_{\text{ref}}c^2|}(1 + r^2) = 0 \implies \frac{|a - a_{\text{ref}}c^2|}{1 + r^2} = \frac{1 - c}{r}\sqrt{2}. \tag{226}$$

This gives an alternative version for the entropy:

$$s_{\text{FP}} = -\frac{(1 - c^2)}{2r^2} \tag{227}$$

And for the condition

$$r\frac{|a - a_{\text{ref}}c^2|}{1 - r^4} > 2 \implies r\frac{1 + r^2}{1 - r^4}\frac{1 - c}{r}\sqrt{2} > \sqrt{2} \implies \frac{\sqrt{2}(1 - c)}{1 - r^2} > \sqrt{2} \implies r^2 > c \tag{228}$$

Thus, the equation for $r$ can be rewritten as

$$\sqrt{2}(1 - c)(1 + r^2) - |a - a_{\text{ref}}c^2|r = 0. \tag{229}$$

This equation admits two solutions:

$$r_{\pm} = \frac{|a - a_{\text{ref}}c^2| \pm \sqrt{(a - a_{\text{ref}}c^2) - 8(1 - c)^2}}{2\sqrt{2}(1 - c)}, \tag{230}$$

which has real solutions only if

$$|a - a_{\text{ref}}c^2| \geq 2\sqrt{2}(1 - c) \implies \frac{|a - a_{\text{ref}}c^2|}{2\sqrt{2}(1 - c)} \geq 1 \tag{231}$$

Notice that

$$r_+ \geq \frac{|a - a_{\text{ref}}c^2|}{2\sqrt{2}(1 - c)} \geq 1, \tag{232}$$

this is in contradiction with the condition $1 \geq r^2$, therefore, we keep only the solution

$$r = \frac{|a - a_{\text{ref}}c^2| - \sqrt{(a - a_{\text{ref}}c^2) - 8(1 - c)^2}}{2\sqrt{2}(1 - c)}, \tag{233}$$

under the conditions $|a - a_{\text{ref}}c^2| \geq 2\sqrt{2}(1 - c)$ and $r \geq \sqrt{c}$. From the later condition, we have :

$$|a - a_{\text{ref}}c^2| - \sqrt{(a - a_{\text{ref}}c^2) - 8(1 - c)^2} > 0 \implies \frac{|a - a_{\text{ref}}c^2|}{1 - c^2} \leq \sqrt{\frac{2}{c}} \tag{234}$$

## G.5 Analysis of solutions

Solutions in Table 2 were derived under the assumption that $0 \leq c < 1$, we conjecture that one of this solutions coincide with the case $c = 1$ by continuity. The entropy of solution 1 reads:

$$s = 1 - c - \frac{(a - a_{\text{ref}}c^2)^2}{4(1 - c^2)} \qquad \text{for} \qquad \frac{|a - a_{\text{ref}}c^2|}{1 - c^2} < \sqrt{\frac{2}{c}} \tag{235}$$

and the entropy of solution 3:

$$s = -\frac{(1 - c)^2}{2r^2} \qquad \text{for} \qquad 1 > r \geq \sqrt{c} \tag{236}$$

We observe that solution 3 is always negative for $c < 1$, and zero entropy in the limit $c \to 1$. Conversely, solution 1 exhibits negatively divergent entropy i.e. $s \to -\infty$ for $a \neq a_{\text{ref}}$ as $c \to 1$. The case $c = 0$ coincides with the entropy of solution 1 and must be the global optimizer, as it shares the same positive entropy as the equilibrium solution. Thus we know that:

- In both solution 1 and solution 3, $c = 1$ implies $a = a_{\text{ref}}$. This is evident in the temperature represen­tation, and also from the energy expression obtained from solution 1, since the entropy at $c = 1 - \epsilon$, for $1 > \epsilon > 0$, is given by

$$s = \epsilon - \frac{(a - a_{\text{ref}})^2}{8\epsilon} \xrightarrow{\epsilon \to 0} -\infty, \tag{237}$$

  which diverges to negative infinity.

- The condition $c = 1$, implying $a = a_{\text{ref}}$, corresponds to a local optimizer only in the frozen phase $\sqrt{2} < a_{\text{ref}} < 2$. In this case, the entropy of solution 1 has a derivative at $c = 1$ and $a = a_{\text{ref}}$ given by

$$\partial_c s_{\text{sol 1}}|_{c=1, \, a=a_{\text{ref}}} = \frac{a^2}{2} - 1, \tag{238}$$

  which is positive if and only if $a > \sqrt{2}$. Thus, when solution 1 determines the FP curve around $c = 1$, this point is not a local maximum. On the other hand, if solution 3 determines the FP curve near $c = 1$, then $c = 1$ is a local maximum, as the entropy in solution 3 is negative everywhere except at $c = 1$. Note that in solution 3, the entropy at $c = 1$ is zero, as expected, since it represents the internal entropy at equilibrium in the frozen phase.

Now, we are interested in the following, less trivial, properties of the entropy and aim to address the two questions below:

- Is the set where the entropy is positive connected?

- Are there any other local maxima with positive entropy for $0 < c < 1$?

Both questions can be addressed by considering solution 1 alone, since we know that in solution 3 the entropy is strictly negative for all $c < 1$.

**Regions of positive entropy:** We need to impose the conditions

$$s = 1 - c - \frac{(a - a_{\text{ref}}c^2)^2}{4(1 - c^2)} > 0$$

$$\sqrt{\frac{2}{c}} > \frac{|a - a_{\text{ref}}c^2|}{1 - c^2}. \tag{239}$$

Numerically, we see that

- For $a_{\text{ref}} \leq \sqrt{2}$ and $a = a_{\text{ref}}$, the entropy is positive for all $c$.

- For $\sqrt{2} < a_{\text{ref}} < 2$ and $a = a_{\text{ref}}$, the entropy is positive in the interval $[0, 4/a^2 - 1]$, then negative in $[4/a^2 - 1, 2/a^2]$ connecting to solution 3 in $c = 2/a^2$ and it is zero at $c = 1$.

- For $a_{\mathrm{ref}} \leq \sqrt{2}$ and $a \neq a_{\mathrm{ref}}$, the entropy is positive in an interval $[0, c_1(a, a_{\mathrm{ref}})]$, and then is negative.

- For $\sqrt{2} < a_{\mathrm{ref}} < 2$ and $\sqrt{2} < a < 2$, the entropy is positive in an interval $[0, c_1(a, a_{\mathrm{ref}})]$.

- For $\sqrt{2} < a_{\mathrm{ref}} < 2$ and $a_*(a_{\mathrm{ref}}) < a < a_{\mathrm{ref}}$, the entropy is positive in an interval $[0, c_1(a, a_{\mathrm{ref}})] \cup [c_2(a, a_{\mathrm{ref}}), c_3(a, a_{\mathrm{ref}})]$ with $c_1 \leq c_2 \leq c_3$; and negative everywhere else.

- For $\sqrt{2} < a_{\mathrm{ref}} < 2$ and $a < a_*(a_{\mathrm{ref}})$, the entropy is positive in an interval $[0, c_1(a, a_{\mathrm{ref}})]$, and is negative everywhere else.

The analytical values of $c_{1,2,3}$ can be computed case by case in Mathematica by imposing the respectively conditions, which also suggest that the analytical expresion of $a_*(a_{\mathrm{ref}})$ is a root of

$$p_*(a) = -128 + 107a_{\mathrm{ref}}^2 + 64a_{\mathrm{ref}}^4 + (-6a_{\mathrm{ref}} - 72a_{\mathrm{ref}}^3)a + (27 - 24a_{\mathrm{ref}}^2 - 16a_{\mathrm{ref}}^4)a^2 + 16a_{\mathrm{ref}}^3 a^3 \qquad (240)$$

**Maxima of the entropy:** We start by studying the stationary points of the entropy. They solve:

$$\partial_c s = -\frac{c(a - a_{\mathrm{ref}}c^2)^2}{2(1 - c^2)^2} + \frac{a_{\mathrm{ref}}c(a - a_{\mathrm{ref}}c^2)}{1 - c^2} - 1 \overset{!}{=} 0 \qquad (241)$$

which gives an order-five polynomial equation for c. Numerically, we see that:

- For $a_{\mathrm{ref}} < \sqrt{2}$ there is no non-trivial local maximum.

- For $\sqrt{2} < a_{\mathrm{ref}} < 2$ and $a < a_{**}(a_{\mathrm{ref}})$ there is non-trivial local maximum.

- For $\sqrt{2} < a_{\mathrm{ref}} < 2$ and $a_{**}(a_{\mathrm{ref}}) < a < a_{\mathrm{ref}}$ there is non-trivial local maximum.

- For $\sqrt{2} < a_{\mathrm{ref}} < 2$ and $a > a_{\mathrm{ref}}$ there is no non-trivial local maximum.

Notice that $a_{**}(a_{\mathrm{ref}}) \leq a_*(a_{\mathrm{ref}})$, as having disconnected positive entropy regions imply existence of a non-trivial local maximum.