# Peer review of "The maximum-average subtensor problem: equilibrium and out-of-equilibrium properties"

_SciPost Physics_

## Round 1 · Referee Report · Anonymous (Referee 1) · 2025-9-16

Strengths

  1. The introduction is well written and explains some of the intricacies involved with predicting typical-case algorithmic hardness.
  2. The authors introduce a new model, which is natural and merits further exploration.
  3. The authors give a thorough analysis of the new model from various different angles.

Weaknesses

  1. For the most part, the findings are qualitatively the same as the previously-studied matrix case, so perhaps not so surprising. Still, this is interesting to know and could not have been predicted ahead of time.
  2. The results fall short of resolving the major question of pinpointing the algorithmic threshold. Still, I find it valuable to report on the attempts that were tried here.

Report

This paper studies the Maximum-Average Subtensor problem, a natural optimization problem where the input is an N-by-...-by-N random tensor (multi-way array filled with Gaussian random variables) and the goal is to find a subtensor of size k-by-...-by-k such that the sum of entries is as large as possible. The matrix case (where the tensor has order p = 2) has been studied before, and this paper is the first to consider tensors of order p > 2.

The introduction gives a nice account of typical-case algorithmic hardness. Various theoretical frameworks exist for predicting which solutions (or which objective values) are accessible by computationally efficient methods, but the relations among these are somewhat intricate and not completely understood. This paper is particularly concerned with exploring whether or not certain physics-style phase transitions such as "frozen 1-RSB" (where most solutions are isolated) are indicators of algorithmic hardness. The authors argue that the Maximum-Average Submatrix problem is a good testbed to explore this, since it is a simple model that was recently shown to exhibit a surprising phenomenon that challenged the conventional wisdom: a frozen 1-RSB phase that is nonetheless algorithmically accessible. This motivated the authors to consider the tensor case, which is a natural extension.

The paper carries out a rather thorough analysis of the phase transitions and algorithmic thresholds, from a few different angles. One finding is that, similar to the matrix case, there is a frozen 1-RSB phase which is nonetheless algorithmically accessible using a greedy algorithm called IGP. On the other hand, a different greedy algorithm called LAS stops working precisely at the freezing threshold. This leaves the question of whether IGP is optimal among efficient algorithms, and whether the threshold achieved by IGP coincides with any physics-style phase transition. The authors explore two different analytic approaches in an attempt to shed light on this. First, the authors obtain some non-matching upper and lower bounds on the algorithmic threshold via the Overlap Gap Property (OGP), and they mention that a concurrent and independent work has used a more refined version of OGP to pin down the precise threshold and prove that IGP is in fact optimal within a large class of methods. Second, the authors use a Franz-Parisi analysis; while this sheds light on the structure of certain rare clusters in the solution space, the authors discover that it does not appear to have any clear connection to the algorithmic threshold.

Requested changes

  1. pg 2: "an replica symmetric" -> a
  2. pg 3: "upper bound base on" -> based
  3. pg 3: "Table 1 subs up" -> sums
  4. pg 3: "still find [it] surprising that such a simple..."
  5. pg 4 (and throughout): I believe p would typically be called the "order" (not "rank") of the tensor, since "rank" has a different meaning similar to matrix rank

Recommendation

Publish (meets expectations and criteria for this Journal)

  • validity: high
  • significance: good
  • originality: good
  • clarity: high
  • formatting: excellent
  • grammar: excellent

Author:  Vittorio Erba  on 2025-11-12  [id 6021]

(in reply to Report 1 on 2025-09-16)

We thank Referee 1 for taking the time to read our work and provide feedback. They raised the following points, to which we answer.

Results are qualitatively the same as the matrix case.

We agree with Referee 1 that the findings are not qualitatively different from the matrix case, but while this is not surprising a a posteriori, it is still non-trivial. For example, it is well known [1] that the classic binary p-spin model at finite magnetisation m has different behaviour at p=2 (RS to full-RSB transition) and p > 2 (RS to 1-RSB to full-RSB transition). Under this light, the fact that the nature of the transitions is the same for p=2 and p>2 for Boolean spins and in the limit m to zero is non-trivial.

Results do not shed full light on the algorithmic properties of the problem.

We actually find this point to be an important observation to share with the community, as it indicates a failure mode of the tools of disordered physics when applied to computational problems. We find that understanding from first principles when (for which problems, under which assumptions) the standard statistical physics toolbox is predictive, and when it is not is an important open problem, and here we provide an analytically tractable negative result.
We will add further commentary in the resubmitted version of the manuscript (in the introduction) to better highlight these aspects.

Typos and nomenclature.

We will correct all typos, and change the notation from "rank" to the more appropriate "order" one, in the resubmitted version.

References

[1] Gardner E. 1985. Spin glasses with p-spin interactions. Nucl. Phys. B 257 747

---

## Round 1 · Referee Report · Anonymous (Referee 2) · 2025-10-4

Strengths

  1. The paper tackles an interesting average-case tensor optimization problem in the Gaussian setting, extending a line of work that had previously been explored only in the matrix regime. The problem itself exhibits rich and intricate algorithmic and geometric phenomenology.

  2. The paper approaches this phenomenology from several perspectives, including: analysis of two algorithms (LSA and IGP), replica calculations of the free energy (yielding the freezing point), as well as variants of the Overlap Gap Property (OGP) and Franz–Parisi (FP) landscape analysis.

Weaknesses

  1. The analysis of the algorithms is underdeveloped and lacks rigor (see detailed comments below).

  2. The landscape-based explanations for algorithmic hardness remain inconclusive and do not provide a definitive account (see below).

Report

The paper considers the maximum k-subtensor problem for a Gaussian tensor, where the goal is to find the k×⋯×k subtensor (henceforth referred to as a solution) with the highest possible average value. The authors focus on the scaling k=m⋅n and then send m→0.

This problem has previously been studied in the matrix case (ignoring concurrent work).

The authors (non-rigorously) identify:

  1. the maximum possible value of a solution,

  2. the freezing value (above which all solutions with that value are isolated),

  3. the performance of two greedy algorithms for the task (LSA and IGP),

  4. an annealed variant of the Overlap Gap Property (OGP), and

  5. the Franz–Parisi landscape of the problem.

Parts (1), (2), (4), and (5) rely on replica calculations (followed by sending m→0). Part (3) analyzes the algorithms under the assumption that each iteration encounters a fresh Gaussian tensor; the authors suggest that this analysis might eventually be made rigorous.

A key finding is that the freezing value is strictly smaller than the maximum possible value. Traditionally, this would suggest that polynomial-time algorithms should fail to surpass the freezing threshold. However, recent results have complicated this picture. In fact, the analysis of IGP suggests that an efficient algorithm does cross this threshold. The authors attempt to explain this apparent contradiction using landscape-based perspectives, but the discussion remains inconclusive.

Overall, I find the choice of problem compelling, as it highlights the gap between freezing thresholds and algorithmic performance. However, I have several concerns:

  1. Lack of rigor in the algorithmic analysis.

While the replica-based parts are standard in this line of work, the analysis of algorithms is more problematic. For LSA, the authors rely on [1], which (i) only characterizes a typical local maximum—something LSA is not guaranteed to find—and (ii) applies only when k=O(1), whereas here k=mn with m→0. For IGP, the assumption of independent iterations is too much to just "assume". The claim that ideas from [2] can address this issue is unconvincing, since [2] only applies when k=o(logn) and it is quite tricky whether this generalizes to any other growing k, let alone almost linear. In my view, the authors should make a serious effort to analyze the algorithms on more rigorous grounds (or provide more evidence for them).

  1. Inconclusiveness of the OGP and Franz–Parisi analyses.

Both analyses leave more to be wanted, as the main point of the paper -- the algorithmic hardness-- is unresolved. Assuming the IGP analysis is correct, one might ask whether the algorithm’s ascending nature—systematically climbing the landscape—explains its ability to surpass the freezing value. The super-level set FP analysis seems to suggest this possibility, but it is unclear whether the authors obtain such an interpretation or not. Achieving such an intepretation would strengthen significantly the paper.

Requested changes

  1. Improve the algorithm's analysis to make a more solid claim on the algorithm's performance.

  2. Attempt to provide some explanation of hardness, perhaps following my suggestion on trying to understand whether the ascending nature of IGP is leveraging some geometric property.

Recommendation

Ask for major revision

  • validity: low
  • significance: good
  • originality: good
  • clarity: ok
  • formatting: good
  • grammar: good

Author:  Vittorio Erba  on 2025-11-12  [id 6022]

(in reply to Report 2 on 2025-10-04)

We thank Referee 2 for taking the time to read our work and provide feedback. Across their report, they raised the following points, to which we answer.

The analysis of the algorithms is underdeveloped and lacks rigor.

Convergence of the LAS algorithm. The referee correctly points out that we only adapted a proof of the structure theorem from [1], and not the matching convergence proof by [2], for the LAS algorithm. We clarified this better in the new version of the manuscript (Section 3.2). We do not think adapting the convergence result of [2] would bring any additional insight here, and we expect there would be no roadblocks as it was the case for the structure theorem.

Analysis of the IGP algorithm. The heuristic derivation we provide here for the final energy of the IGP algorithm for $p \geq 2$ matches with a concurrent rigorous derivation for the same value [4,5], holding for $k= \exp(o(\log n))$, thus confirming the validity of our prediction basically up to almost-linear scaling.

On the relationship between the setting $1 \ll k \ll N$ and $k = mN$ with $m\ll 1$. We agree with the referee that a priori the setting of sub-linear sized subtensors $1 \ll k \ll N$ may behave differently from small, linearly-sized subtensors $k = mN$ with $m\ll 1$. We remark that a similar exchange of limits has been used in several other problems (see [3] , appendix C.1 and C.2, for a recent example) where it has been empirically validated by simulations, or by explicit computation of the complementary limit. Here the computational complexity of dealing with tensors scaling as $O(k^p)$, and the necessity of scaling $k$ with $N$, make such numerical verification very hard. Rigorously adapting the prediction from statistical physics tools to the $k \ll N$ regime is hard (we are not aware of any results in this regime, across all spin glass models commonly studied), and adapting the proof techniques used in the $k \ll N$ regime to the linear scaling seems also hard (see for example [5], Remark 2.8, posted on arXiv just after our submission of this work, where the linear $k$ regime is only studied for large enough tensor order $p$ thanks to convergence to a simpler Random Energy Model behaviour). In conclusion, we think that the heuristic level of presentation we provide in this manuscript is the best we could do, and going beyond to provide a proper comparison between the $1 \ll k \ll N$ regime and $k = mN$ with $m\ll 1$ regime would require significant technical advances. We will comment on this in the resubmitted manuscript (Section 2.2).

Inconclusiveness of the OGP and Franz–Parisi analyses.

As discussed also in the answer to Referee 1, we find that the inconclusiveness of our results is a negative result worth sharing, as it highlights the failure of standard statistical physics tools to predict algorithmic properties (at odds with what happens in other models, such as the colouring on random graphs, and hence at odds to the common knowledge). To rephrase, our results could be interpreted as "We exhibit a model where standard clustering (OGP) and Franz-Parisi tools do not shed light on the behaviour of optimal, simple greedy algorithms". We will add further commentary in the resubmitted manuscript (in the introduction) to better highlight these aspects.

Interpretation of the Franz-Parisi analysis.

Referee 2 states: "Assuming the IGP analysis is correct, one might ask whether the algorithm’s ascending nature—systematically climbing the landscape—explains its ability to surpass the freezing value. The super-level set FP analysis seems to suggest this possibility, but it is unclear whether the authors obtain such an interpretation or not." Our Franz-Parisi analysis was indeed motivated by this intuition, i.e. that clusters around sub-tensors of large energy (uniformly sampled under the equilbrium measure) are surrounded by clusters of lower-energy configurations, which may act as "bridges" between easily accessible equilibrium energy states (lower than the freezing threshold) and those that are hard to find (larger than the freezing threshold). What we find though does not support this intuitive picture. The Franz-Parisi analysis indeed shows that there are no such bridges (all clusters around hard to find equilibrium configurations merge at an energy higher than freezing), and in any case all the thresholds we may identify in the Franz-Parisi analysis do not correlate with the IGP convergence energy. This means that surely IGP is not probing configurations that are close in Hamming distance to equilibrium, large energy configurations, making this algorithmic behaviour crucially out-of-equilibrium. We will add further commentary in the resubmitted manuscript (in Section 5) to better highlight these aspects.

References

[1] Shankar Bhamidi, Partha S. Dey, and Andrew B. Nobel. Energy landscape for large average submatrix detection problems in gaussian random matrices. Probability Theory and Related Fields, 168(3):919–983, 2017. [2] David Gamarnik and Quan Li. Finding a large submatrix of a Gaussian random matrix. The Annals of Statistics, 46(6A):2511 – 2561, 2018. [3] Maillard, A., Troiani, E., Martin, S., Krzakala, F., & Zdeborová, L. (2024). Bayes-optimal learning of an extensive-width neural network from quadratically many samples. Advances in Neural Information Processing Systems, 37, 82085-82132. [4] Bhamidi, S., Gamarnik, D., & Gong, S. (2025). Finding a dense submatrix of a random matrix. Sharp bounds for online algorithms. arXiv preprint arXiv:2507.19259. [5] Abhishek Hegade K, R., & Kızıldağ, E. C. (2025). Large Average Subtensor Problem: Ground-State, Algorithms, and Algorithmic Barriers. arXiv e-prints, arXiv-2506.

---

## Round 3 · Referee Report · Anonymous (Referee 1) · 2025-12-8

Report

I believe the authors have provided a thorough response to the reviewer comments. In particular, I believe they make a good argument for why the paper falls within the scope of physics. Really, it is a paper that combines ideas from both computer science and physics. There are many such papers, and there are both "physicists" and "computer scientists" who care about them (and also many researchers who do not fall squarely into one of those two buckets). Often these papers are published in computer science or math venues, but this particular one is more appropriate for a physics venue (in my opinion) because the analysis is done in the style of physics (whereas a math venue would expect more rigorous theorems and proofs). Personally, I welcome this type of interdisciplinary work and recommend the paper be accepted.

Recommendation

Publish (meets expectations and criteria for this Journal)

---

## Round 3 · Referee Report · Anonymous (Referee 3) · 2025-12-12

Report

The Journal acceptance criteria are met for this paper. The paper analyses the maximum average subtensor problem with techniques developed from statistical physics, such as replica computations, or Franz-Parisi potential, thus it is an interdisciplinary paper that well suited the area of the Journal. The authors have answered the previous suggestions raised by the referee. It is true that the Franz-Parisi analysis leads no insight into the algorithmic threshold for IGP, but this is a negative results that I believe it is worth to stress.

I only suggest to better cite ref. [37]. There is a specific paragraph "Concurrent work: optimality through branching OGP" that explains why this concurrent work, appeared while finalizing the manuscript, is an improvement of the actual sec. 5 of the manuscript. However, in the conclusions this work is not mentioned and it is written "Natural directions to extend our work involve [...] considering more complicated OGPs such as the branching OGP" This direction has already been followed in ref. [37] and it should be mentioned in the conclusions.

Moreover I slightly disagree with the last sentence of the conclusions: "Finally, understanding which subclass of efficient algorithms do not work in equilibrium frozen 1-RSB phases (besides those that respect in some sense the detailed balance) would be interesting." It is true that algorithms that respect detailed balance cannot do sampling efficiently in the frozen phase. However they could still find a good solution to the problem,working out-of-equilibrium without sampling the entire space. This difference, between sampling and finding a single solution, should be stressed.

Requested changes

I list some typos or unclear sentences that could be corrected: - In the definition of the binary perceptron model, pag 3: specify that the vector w should be such that $|\xi^\mu\cdot w|\geq K \sqrt{d}|$ for each $\mu$ - When reference [3] is cited, the authors use the pronoun "we" to cite the achievements of that work. However, the authors of [3] are not the same as the authors of this work, thus I find more convenient to write "the authors of [3]" insetad of "we" (see pagg. 7 , 12, 13) - in eqs. (2) and (3) the subtesor average are defined for general and symmetric tensors, and indicated with the same notation $avg(\sigma)$. However the definitions are different, thus I suggest to call them in a different way, as done for E and m in eqs. (5) and (6). In the same way, I would split eq 7 in two. Moreover, in eq (2) the avg is normalized by N while in (3) it is normalized by k - Pag. 10, it is written "The magnetic field h, intre-state overlap $q_1$ ... satisfy the associated saddle-point equations" : however it seems to me that it is m instead of h to satisfy the saddle-point equation - after eq. (19): "e is related to the sub-tensor average as defined in (13)" Isn't it better to cite eq. (7) instead of (13)? - pag. 12: "they have intra-cluster average overlap $q_1/m=0$" should be substituted by $q_1/m=1$ - pag. 13: "the freezing temperature being zero for the REM" should be substituted by inverse freezing temperature, and also at the end of the paragraph. - Algorithm 2:"$I^a={i^a}$ for all a" it is better to add $1\leq a\leq p$ - Algorithm 3: "Initialize by selecting k indices at random for each dimension" It is better to add that this procedure initialize the $I^a$ - The LAS and IGP algorithms are indicated sometimes with calligraphic letters and sometimes with plain letters. I suggest to unify the notation - pag 17: "OGP bound hints at algorithmic hardness around $q_{max}$" should be replaced by $a_{max}$ - In the caption of Fig. 2 at Pag 22 and also at pag. 23: for $a_{crit,2}(a_{ref})<a<a_{max}$ should be replaced by $a_{crit,2}(a_{ref})<a<a_{ref}$

Recommendation

Publish (meets expectations and criteria for this Journal)

---

## Round 3 · Referee Report · Anonymous (Referee 4) · 2025-12-24

Strengths

1- The paper addresses an important and timely matter. 2- The paper provides solid negative evidence, highlighting a counterintuitive phenomenon and providing importrant direction to future research. 3- The paper is well-written and well-motivated (especially remarkable in light of the large scope and wide variety of techniques employed).

Weaknesses

1- The paper is mostly self-consistent, but it often relies on ref. [3] and some concepts are hard to grasp without reading ref. [3]. 2- Even though most sections of the paper are well motivated, I am not sure I fully understood the motivation for the extension to the cases with p>2.

Report

This paper addresses an important and timely matter the overlap gap property and its relation to the free energy landscape. It provides evidence that an entire family of problems can be solved beyond the 1RSB-freezing threshold, which is a counter-intuitive phenomenon at the center of the discussion. Moreover, the paper provides strong suggestions that out-of-equilibrium phenomena may be the correct explanation, providing importrant direction to future research.

I believe that it fits well the scope of the journal: while the subject may seem far from more common topics in physics, the methods employed fit with a tradition of statistical physics of disordered systems that had great success in dealing with problems from computer science.
Moreover, the discussion around the Overlap Gap Property and its relation to the shape of the free energy landscape is a hot topic, where theoretical physics can give an important contribution. In fact, while the Overlap Gap Property has its roots in computational complexity, similar concepts of computational hardness appeared early in the development of spin glass theory (see the SK model) and physics of supervised learning (see Gardner's theory). Overall, this theoretical tools are important not just for optimization problems, but also to build a theory of AI, and pushing their development further is an important and timely matter.

Requested changes

Minor comments, just to improve the clarity of the manuscript. 1- The authors say that region of algorithmic interest for this problem in the matrix case p = 2 was found to be the limit m → 0. If I understand correctly, this comes from the previous work of ref. [3]. Can this statement be made more explicit? As m is related to the sub-tensor size, it seems to me that you are saying that the region of algorithmic interest is when the sub-tensor size is sub-extensive, but I could not understand why. 2- The introduction makes a great job in explaining the problem, the motivation and the program of the paper. What is not clear to me is if there is a specific motivation to study the p-MAS extension. If I understand correctly from section 5.3, the main motivation for extending the MAS problem to p>2 was to study the effectiveness of annealed OGP bounds for a larger family of problems. Whether this is the case or not, it would help if the authors could motivate the choice more explicitly in the introduction.

Typos: - figure 1, y axis: "rescaled sutensor" - section 7: "In this paper we studies"

Recommendation

Publish (easily meets expectations and criteria for this Journal; among top 50%)

---

## Round 3 · Author Response

We thank the Editor and the Referees for investing time and effort in the evaluation of our manuscript. We replied to the Referees, and we resubmit an improved version of the manuscript. We answer here to the points the Editor raised.

Work is peripheral to the scope of SciPost Physics.

We did not find an explicit argument relating to this in the Referee reports and in the Editorial Recommendation, so we can only answer in general terms. We believe that our work is well within the scope of SciPost Physics: - Our object of study is a timely computational problem, see the concurrent appearance of our work, [1] and [2], which are more mathematical, but still rooted in the physics of spin glasses. For e.g., [2] adapts ideas from the rigorous study of p-spin models at large p, which has been first studied heuristically by Derrida in his classic paper [5]. - Our toolbox is that of statistical physics of disordered systems, using replica theory, Franz-Parisi analysis, and the clustering OGP analysis to probe equilibrium and out-of-equilibrium behavior of the system. We compute entropies, we look for phase transitions. Our problem may be born in computer science, but the questions we ask are those physicists would ask, and the tools we use all appeared first in the study of spin glasses. We remark that questions relating to algorithmic hardness were explored by recent published papers in this venue [3,4], written by physicists, using a physics toolbox, and studying models from computer science.

We wrote the paper as physicists interested in studying computational problems with statistical mechanics, for physicists interested in studying computational problems with statistical mechanics. All in all we believe that our paper is appropriate for a physics audience.

Address the concerns and requested changes of the Referees.

We answered point-by-point the reports of the referees, and implemented their suggestions. See the answers under the original submission, and the list of changes below.

Formatting.

We adapted our formatting to the guidelines, and polished the references. In particular, we reshaped the introduction (which contained already a discussion of perspectives), and added a corresponding mandatory conclusion section.

Relationship with [1].

In [1], the authors study the p-MAS problem in the setting of both linear and sublinear size, for large values of the tensor order p. It is well known that the p-spin model (of which the p-MAS is close, Boolean relative) converges to a Random Energy Model as p increases [5]: the authors of [1] leverage this property (in its non-trivial rigorous formulation) to provide algorithmic bounds for the problem. On the other hand, in our work we explore landscape properties in the limit k/N→0, but for finite tensor order p, thus far from the Random Energy Model limit. We also remark that our results at large p (Section 2.4) are compatible with [1]. In summary, [1] is yet another concurrent and complementary work on the same model, together with [2] and with ours, signalling the interest of the larger interdisciplinary community in the study of this problem.

Multiple arXiv versions.

Versions v1 and v2 on the arXiv are identical, except for the way we cited [2] (at the time of submission [2] was still an unpublished work, and in v1 not all authors names were correctly listed). We were not aware of the existence of [1], which appeared on the arXiv after both our submission of v2 to arXiv, and our original submission to SciPost. We now comment on [1] in the manuscript.

References

[1] Abhishek Hegade K, R., and Eren C. Kızıldağ. "Large Average Subtensor Problem: Ground-State, Algorithms, and Algorithmic Barriers." 2025, arXiv: 2506.17118. [2] Bhamidi, S., Gamarnik, D., & Gong, S. (2025). Finding a dense submatrix of a random matrix. Sharp bounds for online algorithms. arXiv preprint arXiv:2507.19259 [3]: Barbier, D. (2025). How to escape atypical regions in the symmetric binary perceptron: a journey through connected-solutions states. SciPost Physics, 18(3), 115. [4]: Annesi, B. L., Malatesta, E., & Zamponi, F. (2025). Exact full-RSB SAT/UNSAT transition in infinitely wide two-layer neural networks. SciPost Physics, 18(4), 118. [5]: B. Derrida. Random-energy model: Limit of a family of disordered models. Phys. Rev. Lett., 45:79–82, Jul 1980

---

## Round 3 · List of Changes

• After Table 1: we now comment on the negative result of the Franz-Parisi analysis, as argued in the answer to both Referees.
  • Paragraph "Concurrent work: algorithmic properties for large $p$": we now compare with concurrent work [1], which appeared after our submission.
  • Last paragraph of Section 3.2: we comment on the difference between the case 1 << k << N, and k/N = m with m->0.
  • Section 4.1, after Algorithm 2: we stress that the IGP analysis has been put on rigorous grounds by concurrent work [1,2].
  • Last paragraph of Section 4.2: we remark that the LAS analysis is structural, and that we are not adapting the convergence proof.
  • Second paragraph of Section 6: we added a paragraph with more intuition on the Franz-Parisi analysis, along the lines of our discussion with Referee 2.
  • Section 7: we added a Conclusion section, including lines 861-868 that previously appeared in the introduction ("Perspectives" paragraph).
  • Typos: we corrected several typos, including those highlighted by Referee 1.
  • Nomenclature: we changed "rank" to "order" when referring to the number of tensor indices as suggested by Referee 1.

---

## Round 4 · Author Response

We thank the editor and the referees for their comments and feedbacks on our work.
We implemented all the suggestions of the referees, correcting typos and clarifying phrasings.

---

## Round 4 · List of Changes

• Added "and for each $\mu$" at page 3 in the definition of the binary perceptron
  • before paragraph "Concurrent work: optimality of $\IGP$ through branching OGP for $k\ll N$.": added a comment on the relevance of p>2
  • eq (2): corrected a typo, i.e. denominator from N_i to k_i, and added a remark just before to define k_i
  • Modified "we" to "the authors" when citing [3] on page 7, 9, 12, 13, 16.
  • Page 7, just before section 2.2: added comment stressing that eq (2) and (3) hold for both symmetric and non-symmetric tensors.
  • eq (7): we split the symmetric and non-symmetric cases on two lines, and added a remark just before to stress this
  • page 10, after eq 16: corrected "magnetic field" to "magnetization"
  • after eq (19) added "rescaled" to "subtensor average" to clarify why we cite eq (13)
  • beginning of section 3.2: added discussion on why we consider m->0 limit
  • after eq (24): corrected from "have intra-cluster average overlap $q_1/m = 0$" to "have intra-cluster average overlap $q_1/m = 1$"
  • page 13, just before section 3.5, modified "temperature" to "inverse temperature"
  • Algorithm 2: added specification that $1 \leq a \leq p$
  • Algorithm 3: better specified the initialization condition
  • after eq 36: corrected q_max to a_max
  • modified a_max to a_ref in caption of Figure 2, and in Section 6.2.3, 3rd bullet point.
  • Conclusion: added remarks in second paragraph as suggested by referee 2 on branching OGP being done by [37], and on the difference between sampling and finding a single configuration.
  • corrected author lists of [4,38]
  • uniformed notation for LAS and IGP across the paper

---

## Editorial Decision

voting_in_preparation